# Generalized equivalences between subsampling and ridge regularization

**Pratik Patil**

Department of Statistics

University of California
Berkeley, CA 94720, USA
`pratikpatil@berkeley.edu`

**Jin-Hong Du**

Department of Statistics and Data Science
& Machine Learning Department
Carnegie Mellon University
Pittsburgh, PA 15213, USA
`jinhongd@andrew.cmu.edu`

## Abstract

We establish precise structural and risk equivalences between subsampling and ridge regularization for ensemble ridge estimators. Specifically, we prove that linear and quadratic functionals of subsample ridge estimators, when fitted with different ridge regularization levels $\lambda$ and subsample aspect ratios $\psi$, are asymptotically equivalent along specific paths in the $(\lambda, \psi)$-plane (where $\psi$ is the ratio of the feature dimension to the subsample size). Our results only require bounded moment assumptions on feature and response distributions and allow for arbitrary joint distributions. Furthermore, we provide a data-dependent method to determine the equivalent paths of $(\lambda, \psi)$. An indirect implication of our equivalences is that optimally tuned ridge regression exhibits a monotonic prediction risk in the data aspect ratio. This resolves a recent open problem raised by Nakkiran et al. [1] for general data distributions under proportional asymptotics, assuming a mild regularity condition that maintains regression hardness through linearized signal-to-noise ratios.

## 1 Introduction

Ensemble methods, such as bagging [2, 3], are powerful tools that combine weak predictors to improve predictive stability and accuracy. This paper focuses on sampling-based ensembles, which exhibit an implicit regularization effect [4–6]. Specifically, we investigate subsample ridge ensembles, where the ridge predictors are fitted on independently subsampled datasets [7–9]. Recent work has demonstrated that a full ensemble (that is fitted on all possible subsampled datasets) of ridgeless [10] predictors achieves the same squared prediction risk as a ridge predictor fitted on full data [11–13].

To be precise, let $\phi$ be the limiting dataset aspect ratio $p/n$, where $p$ is the feature dimension, and $n$ is the number of observations. For a given $\phi$, the limiting prediction risk of subsample ridge ensembles is parameterized by $(\lambda, \psi)$, where $\lambda \geq 0$ is the ridge regularization parameter and $\psi \geq \phi$ is the limiting subsample aspect ratio $p/k$, with $k$ being the subsample size [11, 12]. Under isotropic features and a well-specified linear model, [11, 12] show that the squared prediction risk at $(0, \psi^*)$ (the optimal ridgeless ensemble) is the same as the risk at $(\lambda^*, \phi)$ (the optimal ridge), where $\psi^*$ and $\lambda^*$ are the optimal subsample aspect ratio and ridge penalty, respectively. Furthermore, this equivalence of prediction risk between subsampling and ridge regularization is extended in [13] to anisotropic linear models. As an application, [13] also demonstrates how generalized cross-validation for ridge regression can be naturally transferred to subsample ridge ensembles. In essence, these results suggest that subsampling a smaller number of observations has the same effect as adding an appropriately larger level of ridge penalty. These findings prompt two important open questions:

(a) **The extent of equivalences.** The previous works all focus on equivalences of the squared prediction risk when the test distribution matches the train distribution. In real-world scenarios,

37th Conference on Neural Information Processing Systems (NeurIPS 2023).

Table 1: Comparison with related work. The marker "✓°" indicates a partial equivalence result connecting the *optimal* prediction risk of the ridge predictor and the full ridgeless ensemble predictor.

| | Type of equivalence results | | | Type of data assumptions | | |
|---|---|---|---|---|---|---|
| | pred. risk | gen. risk | estimator | response | feature | lim. spectrum |
| LeJeune et al. [11] | ✓° | | | linear | isotropic Gaussian | exists |
| Patil et al. [12] | ✓° | | | linear | isotropic RMT | exists |
| Du et al. [13] | ✓ | | | linear | anisotropic RMT | exists |
| This work | ✓ | ✓ | ✓ | arbitrary | anisotropic RMT | need not exist |

however, there are often covariate and label shifts, making it crucial to consider prediction risks under such shifts. In addition, other risk measures, such as training error, estimation risk, coefficient errors, and more, are also of interest in various inferential tasks. A natural question is then whether similar equivalences hold for such "generalized" risk measures. At a basic level, the question boils down to whether any equivalences exist at the "estimator level". Answering this question would establish a connection between the ensemble and the ridge estimators, facilitating the exchange of various downstream inferential statements between the two sets of estimators.

(b) **The role of linear model.** All previous works assume a well-specified linear model between the responses and the features, which rarely holds in practical applications. A natural question is whether the equivalences hold for *arbitrary* joint distributions of the response and the features or whether they are merely an artifact of the linear model. Addressing this question broadens the applicability of such equivalences beyond simplistic models to real-world scenarios, where the relationship between the response and the features is typically intricate and unknown.

We provide answers to questions raised in both directions. We demonstrate that the equivalences hold for the generalized squared risks in the full ensemble. Further, these equivalences fundamentally occur at the estimator level for any arbitrary ensemble size. Importantly, these results are not limited to linear models of the data-generating process. Below we provide a summary of our main results.

## 1.1 Summary of contributions

(1) **Risk equivalences.** We establish asymptotic equivalences of the full-ensemble ridge estimators at different ridge penalties $\lambda$ and subsample ratios $\psi$ along specific paths in the $(\lambda, \psi)$-plane for a variety of generalized risk functionals. This class of functionals includes commonly used risk measures (see Table 2), and our results hold for both independent and dependent coefficient matrices that define these risk functionals (see Theorem 1 and Proposition 2). In addition, we demonstrate that the equivalence path remains unchanged across all the functionals examined.

(2) **Structural equivalences.** We establish structural equivalences in the form of linear functionals of the ensemble ridge estimators, which hold for arbitrary ensemble sizes (see Theorem 3). Our proofs for both structural and risk equivalences exploit properties of certain fixed equations that arise in our analysis, enabling us to explicitly characterize paths that yield equivalent estimators in the $(\lambda, \psi)$-plane (see Equation (5)). In addition, we provide an entirely data-driven construction of this path using certain companion Stieltjes transform relations of random matrices (see Proposition 4).

(3) **Equivalence implications.** As an implication of our equivalences, we show that the prediction risk of an optimally-tuned ridge estimator is monotonically increasing in the data aspect ratio under mild regularity conditions (see Theorem 6). This is an indirect consequence of our general equivalence results that leverages the provable monotonicity of the subsample-optimized estimator. Under proportional asymptotics, our result settles a recent open question raised by Nakkiran et al. [1, Conjecture 1] concerning the monotonicity of optimal ridge regression under anisotropic features and general data models while maintaining a regularity condition that preserves the linearized signal-to-noise ratios across regression problems.

(4) **Generality of equivalences.** Our main results apply to arbitrary responses with bounded $4 + \mu$ moments for some $\mu > 0$, as well as features with similar bounded moments and general

covariance structures. We demonstrate the practical implications of our findings on real-world datasets (see Section 6). On the technical side, we extend the tools developed in [14] to handle the dependency between the linear and the non-linear components and obtain model-free equivalences. Furthermore, we extend our analysis to include generalized ridge regression (see Corollary 5). Through experiments, we demonstrate the universality of the equivalences for random and kernel features and conjecture the associated data-driven prediction risk equivalence paths (see Section 6).

## 1.2 Related work

**Ensemble learning.** Ensemble methods yield strong predictors by combining weak predictors [15]. The most common strategy for building ensembles is based on subsampling observations, which includes bagging [2, 3], random forests [16], neural network ensembles [17–19], among the others. The effect of sampling-based ensembles has been studied in the statistical physics literature [20–22] as well as statistical learning literature [2, 3]. Recent works have attempted to explain the success of ensemble learning by suggesting that subsampling induces a regularizing effect [4, 6]. Under proportional asymptotics, the effect of ensemble random feature models has been investigated in [23–25]. There has been growing interest in connecting the effect of ensemble learning to explicit regularization: see [26] for related experimental evidence under the umbrella of "mini-patch" learning. Towards making these connections precise, some work has been done in the context of the ridge and ridgeless ensembles: [11–13] investigate the relationship between subsampling and regularization for ridge ensembles in the overparameterized regime. We will delve into these works in detail next.

**Ensembles and ridge equivalences.** In the study of ridge ensembles, LeJeune et al. [11] show that the optimal ridge predictor has the same prediction risk as the ridgeless ensemble under the Gaussian isotropic design, which has been extended in [12] for RMT features (see Assumption 2 for definition). Specifically, these works establish the prediction risk equivalences for a specific pair of $(\lambda^*, \phi)$ and $(0, \psi^*)$. The results are then extended to the entire range of $(\lambda, \psi)$ using deterministic asymptotic risk formulas [13], assuming a well-specified linear model, RMT features, and the existence of a fixed limiting spectral distribution of the covariance matrix. Our work significantly broadens the scope of results connecting subsampling and ridge regularization. In particular, we allow for arbitrary joint distributions of the data, anisotropic features with only bounded moments, and do not assume the convergence of the spectrum of the population covariance matrix. Furthermore, we expand the applicability of equivalences to encompass both the estimators and various generalized squared risk functionals. See Table 1 for an explicit comparison and Section 3 for additional related comparisons.

**Other ridge connections.** Beyond the connection connections between implicit regularization induced by subsampling and explicit ridge regularization examined in this paper, ridge regression shares ties with various other forms of regularization. For example, ridge regression has been linked to dropout regularization [27, 28], variable splitting in random forests [2], noisy training [29, 30], random sketched regression [31–33], various forms of data augmentation [34], feature augmentation [9, 35], early stopping in gradient descent and its variants [36–39], among others. These connections highlight the pervasiveness and the significance of understanding various "facets" of ridge regularization. In that direction, our work contributes to expand the equivalences between subsampling and ridge regularization.

## 2 Notation and preliminaries

**Ensemble estimators.** Let $\mathcal{D}_n = \{(\boldsymbol{x}_i, y_i) : i \in [n]\}$ be a dataset containing i.i.d. samples in $\mathbb{R}^p \times \mathbb{R}$. Let $\boldsymbol{X} \in \mathbb{R}^{n \times p}$ and $\boldsymbol{y} \in \mathbb{R}^n$ be the feature matrix and response vector with $\boldsymbol{x}_i^\top$ and $y_i$ in $i$-th rows. For an index set $I \subseteq [n]$ of size $k$, let $\mathcal{D}_I = \{(\boldsymbol{x}_i, y_i) : i \in I\}$ be the associated subsampled dataset. Let $\boldsymbol{L}_I \in \mathbb{R}^{n \times n}$ be a diagonal matrix such that its $i$-th diagonal entry is 1 if $i \in I$ and 0 otherwise. Note that the feature matrix and response vector associated with $\mathcal{D}_I$ respectively are $\boldsymbol{L}_I \boldsymbol{X}$ and $\boldsymbol{L}_I \boldsymbol{y}$. Given a ridge penalty $\lambda > 0$, the *ridge* estimator fitted on $\mathcal{D}_I$ consisting of $k$ samples is given by:

$$\widehat{\boldsymbol{\beta}}_k^\lambda(\mathcal{D}_I) = \underset{\boldsymbol{\beta} \in \mathbb{R}^p}{\mathrm{argmin}} \frac{1}{k} \sum_{i \in I} (y_i - \boldsymbol{x}_i^\top \boldsymbol{\beta})^2 + \lambda \|\boldsymbol{\beta}\|_2^2 = \left( \frac{1}{k} \boldsymbol{X}^\top \boldsymbol{L}_I \boldsymbol{X} + \lambda \boldsymbol{I}_p \right)^{-1} \frac{\boldsymbol{X}^\top \boldsymbol{L}_I \boldsymbol{y}}{k}. \quad (1)$$

The *ridgeless* estimator $\widehat{\boldsymbol{\beta}}_k^0(\mathcal{D}_I) = (\boldsymbol{X}^\top \boldsymbol{L}_I \boldsymbol{X}/k)^+ \boldsymbol{X}^\top \boldsymbol{L}_I \boldsymbol{y}/k$ is obtained by sending $\lambda \to 0^+$, where $\boldsymbol{M}^+$ denotes the Moore-Penrose inverse of matrix $\boldsymbol{M}$. To define the ensemble estimator, denote

Table 2: Summary of various generalized risks and their corresponding statistical learning problems. The asymptotic equivalences between subsampling and ridge regularization are analyzed in Theorem 1 and Proposition 2 for generalized risks (3) defined through functionals $L_{\boldsymbol{A},\boldsymbol{b}}$ for various $\boldsymbol{A}$ and $\boldsymbol{b}$.

| Statistical learning problem | $L_{\boldsymbol{A},\boldsymbol{b}}(\widehat{\boldsymbol{\beta}} - \boldsymbol{\beta}_0)$ | $\boldsymbol{A}$ | $\boldsymbol{b}$ | $\mathrm{nrow}(\boldsymbol{A})$ |
|---|---|---|---|---|
| vector coefficient estimation | $\widehat{\boldsymbol{\beta}} - \boldsymbol{\beta}_0$ | $\boldsymbol{I}_p$ | $0$ | $p$ |
| projected coefficient estimation | $\boldsymbol{a}^\top(\widehat{\boldsymbol{\beta}} - \boldsymbol{\beta}_0)$ | $\boldsymbol{a}^\top$ | $0$ | $1$ |
| training error estimation | $\boldsymbol{X}\widehat{\boldsymbol{\beta}} - \boldsymbol{y}$ | $\boldsymbol{X}$ | $-\boldsymbol{f}_{\mathrm{NL}}$ | $n$ |
| in-sample prediction | $\boldsymbol{X}(\widehat{\boldsymbol{\beta}} - \boldsymbol{\beta}_0)$ | $\boldsymbol{X}$ | $0$ | $n$ |
| out-of-sample prediction | $\boldsymbol{x}_0^\top\widehat{\boldsymbol{\beta}} - y_0$ | $\boldsymbol{x}_0^\top$ | $-\varepsilon_0$ | $1$ |

the set of all $k$ distinct elements from $[n]$ by $\mathcal{I}_k = \{\{i_1, i_2, \ldots, i_k\} : 1 \le i_1 < i_2 < \ldots < i_k \le n\}$. For $\lambda \ge 0$, the $M$-*ensemble* and the *full-ensemble* estimators are respectively defined as follows:

$$\widehat{\boldsymbol{\beta}}_{k,M}^\lambda(\mathcal{D}_n; \{I_\ell\}_{\ell=1}^M) = \frac{1}{M}\sum_{\ell \in [M]}\widehat{\boldsymbol{\beta}}_k^\lambda(\mathcal{D}_{I_\ell}), \quad \text{and} \quad \widehat{\boldsymbol{\beta}}_{k,\infty}^\lambda(\mathcal{D}_n) = \mathbb{E}[\widehat{\boldsymbol{\beta}}_k^\lambda(\mathcal{D}_I) \mid \mathcal{D}_n]. \tag{2}$$

Here $\{I_\ell\}_{\ell=1}^M$ and $I$ are simple random samples from $\mathcal{I}_k$. In (2), the full-ensemble ridge estimator $\widehat{\boldsymbol{\beta}}_{k,\infty}^\lambda(\mathcal{D}_n)$ is the average of predictors fitted on all possible subsampled datasets. It is shown in Lemma A.1 of [13] that $\widehat{\boldsymbol{\beta}}_{k,\infty}^\lambda(\mathcal{D}_n)$ is also asymptotically equivalent to the limit of $\widehat{\boldsymbol{\beta}}_{k,M}^\lambda(\mathcal{D}_n; \{I_\ell\}_{\ell=1}^M)$ as the ensemble size $M \to \infty$ (conditioning on the full dataset $\mathcal{D}_n$). This also justifies using $\infty$ in the subscript to denote the full-ensemble estimator. For brevity, we simply write the estimators as $\widehat{\boldsymbol{\beta}}_{k,M}^\lambda$, $\widehat{\boldsymbol{\beta}}_{k,\infty}^\lambda$, and drop the dependency on $\mathcal{D}_n$, $\{I_\ell\}_{\ell=1}^M$. We will show certain structural equivalences in terms of linear projections in the family of estimators $\widehat{\boldsymbol{\beta}}_{k,M}^\lambda$ along certain paths in the $(\lambda, p/k)$-plane for arbitrary ensemble size $M \in \mathbb{N} \cup \{\infty\}$ (in Theorem 3). Apart from connecting the estimators, we will also show equivalences of various notions of risks (in Theorem 1), which we describe next.

**Generalized risks.** Since the ensemble ridge estimators are linear estimators, we evaluate their performance relative to the oracle parameter: $\boldsymbol{\beta}_0 = \mathbb{E}[\boldsymbol{x}\boldsymbol{x}^\top]^{-1}\mathbb{E}[\boldsymbol{x}y]$, which is the best (population) linear projection of $y$ onto $\boldsymbol{x}$ and minimizes the linear regression error (see, e.g., [40–42]). Note that we can decompose any response $y$ into: $y = f_{\mathrm{LI}}(\boldsymbol{x}) + f_{\mathrm{NL}}(\boldsymbol{x})$, where $f_{\mathrm{LI}}(\boldsymbol{x}) = \boldsymbol{\beta}_0^\top\boldsymbol{x}$ is the oracle linear predictor, and $f_{\mathrm{NL}}(\boldsymbol{x}) = y - f_{\mathrm{LI}}(\boldsymbol{x})$ is the nonlinear component that is not explained by $f_{\mathrm{LI}}(\boldsymbol{x})$. The best linear projection has the useful property that $f_{\mathrm{LI}}(\boldsymbol{x})$ is (linearly) uncorrelated with $f_{\mathrm{NL}}(\boldsymbol{x})$, although they are generally dependent. It is worth mentioning that this does not imply that $y$ and $\boldsymbol{x}$ follow a linear regression model. Indeed, our framework allows any nonlinear dependence structure between them and is model-free for the joint distribution of $(\boldsymbol{x}, y)$. Relative to $\boldsymbol{\beta}_0$, we measure the performance of an estimator $\widehat{\boldsymbol{\beta}}$ via the generalized mean squared risk defined compactly as follows:

$$R(\widehat{\boldsymbol{\beta}}; \boldsymbol{A}, \boldsymbol{b}, \boldsymbol{\beta}_0) = \frac{1}{\mathrm{nrow}(\boldsymbol{A})}\|L_{\boldsymbol{A},\boldsymbol{b}}(\widehat{\boldsymbol{\beta}} - \boldsymbol{\beta}_0)\|_2^2, \tag{3}$$

where $L_{\boldsymbol{A},\boldsymbol{b}}(\boldsymbol{\beta}) = \boldsymbol{A}\boldsymbol{\beta} + \boldsymbol{b}$ is a linear functional. Note that $\boldsymbol{A}$ and $\boldsymbol{b}$ can potentially depend on the data, and their dimensions can vary depending on the statistical learning problem at hand. This framework includes various important statistical learning problems, as summarized in Table 2. Observe that the framework also includes various notions of prediction risks. One can use the test error formulation in Table 2 to obtain the prediction risk at any test point $(\boldsymbol{x}_0, y_0)$, where $y_0 = \boldsymbol{x}_0^\top\boldsymbol{\beta}_0 + \varepsilon_0$, and $\varepsilon_0$ may depend on $\boldsymbol{x}_0$ and have non-zero mean. This also permits the test point to be drawn from a distribution that differs from the training distribution. Specifically, when $\varepsilon_0 = f_{\mathrm{NL}}(\boldsymbol{x}_0)$ but $\boldsymbol{x}_0$ has a distribution different from $\boldsymbol{x}$, we obtain the prediction error under *covariate shift*. Similarly, when $\varepsilon_0 \ne f_{\mathrm{NL}}(\boldsymbol{x}_0)$ but $\boldsymbol{x}_0$ and $\boldsymbol{x}$ have the same distribution, we get the prediction error under *label shift*.

**Asymptotic equivalence.** For our theoretical analysis, we consider the proportional asymptotics regime, where the ratio of the feature size $p$ to the sample size $n$ tends to a fixed limiting *data aspect ratio* $\phi \in (0, \infty)$. To concisely present our results, we will use the elegant framework of asymptotic equivalence [12, 43, 44]. Let $\boldsymbol{A}_p$ and $\boldsymbol{B}_p$ be sequences of (additively) conformable matrices of arbitrary dimensions (including vectors and scalars). We say that $\boldsymbol{A}_p$ and $\boldsymbol{B}_p$ are *asymptotically equivalent*, denoted as $\boldsymbol{A}_p \simeq \boldsymbol{B}_p$, if $\lim_{p \to \infty} |\mathrm{tr}[\boldsymbol{C}_p(\boldsymbol{A}_p - \boldsymbol{B}_p)]| = 0$ almost surely for any sequence of random matrices $\boldsymbol{C}_p$ with bounded trace norm that are (multiplicatively) conformable and independent of $\boldsymbol{A}_p$ and $\boldsymbol{B}_p$. Note that for sequences of scalar random variables, the definition simply reduces to the typical almost sure convergence of sequences of random variables involved.

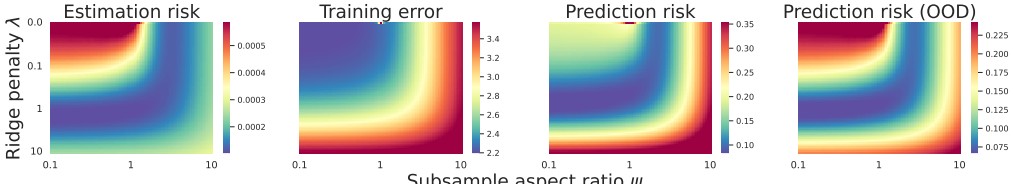

Figure 1: Heat map of the various generalized risks (estimation risk, training error, prediction risk, out-of-distribution (OOD) prediction risk) of full-ensemble ridge estimators (approximated with $M = 100$), for varying ridge penalties $\lambda$ and subsample aspect ratios $\psi = p/k$ on the log-log scale. The data model is described in Appendix F.2 with $p = 500$, $n = 5000$, and $\phi = p/n = 0.1$. Observe that along the same path the values match well for all risks, in line with Theorem 1 and Proposition 2.

## 3 Generalized risk equivalences of ensemble estimators

We begin by examining the risk equivalences among different ensemble ridge estimators for various generalized risks defined as in (3). To prepare for our upcoming results, we impose two structural and moment assumptions on the distributions of the response variable and the feature vector.

**Assumption 1** (Response variable distribution). Each response variable $y_i$ for $i \in [n]$ has mean 0 and satisfies $\mathbb{E}[|y_i|^{4+\mu}] \leq M_\mu < \infty$ for some $\mu > 0$ and a constant $M_\mu$.

**Assumption 2** (Feature vector distribution). Each feature vector $\boldsymbol{x}_i$ for $i \in [n]$ can be decomposed as $\boldsymbol{x}_i = \boldsymbol{\Sigma}^{1/2} \boldsymbol{z}_i$, where $\boldsymbol{z}_i \in \mathbb{R}^p$ contains i.i.d. entries $z_{ij}$ for $j \in [p]$ with mean 0, variance 1, and satisfy $\mathbb{E}[|z_{ij}|^{4+\nu}] \leq M_\nu < \infty$ for some $\nu > 0$ and a constant $M_\nu$, and $\boldsymbol{\Sigma} \in \mathbb{R}^{p \times p}$ is a deterministic and symmetric matrix with eigenvalues uniformly bounded between constants $r_{\min} > 0$ and $r_{\max} < \infty$.

Note that we do not impose any specific model assumptions on the response variable $y$ in relation to the feature vector $\boldsymbol{x}$. We only assume bounded moments as stated in Assumption 1, making all of our subsequent results model-free. The zero-mean assumption for $y$ is for simplicity since, in practice, centering can always be done by subtracting the sample mean. The bounded moment condition can also be satisfied if one imposes a stronger distributional assumption (e.g., sub-Gaussianity). Assumption 2 on the feature vector is common in the study of random matrix theory [45, 46] and the analysis of ridge and ridgeless regression [10, 47–49], which we refer to as *RMT features* for brevity.

Given a limiting data aspect ratio $\phi \in (0, \infty)$ and a limiting *subsample aspect ratio* $\overline{\psi} \in [\phi, \infty]$, our statement of equivalences between different ensemble estimators is defined through certain paths characterized by two endpoints $(0, \overline{\psi})$ and $(\overline{\lambda}, \phi)$. These endpoints correspond to the subsample ridgeless ensemble and the (non-ensemble) ridge predictor, respectively, with $\overline{\lambda}$ being the ridge penalty to be defined next. For that, let $H_p$ be the empirical spectral distribution of $\boldsymbol{\Sigma}$: $H_p(r) = p^{-1} \sum_{i=1}^{p} \mathbb{1}_{\{r_i \leq r\}}$, where $r_i$'s are the eigenvalues of $\boldsymbol{\Sigma}$. Consider the following system of equations in $\overline{\lambda}$ and $v$:

$$\frac{1}{v} = \overline{\lambda} + \phi \int \frac{r}{1 + vr}\, \mathrm{d}H_p(r), \quad \text{and} \quad \frac{1}{v} = \overline{\psi} \int \frac{r}{1 + vr}\, \mathrm{d}H_p(r). \tag{4}$$

Existence and uniqueness of the solution $(\overline{\lambda}, v) \in [0, \infty]^2$ to the above equations are guaranteed by Corollary E.4. Now, define a path $\mathcal{P}(\overline{\lambda}; \phi, \overline{\psi})$ that passes through the endpoints $(0, \overline{\psi})$ and $(\overline{\lambda}, \phi)$:

$$\mathcal{P}(\overline{\lambda}; \phi, \overline{\psi}) = \left\{ (1 - \theta) \cdot (\overline{\lambda}, \phi) + \theta \cdot (0, \overline{\psi}) \mid \theta \in [0, 1] \right\}. \tag{5}$$

We are ready to state our results. We consider two cases for the generalized risk (3) depending on the relationships between $(\boldsymbol{A}, \boldsymbol{b})$ and $\boldsymbol{X}$. In the first case, when both $\boldsymbol{A}$ and $\boldsymbol{b}$ are independent of the data, Theorem 1 shows that the generalized risks are equivalent along the path (5).

**Theorem 1** (Risk equivalences when $(\boldsymbol{A}, \boldsymbol{b}) \perp\!\!\!\perp (\boldsymbol{X}, \boldsymbol{y})$). *Suppose Assumptions 1–2 hold. Let $(\boldsymbol{A}, \boldsymbol{b})$ be independent of $(\boldsymbol{X}, \boldsymbol{y})$ such that $\mathrm{nrow}(\boldsymbol{A})^{-1/2} \|\boldsymbol{A}\|_{\mathrm{op}}$ and $\|\boldsymbol{b}\|_2$ are almost surely bounded. Let $n, p \to \infty$ such that $p/n \to \phi \in (0, \infty)$. For any $\overline{\psi} \in [\phi, +\infty]$, let $\overline{\lambda}$ be as defined in (4). Then, for any pair of $(\lambda_1, \psi_1)$ and $(\lambda_2, \psi_2)$ on the path $\mathcal{P}(\overline{\lambda}; \phi, \overline{\psi})$ as defined in (5), the generalized risk functionals (3) of the full-ensemble estimator are asymptotically equivalent:*

$$R\big(\widehat{\boldsymbol{\beta}}^{\lambda_1}_{\lfloor p/\psi_1 \rfloor, \infty}; \boldsymbol{A}, \boldsymbol{b}, \boldsymbol{\beta}_0\big) \simeq R\big(\widehat{\boldsymbol{\beta}}^{\lambda_2}_{\lfloor p/\psi_2 \rfloor, \infty}; \boldsymbol{A}, \boldsymbol{b}, \boldsymbol{\beta}_0\big). \tag{6}$$

In other words, Theorem 1 establishes the equivalences of subsampling and ridge regularization in terms of generalized risk for many statistical learning problems, encompassing coefficient estimation,

coefficient confidence interval, and test error estimation. All of these problems fall in the category when $(\boldsymbol{A}, \boldsymbol{b})$ are independent of $\boldsymbol{X}$. However, there are other statistical learning problems not covered in Theorem 1, such as the in-sample prediction risk and the training error, which correspond to the case when $\boldsymbol{A} = \boldsymbol{X}$. Fortunately, the equivalences also apply to them, as summarized in Proposition 2.

**Proposition 2** (Risk equivalences when $\boldsymbol{A} = \boldsymbol{X}$). *Under Assumptions 1–2, when $\boldsymbol{A} = \boldsymbol{X}$, the conclusion in Theorem 1 continue to hold for the cases of in-sample prediction risk ($\boldsymbol{b} = \boldsymbol{0}$) and training error ($\boldsymbol{b} = -\boldsymbol{f}_{\mathrm{NL}}$).*

Both Theorem 1 and Proposition 2 provide specific second-order equivalences for the full-ensemble ridge estimators, in the sense that the quadratic functionals of the estimators associated with $(\boldsymbol{A}, \boldsymbol{b})$ are asymptotically the same. See Figure 1 for visual illustrations of both equivalences. These statements are presented separately because their proofs differ, with the proof for the dependent case being more intricate. We note that by combining the risk functionals in Table 2, it is possible to further extend the equivalences to other types of functionals not directly covered by statements above, using composition and continuous mapping. For example, it is not difficult to show that similar equivalences hold for the generalized cross-validation in the full ensembles [13]. This follows by combining the result for training error from Proposition 2 and for denominator concentration proved in Lemma 3.4 of [13].

Theorem 1 generalizes the existing equivalence results for the prediction risk [11–13] to include general risk functionals under milder assumptions. As summarized in Table 1, we allow for a general response model and feature covariance without assuming convergence of the spectral distributions $H_p$ to a fixed distribution $H$. This flexibility is achieved by showing equivalences of the underlying resolvents in the estimators rather than deriving the limiting functionals as done in previous works. To extend results allowing for a general response, we generalize certain concentration results by leveraging tools from [14] to handle the dependency between the linear and non-linear components.

As alluded to earlier, it is worth noting that the generalized risk equivalences only hold in the full ensemble. However, by using the risk decomposition obtained from Lemma S.1.1 of [12] for finite ensembles, one can obtain the following relationship along the path in Theorem 3: $R\big(\widehat{\boldsymbol{\beta}}^{\lambda_1}_{\lfloor p/\psi_1 \rfloor, M}; \boldsymbol{A}, \boldsymbol{b}, \boldsymbol{\beta}_0\big) - R\big(\widehat{\boldsymbol{\beta}}^{\lambda_2}_{\lfloor p/\psi_2 \rfloor, M}; \boldsymbol{A}, \boldsymbol{b}, \boldsymbol{\beta}_0\big) \simeq \Delta/M$, for some $\Delta$ (independent of $M$) that is eventually almost surely bounded. In other words, the difference between any two estimators on the same path scales as $1/M$ when $n$ is sufficiently large, as numerically verified in Figure F5.

## 4 Structural equivalences of ensemble estimators

The equivalences established in the previous section only hold for the full-ensemble estimators in a second-order sense. We can go a step further and ask if there exist any equivalences for the finite ensemble and if there are any equivalences at the estimator coordinate level between the estimators. Specifically, we aim to inspect whether each coordinate of the $p$-dimensional estimated coefficients asymptotically equals. This section establishes structural relationships between the estimators in a first-order sense that any bounded linear functionals of them are asymptotically the same.

**Theorem 3** (Structural equivalences). *Suppose Assumptions 1–2 hold. Let $n, p \to \infty$ with $p/n \to \phi \in (0, \infty)$. For any $\overline{\psi} \in [\phi, +\infty]$, let $\overline{\lambda}$ be as in (4). Then, for any $M \in \mathbb{N} \cup \{\infty\}$ and any pair of $(\lambda_1, \psi_1)$ and $(\lambda_2, \psi_2)$ on the path (5), the $M$-ensemble estimators are asymptotically equivalent:*

$$\widehat{\boldsymbol{\beta}}^{\lambda_1}_{\lfloor p/\psi_1 \rfloor, M} \simeq \widehat{\boldsymbol{\beta}}^{\lambda_2}_{\lfloor p/\psi_2 \rfloor, M}. \tag{7}$$

Put another way, Theorem 3 implies that for any fixed ensemble size $M$, the $M$-ensemble ridgeless estimator at the limiting aspect ratio $\overline{\psi}$ is equivalent to ridge regression at the regularization level $\overline{\lambda}$. This equivalence is visually illustrated in Figure 2, where the data is simulated from an anisotropic and nonlinear model (see Appendix F.2 for more details). Furthermore, the contour path $\mathcal{P}(\overline{\lambda}; \phi, \overline{\psi})$ connecting the endpoints $(0, \overline{\psi})$ and $(\overline{\lambda}, \phi)$ is a straight line, although it may have varying slopes. Along any path, all $M$-ensemble estimators at the limiting aspect ratio $\psi$ and ridge penalty $\lambda$ are equivalent for all $(\lambda, \psi) \in \mathcal{P}(\overline{\lambda}; \phi, \overline{\psi})$. The equivalences here are for any arbitrary linear combinations of the estimators. In particular, this implies that the predicted values (or even any continuous function applied to them due to the continuous mapping theorem) of any test point will eventually be the same, almost surely, with respect to the training data. Finally, we remark that in the statement of Theorem 3, the equivalence is defined on extended reals in order to incorporate the ridgeless case when $\overline{\psi} = 1$.

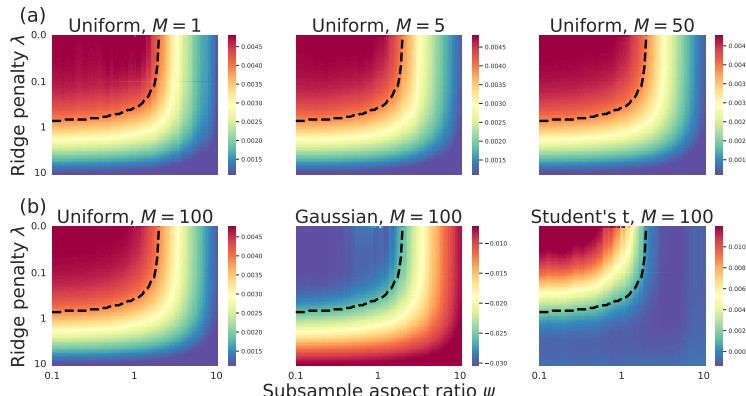

Figure 2: Heat map of the values of various linear functionals of ensemble ridge estimators $\boldsymbol{a}^\top \widehat{\boldsymbol{\beta}}_{k,M}^\lambda$, for varying ridge penalties $\lambda$, subsample aspect ratios $\psi = p/k$, and ensemble size $M$ on the log-log scale. The data model is described in Appendix F.2 with $p = 1000$, $n = 10000$, and $\phi = p/n = 0.1$. (a) The values of the uniformly weighted projection for varying ensemble sizes ($M = 1, 5$, and $50$). (b) The values of $M = 100$ for varying projection vectors (uniformly weighted, random Gaussian, and random student's $t$). The black dashed line is estimated based on Proposition 4 with $\overline{\psi} = 2$. Observe that the values along the data-dependent paths are indeed very similar, in line with Proposition 4.

The paths (5) in Theorem 3 are defined via the spectral distribution $H_p$, which requires knowledge of $\boldsymbol{\Sigma}$. This is often difficult to obtain in practice from the observed data. Fortunately, we can provide an alternative characterization for the path (5) solely through the data, as summarized in Proposition 4.

**Proposition 4** (Data-dependent equivalence path characterization). *Suppose Assumptions 1–2 hold. Define $\phi_n = p/n$. Let $k \le n$ be the subsample size and denote by $\overline{\psi}_n = p/k$. For any $M \in \mathbb{N} \cup \{\infty\}$, let $\overline{\lambda}_n$ be the value that satisfies the following equation in ensemble ridgeless and ridge gram matrices:*

$$\frac{1}{M} \sum_{\ell=1}^{M} \frac{1}{k} \operatorname{tr}\left[ \left( \frac{1}{k} \boldsymbol{L}_{I_\ell} \boldsymbol{X} \boldsymbol{X}^\top \boldsymbol{L}_{I_\ell} \right)^+ \right] = \frac{1}{n} \operatorname{tr}\left[ \left( \frac{1}{n} \boldsymbol{X} \boldsymbol{X}^\top + \overline{\lambda}_n \boldsymbol{I}_n \right)^{-1} \right]. \tag{8}$$

*Define the data-dependent path $\mathcal{P}_n = \mathcal{P}(\overline{\lambda}_n; \phi_n, \overline{\psi}_n)$. Then, the conclusion in Theorem 3 continues to hold if we replace the (population) path $\mathcal{P}$ by the (data-dependent) path $\mathcal{P}_n$.*

The term "data-dependent path" signifies that we can estimate the level of implicit regularization induced by subsampling solely based on the observed data by solving (8). We remark that (8) always has at least one solution for a given triplet of $(n, k, p)$. This is because the right-hand side of (8) is monotonically decreasing in $\lambda$, and the left-hand side always lies within the range of the right-hand side. The powerful implication of the characterization in Proposition 4 is that it enables practical computation of the path using real-world data. In Section 6, we will demonstrate this approach on various real-world datasets, allowing us to predict an equivalent amount of explicit regularization matching the implicit regularization due to subsampling.

One natural extension of the results is to consider the generalized ridge as a base estimator (1):

$$\widehat{\boldsymbol{\beta}}_k^{\lambda, \boldsymbol{G}}(\mathcal{D}_I) = \operatorname*{argmin}_{\boldsymbol{\beta} \in \mathbb{R}^p} \frac{1}{k} \sum_{i \in I} (y_i - \boldsymbol{x}_i^\top \boldsymbol{\beta})^2 + \lambda \| \boldsymbol{G}^{\frac{1}{2}} \boldsymbol{\beta} \|_2^2, \tag{9}$$

where $\boldsymbol{G} \in \mathbb{R}^{p \times p}$ is a positive definite matrix. The equivalences still hold, albeit on different paths:

**Corollary 5** (Equivalences for generalized ridge regression). *Suppose Assumptions 1–2 hold. Let $\boldsymbol{G} \in \mathbb{R}^{p \times p}$ be a deterministic and symmetric matrix with eigenvalues uniformly bounded between constants $g_{\min} > 0$ and $g_{\max} < \infty$. Let $n, p \to \infty$ such that $p/n \to \phi \in (0, \infty)$. For any $\overline{\psi} \in [\phi, +\infty]$, let $\overline{\lambda}$ be as defined in (4) with $H_p$ replaced $\widetilde{H}_p$, the empirical spectral distribution of $\boldsymbol{G}^{-1/2} \boldsymbol{\Sigma} \boldsymbol{G}^{-1/2}$. Then, the conclusions in Theorems 1 and 3 continue to hold for the generalized ridge ensemble predictors.*

Another extension of our results is to incorporate subquadratic risk functionals in Theorem 1 to derive equivalences of the cumulative distribution functions of the predictive error distributions associated

with the estimators along the equivalence paths. One can use such equivalences for downstream inference questions. Finally, while our statements are asymptotic in nature, we expect a similar analysis as done in [50] can yield finite-sample statements. We leave these extensions for the future.

# 5 Implications of equivalences

The generalized risk equivalences establish a connection between the risks of different ridge ensembles, including the ridge predictors on full data and the full-ensemble ridgeless predictors. These connections enable us to transfer properties from one estimator to the other. For example, by examining the impact of subsampling on the estimator's performance, we can better understand how to optimize the ridge penalty. We will next provide an example of this. Many common methods, such as ridgeless or lassoless predictors, have been recently shown to exhibit non-monotonic behavior in the sample size or the limiting aspect ratio. An open problem raised by Nakkiran et al. [1, Conjecture 1] asks whether the prediction risk of the ridge regression with optimal ridge penalty $\lambda^*$ is monotonically increasing in the data aspect ratio $\phi = p/n$. The non-monotonicity risk behavior implies that more data can hurt performance, and it's important to investigate whether optimal ridge regression also suffers from this issue. Our equivalence results provide a surprising (even to us) indirect way to answer this question affirmatively under proportional asymptotics.

To analyze the monotonicity of the risk, we will need to impose two additional regularity assumptions to guarantee the convergence of the prediction risk. Let $\boldsymbol{\Sigma} = \sum_{j=1}^{p} r_j \boldsymbol{w}_j \boldsymbol{w}_j^\top$ denote the eigenvalue decomposition, where $(r_j, \boldsymbol{w}_j)$'s are pairs of associated eigenvalue and normalized eigenvector. The following assumptions ensure that the hardness of the underlying regression problems across different limiting data aspect ratios is comparable.

**Assumption 3** (Spectral convergence). We assume there exists a deterministic distribution $H$ such that the empirical spectral distribution of $\boldsymbol{\Sigma}$, $H_p(r) = p^{-1} \sum_{i=1}^{p} \mathbb{1}_{\{r_i \leq r\}}$, weakly converges to $H$, almost surely (with respect to $\boldsymbol{x}$).

**Assumption 4** (Limiting signal and noise energies). We assume there exists a deterministic distribution $G$ such that the empirical distribution of $\boldsymbol{\beta}_0$'s (squared) projection onto $\boldsymbol{\Sigma}$'s eigenspace, $G_p(r) = \|\boldsymbol{\beta}_0\|_2^{-2} \sum_{i=1}^{p} (\boldsymbol{\beta}_0^\top \boldsymbol{w}_i)^2 \mathbb{1}_{\{r_j \leq r\}}$, weakly converges to $G$. As $n, p \to \infty$ and $p/n \to \phi \in (0, \infty)$, the limiting linearized energy $\rho^2 = \lim \|\boldsymbol{\beta}_0\|_2^2$ and the limiting nonlinearized energy $\sigma^2 = \lim \|f_{\mathrm{NL}}\|_{L_2}^2$ are finite.

Assumption 3 is commonly used in random matrix theory and overparameterized learning [10, 12, 13] and ensures that the limiting spectral distribution of the sample covariance matrix converges to a fixed distribution. Under these assumptions, we show in Appendix C that there exists a deterministic function $\mathscr{R}(\lambda; \phi, \psi)$, such that $R(\widehat{\boldsymbol{\beta}}_{k,\infty}^\lambda; \boldsymbol{A}, \boldsymbol{b}, \boldsymbol{\beta}_0) \simeq \mathscr{R}(\lambda; \phi, \psi)$. Notably, the deterministic profile on the right-hand side is a monotonically increasing and a continuous function in the first parameter $\phi$ when limiting values of $\|\boldsymbol{\beta}_0\|_2^2$ and $\|f_{\mathrm{NL}}\|_{L_2}^2$ are fixed (i.e., when Assumption 4 holds). Moreover, the deterministic risk equivalent of the optimal ridge predictor matches that of the optimal full-ensemble ridgeless predictor; in other words, $\min_{\lambda \geq 0} \mathscr{R}(\lambda; \phi, \phi) = \min_{\psi \geq \phi} \mathscr{R}(0; \phi, \psi)$. The same monotonicity property of the two optimized functions leads to the following result.

**Theorem 6** (Monotonicity of prediction risk with optimal ridge regularization). *Suppose the conditions of Theorem 1 and Assumptions 3–4 hold. Let $k, n, p \to \infty$ such that $p/n \to \phi \in (0, \infty)$ and $p/k \to \psi \in [\phi, \infty]$. Then, for $\boldsymbol{A} = \boldsymbol{\Sigma}^{1/2}$ and $\boldsymbol{b} = \boldsymbol{0}$, the optimal risk of the ridgeless ensemble, $\min_{\psi \geq \phi} \mathscr{R}(0; \phi, \psi)$, is monotonically increasing in $\phi$. Consequently, the optimal risk of the ridge predictor, $\min_{\lambda \geq 0} \mathscr{R}(\lambda; \phi, \phi)$, is also monotonically increasing in $\phi$.*

Under Assumptions 3 and 4, the linearized signal-to-noise ratio (SNR) is maintained at the same level across varying distributions as $\phi$ changes. The key message of Theorem 6 is then that, for a sequence of problems with the same SNR (indicating the same level of regression hardness), the asymptotic prediction risk of optimized ridge risk gets monotonically worse as the aspect ratio of the problem increases. This is intuitive because a smaller $\phi$ corresponds to more samples than features. In this sense, Theorem 6 certifies that optimal ridge uses the available data effectively, avoiding sample-wise non-monotonicity. Moreover, as the null risk is finite, this also shows that the optimal ridge estimator mitigates the "double or multiple descents" behaviors in the generalization error [1, 5, 51].

Attempting to prove Theorem 6 directly is challenging due to the lack of a closed-form expression for the optimal risk of ridge regression in general. Moreover, the risk profile of ridge regression,

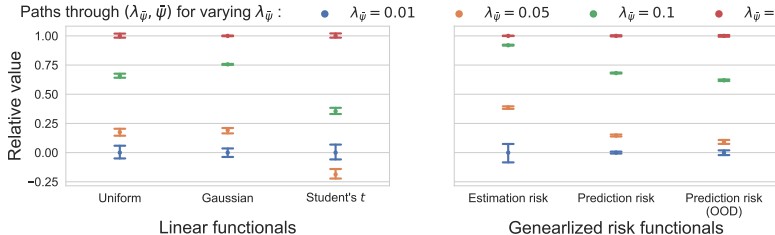

Figure 3: Comparison of various linear functionals (Theorem 3) and generalized risk functionals (Theorem 1) on different paths evaluated on CIFAR-10. The aspect ratios $\phi = p/n$ and $\overline{\psi} = 4\phi$ are estimated from the dataset and fixed. For different values of $\lambda_{\overline{\psi}}$ (different colors), we estimate $\overline{\lambda}$ from Proposition 4, which gives a path between $(\lambda_{\overline{\psi}}, \overline{\psi})$ and $(\overline{\lambda}, \phi)$. For each path, we uniformly sample 5 points and compute the functionals of the ridge ensemble using $M = 100$. The values of different paths are then normalized by subtracting the mean value in the first path and dividing by the mean difference of values on the last and first paths.

$\mathcal{R}(\lambda; \phi, \phi)$, does not exhibit any particular structure as a function of $\lambda$. On the other hand, the risk profile of the full-ensemble ridgeless, $\mathcal{R}(0; \phi, \psi)$, has a very nice structure in terms of $\psi$. This, coupled with our equivalence result, allows one to prove quite non-trivial behavior of optimal ridge. See Figure F6 for illustrations.

## 6  Discussion

Motivated by the recent subsampling and ridge equivalences for the prediction risk [11–13], this paper establishes generalized risk equivalences (Section 3) and structural equivalences (Section 4) within the family of ensemble ridge estimators. Our results precisely link the implicit regularization of subsampling to explicit ridge penalization via a path $\mathcal{P}$ defined in (5), which connects two endpoints $(0, \overline{\psi})$ and $(\overline{\lambda}, \phi)$ regardless of the risk functionals. Furthermore, we provide a data-dependent method (Proposition 4) to estimate this path. Our results do not assume any specific relationship between the response variable $y$ and the feature vector $\boldsymbol{x}$. We next explore some extensions of these equivalences.

**Real data.** While our theoretical results assume RMT features (Assumption 2), we anticipate that the equivalences will very likely hold under more general features [50]. To verify this, we examine the equivalences in Proposition 4 with $M = 100$ on real-world datasets. Figure 3 depicts both the linear functionals in Theorem 3 and the generalized quadratic functionals in Theorem 1 computed along four paths, shown in different colors. We observe that the variation within each path is small, while different paths have different functional values. This suggests that the theoretical finding in the previous sections also hold quite well on real-world datasets. We investigate this further by varying $\overline{\psi}$ on the three image datasets, CIFAR-10, MNIST, and USPS. Figure F7 shows similar values and trends at the two points. These experiments demonstrate that the amount of explicit regularization induced by subsampling can indeed be accurately predicted based on the observed data using Proposition 4.

**Random features.** Closely related to two-layer neural networks [52], we can consider the random feature model, $f(\boldsymbol{x}; \boldsymbol{\beta}, \boldsymbol{F}) = \boldsymbol{\beta}^\top \varphi(\boldsymbol{F}\boldsymbol{x})$, where $\boldsymbol{F} \in \mathbb{R}^{d \times p}$ is some randomly initialized weight matrix, and $\varphi : \mathbb{R} \to \mathbb{R}$ is a nonlinear activation function applied element-wise to $\boldsymbol{F}\boldsymbol{x}$. As from "universality/invariance", certain random feature models are asymptotically equivalent to a surrogate linear Gaussian model with a matching covariance matrix [53], we expect the theoretical results in Theorems 3–1 to likely hold, although the relationship (4) will now depend on the non-linear activation functions. Empirically, we indeed observe similar equivalence phenomena of the prediction risks with random features ridge regression using sigmoid, ReLU, and tanh activation functions, as shown in Figure 4. Analogous to Proposition 4, we conjecture a similar data-driven relationship between $(\overline{\lambda}_n, \phi_n)$ and $(0, \overline{\psi}_n)$ for random features $\overline{\boldsymbol{X}} = \varphi(\boldsymbol{X}\boldsymbol{F}^\top)$:

**Conjecture 7** (Data-dependent equivalences for random features (informal)). *Suppose Assumptions 1–2 hold. Define $\phi_n = p/n$. Let $k \leq n$ be the subsample size and denote by $\overline{\psi}_n = p/k$. Suppose $\varphi$ satisfies certain regularity conditions. For any $M \in \mathbb{N} \cup \{\infty\}$, let $\overline{\lambda}_n$ be the value that satisfies*

$$\frac{1}{M}\sum_{\ell=1}^{M}\frac{1}{k}\operatorname{tr}\left[\left(\frac{1}{k}\varphi(\boldsymbol{L}_{I_\ell}\boldsymbol{X}\boldsymbol{F}^\top)\varphi(\boldsymbol{L}_{I_\ell}\boldsymbol{X}\boldsymbol{F}^\top)^\top\right)^+\right] = \frac{1}{n}\operatorname{tr}\left[\left(\frac{1}{n}\varphi(\boldsymbol{X}\boldsymbol{F}^\top)\varphi(\boldsymbol{X}\boldsymbol{F}^\top)^\top + \overline{\lambda}_n\boldsymbol{I}_n\right)^{-1}\right].$$

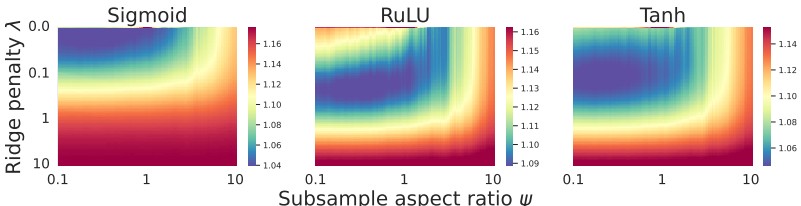

Figure 4: Heat map of prediction risks of full-ensemble ridge estimators (approximated with $M = 100$) using random features, for varying ridge penalties $\lambda$, subsample aspect ratios $\psi = p/d$ on the log-log scale. We consider random features $\varphi(\boldsymbol{F}\boldsymbol{x}_i) \in \mathbb{R}^d$, where $\varphi$ is an activation function (sigmoid, ReLU, or tanh). The data model is described in Appendix F.6 with $p = 250$, $d = 500$, $n = 5000$, and $\phi = d/n = 0.1$. As in Theorem 1, we see clear risk equivalence paths across activations.

*Define the data-dependent path $\mathcal{P}_n = \mathcal{P}(\bar{\lambda}_n; \phi_n, \bar{\psi}_n)$. Then, the conclusions in Theorem 1, Proposition 2, and Theorem 3 continue to hold on $(\varphi(\boldsymbol{X}\boldsymbol{F}^\top), \boldsymbol{y})$ with $\mathcal{P}$ replaced by $\mathcal{P}_n$.*

**Kernel features.** We can also consider kernel ridge regression. For a given feature map $\Phi : \mathbb{R}^p \to \mathbb{R}^d$, the kernel ridge estimator (in the primal form) is defined as:

$$\widehat{\boldsymbol{\beta}}_k^\lambda(\mathcal{D}_I) = \underset{\boldsymbol{\beta} \in \mathbb{R}^p}{\arg\min} \sum_{i \in I} (k^{-1/2}y_i - k^{-1/2}\Phi(\boldsymbol{x}_i)^\top \boldsymbol{\beta})^2 + \lambda\|\boldsymbol{\beta}\|_2^2 \sum_{i \in I}(y_i - \Phi(\boldsymbol{x}_i)^\top \boldsymbol{\beta})^2 + \frac{k}{p}\lambda\|\boldsymbol{\beta}\|_2^2.$$

Leveraging the kernel trick, the preceding optimization problem translates to solving the following problem (in the dual domain): $\widehat{\boldsymbol{\alpha}}_k^\lambda(\mathcal{D}_I) = \arg\min_{\boldsymbol{\alpha} \in \mathbb{R}^k} \boldsymbol{\alpha}^\top (\boldsymbol{K}_I + k\lambda\boldsymbol{I}_k)\boldsymbol{\alpha} + \boldsymbol{\alpha}^\top \boldsymbol{y}_I$, where $\boldsymbol{K}_I = \boldsymbol{\Phi}_I\boldsymbol{\Phi}_I^\top \in \mathbb{R}^{k \times k}$ is the kernel matrix and $\boldsymbol{\Phi}_I = (\Phi(\boldsymbol{x}_i))_{i \in I} \in \mathbb{R}^{n \times d}$ is the feature matrix. The correspondence between the dual and primal solutions is simply given by: $\widehat{\boldsymbol{\beta}}_k^\lambda(\mathcal{D}_I) = \boldsymbol{\Phi}_I^\top \widehat{\boldsymbol{\alpha}}_k^\lambda(\mathcal{D}_I)$.

Figure F8 illustrate results for kernel ridge regression using the same data-generating process as in the previous subsection. The figure shows the prediction risk of kernel ridge ensembles for polynomial, Gaussian, and Laplace kernels exhibit similar equivalence patterns, leading us to formulate an analogous conjecture for kernel ridge regression (which when $\Phi(\boldsymbol{x}) = \boldsymbol{x}$ gives back Proposition 4 with appropriate rescaling of features):

**Conjecture 8** (Data-dependent equivalences for kernel features (informal)). *Suppose Assumptions 1–2 hold. Define $\phi_n = p/n$. Suppose the kernel $K$ satisfies certain regularity conditions. Let $k \leq n$ be the subsample size and denote by $\bar{\psi}_n = p/k$. For any $M \in \mathbb{N} \cup \{\infty\}$, let $\bar{\lambda}_n$ be a solution to*

$$\frac{1}{M}\sum_{\ell=1}^{M} \operatorname{tr}\left[\boldsymbol{K}_{I_\ell}^+\right] = \operatorname{tr}\left[\left(\boldsymbol{K}_{[n]} + \frac{n}{p}\bar{\lambda}_n\boldsymbol{I}_n\right)^{-1}\right].$$

*Define the data-dependent path $\mathcal{P}_n = \mathcal{P}(\bar{\lambda}_n; \phi_n, \bar{\psi}_n)$. Then, the conclusions in Theorem 1, Proposition 2, and Theorem 3 continue to hold for kernel ridge ensembles with $\mathcal{P}$ replaced by $\mathcal{P}_n$.*

The empirical evidence here strongly suggests that the relationship between subsampling and ridge regression, which we have proved for ridge regression, holds true at least when the gram matrix "linearizes" in the sense of asymptotic equivalence [14, 54, 55]. This intriguing observation opens up a whole host of avenues towards fully understanding the universality of this relationship and establishing precise connections for a broader range of models [17, 18, 20, 21]. We hope to share more on these compelling directions in the near future.

**Tying other implicit regularizations.** Finally, as noted in related work, the equivalences between ridge regularization and subsampling established in this paper naturally offer opportunities to interpret and understand the other implicit regularization effects such as dropout regularization [27, 28], early stopping in gradient descent variants [36, 38, 39]. Furthermore, these equivalences provide avenues to understand the combined effects of these forms of implicit regularization, such as the effects of both subsample aggregating and gradient descent in mini-batch gradient descent, and contrast them to explicit regularization. Whether the explicit regularization can always be cleanly expressed as ridge-like regularization or takes a more generic form is an intriguing question. Exploring this is exciting, and we hope our readers also share this excitement!

## Acknowledgments and Disclosure of Funding

We are indebted to Ryan Tibshirani, Alessandro Rinaldo, Arun Kuchibhotla, Yuting Wei, Matey Neykov, Edgar Dobriban, Daniel LeJeune, Shamindra Shrotriya, Yuchen Wu for many insightful conversations. We are also grateful to anonymous reviewers of the precursor [13] for encouraging us to work on several follow-up directions that make up parts of this successor. On a more personal note: The lack of proof of risk monotonicity of optimally-tuned ridge in general had been "monotone bugging" (in a good way) the first author, ever since plotting Figure 8 (and other similar plots) in [51]. The resolution in Theorem 6 is of great personal significance to him and he is heartily appreciative of all the collaborators on related threads, especially the second author, for being a dedicated climbing partner on this technical "ascent". At the risk of over analogizing, the metaphor of letting a nut soak in water instead of forcibly cracking it open with a hammer, from Grothendieck[2], may perhaps describe the proof of Theorem 6. Establishing subsampling and ridge equivalences first and using it to demonstrate ridge monotonicity, in retrospect, seems a much less arduous route than a direct brute-force attempt (with calculus, say). But we will let the readers decide!

This work used the Bridges2 system at the Pittsburgh Supercomputing Center (PSC) through allocations MTH230020 from the Advanced Cyberinfrastructure Coordination Ecosystem: Services & Support (ACCESS) program. The code for reproducing the results of this paper can be found at `https://jaydu1.github.io/overparameterized-ensembling/equiv`.

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

# Supplementary material for
# "Generalized equivalences between
# subsampling and ridge regularization"

This document serves as a supplement to the paper "Generalized equivalences betweensubsampling and ridge regularization." The initial (unnumbered) section of the supplement provides a summary of the notation used in both the paper and supplement, followed by an organization for the supplement.

**Notation.** Below we provide an overview of the notation used throughout.

General notation:

| Notation | Description |
|---|---|
| Non-bold | Denotes univariates (scalars, functions, distributions, etc.) (e.g., $\lambda$, $f$, $H$). |
| Bold lower case | Denotes vectors (e.g., $\boldsymbol{x}$, $\boldsymbol{\beta}$). |
| Bold upper case | Denotes matrices (e.g., $\boldsymbol{X}$). |
| Calligraphic font | Denotes sets (e.g., $\mathcal{D}$). |
| Script font | Denotes certain limiting functions (e.g., $\mathscr{R}$ in (27)). |
| $\mathbb{N}$ | Set of positive integers. |
| $\mathbb{R}$, $\mathbb{R}_+$ | Set of real numbers and positive real numbers. |
| $\mathbb{C}$, $\mathbb{C}^+$ | Set of complex numbers and complex numbers with positive imaginary part. |
| $[n]$ | Set $\{1, \ldots, n\}$ for natural number $n$. |
| $(x)_+$ | The positive part of real number $x$. |
| $\lfloor x \rfloor$, $\lceil x \rceil$ | The floor and ceiling of real number $x$. |
| $\|f\|_{L_2}$ | The $L_2$ norm of function $f$, $\|f\|_{L_2} = \mathbb{E}_{\boldsymbol{x}}[f^2(\boldsymbol{x})]$. |
| $\|\boldsymbol{\beta}\|_2$ | The $\ell_2$ norm of vector $\boldsymbol{\beta}$. |
| $\langle \boldsymbol{v}, \boldsymbol{w} \rangle$ | The inner product of vectors $\boldsymbol{v}$ and $\boldsymbol{w}$. |
| $\boldsymbol{X}^\top$, $\boldsymbol{X}^+$ | The transpose and Moore-Penrose inverse of matrix $\boldsymbol{X} \in \mathbb{R}^{n \times p}$. |
| $\mathrm{tr}[\boldsymbol{A}]$, $\boldsymbol{A}^{-1}$ | The trace and inverse (if invertible) of a square matrix $\boldsymbol{A} \in \mathbb{R}^{p \times p}$. |
| $\boldsymbol{\Sigma}^{1/2}$ | The principal square root of positive semi-definite matrix $\boldsymbol{\Sigma}$. |
| $\boldsymbol{I}_p$ or $\boldsymbol{I}$ | The $p \times p$ identity matrix. |
| $\|\boldsymbol{X}\|_{\mathrm{op}}$ | The operator norm (spectral norm) of real matrix $\boldsymbol{X}$. |
| $\|\boldsymbol{X}\|_{\mathrm{tr}}$ | The trace norm (nuclear norm) $\mathrm{tr}[(\boldsymbol{X}^\top \boldsymbol{X})^{1/2}]$ of real matrix $\boldsymbol{X}$. |
| $f(\boldsymbol{A})$ | The matrix $\boldsymbol{V} f(\boldsymbol{R}) \boldsymbol{V}^{-1}$ where $\boldsymbol{A} = \boldsymbol{V} \boldsymbol{R} \boldsymbol{V}^{-1}$ is eigenvalue decomposition, and $f(\boldsymbol{R})$ is $f$ applied (elementwise) to the diagonals of $\boldsymbol{R}$. |
| $\boldsymbol{A} \preceq \boldsymbol{B}$ | The Loewner ordering for symmetric matrices $\boldsymbol{A}$ and $\boldsymbol{B}$. |
| $\xrightarrow{\mathrm{p}}$, $\xrightarrow{\mathrm{a.s.}}$, $\xrightarrow{\mathrm{d}}$ | Almost sure convergence, convergence in probability, and weak convergence. |

Specific notation:

| Notation | Description |
|---|---|
| $(\boldsymbol{x}, y)$ | The population random vector in $\mathbb{R}^p \times \mathbb{R}$. |
| $\mathcal{D}_n$ | A dataset containing i.i.d. samples of $(\boldsymbol{x}, y)$: $\mathcal{D}_n = \{(\boldsymbol{x}_i, y_i) : i \in [n]\}$. |
| $\boldsymbol{X}, \boldsymbol{y}$ | The feature matrix in $\mathbb{R}^{n \times p}$ and the response vector in $\mathbb{R}^n$ |
| $\mathcal{D}_I$ | A subsampled dataset $\mathcal{D}_I = \{(\boldsymbol{x}_i, y_i) : i \in I\}$. |
| $\widehat{\boldsymbol{\beta}}_k^\lambda(\mathcal{D}_I)$ | A ridge estimator fitted on $\mathcal{D}_I$ with $|I| = k$ observations and ridge penalty $\lambda$. |
| $\widehat{\boldsymbol{\beta}}_{k,M}^\lambda(\mathcal{D}_n; \{I_\ell\}_{\ell=1}^M)$ | An $M$-ensemble ridge estimator fitted on $\{\mathcal{D}_{I_\ell}\}_{\ell=1}^M$ with subsample size $k$ and ridge penalty $\lambda$. |
| $\boldsymbol{\beta}_0$ | The best (population) linear projection coefficient of $y$ onto $\boldsymbol{x}$: $\mathbb{E}[\boldsymbol{x}\boldsymbol{x}^\top]^{-1}\mathbb{E}[\boldsymbol{x}y]$. |
| $f_{\mathrm{LI}}(\boldsymbol{x})$ | The best (population) linear projection of $y$ onto $\boldsymbol{x}$: $\boldsymbol{x}^\top \boldsymbol{\beta}_0$. |
| $f_{\mathrm{NL}}(\boldsymbol{x})$ | The component of $y$ that is not explained by $\boldsymbol{x}$: $y - \boldsymbol{x}^\top \boldsymbol{\beta}_0$. |
| $\boldsymbol{f}_{\mathrm{LI}}, \boldsymbol{f}_{\mathrm{NL}}$ | $\boldsymbol{f}_{\mathrm{LI}} = \boldsymbol{X}\boldsymbol{\beta}_0$, $\boldsymbol{f}_{\mathrm{NL}} = [f_{\mathrm{NL}}(\boldsymbol{x}_i)]_{i \in [n]}$. |

A note on indexing of sequences:

In the subsequent sections, we will prove the results for $n, k, p$ being a sequence of integers $\{n_m\}_{m=1}^{\infty}$, $\{k_m\}_{m=1}^{\infty}$, $\{p_m\}_{m=1}^{\infty}$. Alternatively, one can also view $k$ and $p$ as sequences $k_n$ and $p_n$ that are indexed by $n$. For notational brevity, we drop the subscript when it is clear from the context.

**Organization.** Below we outline the structure of the rest of the supplement. The technical lemmas refer to the main ingredients that we use to prove results in the corresponding sections.

Table 3: Outline of the supplement.

| Section | Description | Technical lemmas |
|---------|-------------|------------------|
| Appendix A | Proof of Theorem 1, Proposition 2 (from Section 3) | Lemma A.1, Lemma A.2, Lemma A.3 |
| Appendix B | Proof of Theorem 3, Proposition 4, Corollary 5 (from Section 4) | Lemma B.1, Lemma B.2 |
| Appendix C | Proof of Theorem 6 (from Section 5) | Lemma C.1 |
| Appendix D | New asymptotic equivalents, concentration results and other useful lemmas (used in Appendices A–C) | Lemma D.1, Lemma D.2, Lemma D.3, Lemma D.5 |
| Appendix E | Asymptotic equivalents: background and known results (used in Appendices A–D) | |
| Appendix F | Additional details for all the experiments | |

# A    Proofs of results in Section 3

## A.1    Proof of Theorem 1

Given an observation $(\boldsymbol{x}, y)$, recall the decomposition $y = f_{\mathrm{LI}}(\boldsymbol{x}) + f_{\mathrm{NL}}(\boldsymbol{x})$ explained in Section 2. For $n$ i.i.d. samples from the same distribution as $(\boldsymbol{x}, y)$, we define analogously the vector decomposition:

$$\boldsymbol{y} = \boldsymbol{f}_{\mathrm{LI}} + \boldsymbol{f}_{\mathrm{NL}}, \tag{10}$$

where $\boldsymbol{f}_{\mathrm{LI}} = \boldsymbol{X}\boldsymbol{\beta}_0$ and $\boldsymbol{f}_{\mathrm{NL}} = [f_{\mathrm{NL}}(\boldsymbol{x}_i)]_{i\in[n]}$. Let $n_{\boldsymbol{A}} = \mathrm{nrow}(\boldsymbol{A})$. Note that

$$R(\widehat{\boldsymbol{\beta}}_{k,\infty}^{\lambda}; \boldsymbol{A}, \boldsymbol{b}, \boldsymbol{\beta}_0) = R(\widehat{\boldsymbol{\beta}}_{k,\infty}^{\lambda}; \boldsymbol{A}, \boldsymbol{0}, \boldsymbol{\beta}_0) + 2n_{\boldsymbol{A}}^{-1}\boldsymbol{b}^{\top}\boldsymbol{A}(\widehat{\boldsymbol{\beta}}_{k,\infty}^{\lambda} - \boldsymbol{\beta}_0) + n_{\boldsymbol{A}}^{-1}\|\boldsymbol{b}\|_2^2.$$

By Theorem 3, the cross term vanishes, i.e., $n_{\boldsymbol{A}}^{-1}\boldsymbol{b}^{\top}\boldsymbol{A}(\widehat{\boldsymbol{\beta}}_{k,\infty}^{\lambda} - \boldsymbol{\beta}_0) \xrightarrow{\mathrm{a.s.}} 0$. We then have

$$|R(\widehat{\boldsymbol{\beta}}_{k_1,\infty}^{\lambda_1}; \boldsymbol{A}, \boldsymbol{b}, \boldsymbol{\beta}_0) - R(\widehat{\boldsymbol{\beta}}_{k_2,\infty}^{\lambda_2}; \boldsymbol{A}, \boldsymbol{b}, \boldsymbol{\beta}_0)| \xrightarrow{\mathrm{a.s.}} |R(\widehat{\boldsymbol{\beta}}_{k_1,\infty}^{\lambda_1}; \boldsymbol{A}, \boldsymbol{0}, \boldsymbol{\beta}_0) - R(\widehat{\boldsymbol{\beta}}_{k_2,\infty}^{\lambda_2}; \boldsymbol{A}, \boldsymbol{0}, \boldsymbol{\beta}_0)|.$$

It suffices to analyze $R(\widehat{\boldsymbol{\beta}}_{k,\infty}^{\lambda}; \boldsymbol{A}, \boldsymbol{0}, \boldsymbol{\beta}_0)$.

For simplicity, we treat $\boldsymbol{A}$ as the normalized matrix $n_{\boldsymbol{A}}^{-1/2}\boldsymbol{A}$ to avoid the notation of $\mathrm{nrow}(\boldsymbol{A})$. Note that $\boldsymbol{y} = \boldsymbol{X}\boldsymbol{\beta}_0 + \boldsymbol{f}_{\mathrm{NL}} + \boldsymbol{\varepsilon}$, where $\boldsymbol{\beta}_0$ is the best linear projection of $\boldsymbol{y}$ on $\boldsymbol{X}$, $\boldsymbol{f}_{\mathrm{NL}}$ is the nonlinear residual and $\boldsymbol{\varepsilon}$ is the independent noise. Let $\boldsymbol{A}_k^{\lambda} = \mathbb{E}_{I\sim\mathcal{I}_k}[(\boldsymbol{X}^{\top}\boldsymbol{L}_I\boldsymbol{X}/k + \lambda\boldsymbol{I}_p)^{-1}\boldsymbol{X}^{\top}\boldsymbol{L}_I/k]$ and $\boldsymbol{B}_k^{\lambda} = \boldsymbol{I}_p - \boldsymbol{A}_k^{\lambda}\boldsymbol{X} = \mathbb{E}_{I\sim\mathcal{I}_k}[\lambda(\boldsymbol{X}^{\top}\boldsymbol{L}_I\boldsymbol{X}/k + \lambda\boldsymbol{I}_p)^{-1}]$. We begin by decomposing the generalized risk for arbitrary $(\lambda, k)$ into different terms:

$$\begin{aligned}
R(\widehat{\boldsymbol{\beta}}_{k,\infty}^{\lambda}; \boldsymbol{A}, \boldsymbol{0}, \boldsymbol{\beta}_0) &= \|\boldsymbol{A}(\widehat{\boldsymbol{\beta}}_{k,\infty}^{\lambda} - \boldsymbol{\beta}_0)\|_2^2 \\
&= \|\boldsymbol{A}(\boldsymbol{A}_k^{\lambda}\boldsymbol{y} - \boldsymbol{\beta}_0)\|_2^2 \\
&= \|\boldsymbol{A}\boldsymbol{A}_k^{\lambda}(\boldsymbol{X}\boldsymbol{\beta}_0 + \boldsymbol{f}_{\mathrm{NL}}) - \boldsymbol{A}\boldsymbol{\beta}_0\|_2^2 \\
&= \boldsymbol{\beta}_0^{\top}(\boldsymbol{A}_k^{\lambda}\boldsymbol{X} - \boldsymbol{I}_p)^{\top}\boldsymbol{A}^{\top}\boldsymbol{A}(\boldsymbol{A}_k^{\lambda}\boldsymbol{X} - \boldsymbol{I}_p)\boldsymbol{\beta}_0 + \boldsymbol{f}_{\mathrm{NL}}^{\top}(\boldsymbol{A}_k^{\lambda})^{\top}\boldsymbol{A}^{\top}\boldsymbol{A}\boldsymbol{A}_k^{\lambda}\boldsymbol{f}_{\mathrm{NL}} \\
&\quad + 2\boldsymbol{f}_{\mathrm{NL}}^{\top}(\boldsymbol{A}_k^{\lambda})^{\top}\boldsymbol{A}^{\top}\boldsymbol{A}(\boldsymbol{A}_k^{\lambda}\boldsymbol{X} - \boldsymbol{I}_p)\boldsymbol{\beta}_0 \\
&= \boldsymbol{\beta}_0^{\top}\boldsymbol{B}_k^{\lambda}\boldsymbol{A}^{\top}\boldsymbol{A}\boldsymbol{B}_k^{\lambda}\boldsymbol{\beta}_0 + \boldsymbol{f}_{\mathrm{NL}}^{\top}(\boldsymbol{A}_k^{\lambda})^{\top}\boldsymbol{A}^{\top}\boldsymbol{A}\boldsymbol{A}_k^{\lambda}\boldsymbol{f}_{\mathrm{NL}} - 2\boldsymbol{f}_{\mathrm{NL}}^{\top}(\boldsymbol{A}_k^{\lambda})^{\top}\boldsymbol{A}^{\top}\boldsymbol{A}\boldsymbol{B}_k^{\lambda}\boldsymbol{\beta}_0.
\end{aligned}$$

When $\lambda > 0$, from Lemma A.3, we know that the cross terms $2\boldsymbol{f}_{\mathrm{NL}}^\top (\boldsymbol{A}_k^\lambda)^\top \boldsymbol{A}^\top \boldsymbol{A} \boldsymbol{B}_k^\lambda \boldsymbol{\beta}_0$ vanishes as $p$ tends to infinity. Further, by Lemma A.1 and Lemma A.2, it follows that when $\boldsymbol{A} \perp\!\!\!\perp (\boldsymbol{X}, \boldsymbol{y})$, as $p/n \to \phi$ and $p/k \to \psi$,

$$|R(\widehat{\boldsymbol{\beta}}_{k,\infty}^\lambda; \boldsymbol{A}, \boldsymbol{0}, \boldsymbol{\beta}_0) - R_p(\lambda; \phi, \psi)| \xrightarrow{\text{a.s.}} 0,$$

where the function $R_p$ is defined as

$$R_p(\lambda; \phi, \psi) = \widetilde{c}_p(-\lambda; \phi, \psi, \boldsymbol{A}^\top \boldsymbol{A}) + \|f_{\mathrm{NL}}\|_{L_2}^2 \widetilde{v}_p(-\lambda; \phi, \psi, \boldsymbol{A}^\top \boldsymbol{A}), \tag{11}$$

and the nonnegative constants $\widetilde{c}_p(-\lambda; \phi, \psi, \boldsymbol{A}^\top \boldsymbol{A})$ and $\widetilde{v}_p(-\lambda; \phi, \psi, \boldsymbol{A}^\top \boldsymbol{A})$ are as defined in Lemma A.1.

When $\lambda = 0$, as in the proof of Theorem 3, we again show that $P_{n,\lambda} - Q_{n,\lambda}$ is equivcontinuous over $\Lambda = [0, \lambda_{\max}]$ for any $\lambda_{\max} \in (0, \infty)$ fixed, where we define

$$P_{n,\lambda} = R(\widehat{\boldsymbol{\beta}}_{k,\infty}^\lambda; \boldsymbol{A}, \boldsymbol{0}, \boldsymbol{\beta}_0), \quad \text{and} \quad Q_{n,\lambda} = R_p(\lambda; \phi, \psi).$$

When $\psi \neq 1$, it can be verified that $|P_{n,\lambda}|$, $|\partial P_{n,\lambda}/\partial\lambda|$, $|Q_{n,\lambda}|$, $|\partial Q_{n,\lambda}/\partial\lambda|$ are bounded almost surely. Thus, by the Moore-Osgood theorem, it follows that, with probability one,

$$\lim_{n\to\infty} |R(\widehat{\boldsymbol{\beta}}_{k,\infty}^0; \boldsymbol{A}, \boldsymbol{0}, \boldsymbol{\beta}_0) - R_p(0; \phi, \psi)| = \lim_{\lambda\to 0^+} \lim_{n\to\infty} |R(\widehat{\boldsymbol{\beta}}_{k,\infty}^\lambda; \boldsymbol{A}, \boldsymbol{0}, \boldsymbol{\beta}_0) - R_p(\lambda; \phi, \psi)| = 0.$$

Note that $\widetilde{c}_p(-\lambda; \phi, \psi, \boldsymbol{A}^\top \boldsymbol{A})$ and $\widetilde{v}_p(-\lambda; \phi, \psi, \boldsymbol{A}^\top \boldsymbol{A})$ are functions of the fixed-point solution $v_p(-\lambda; \psi)$. By Lemma C.1 we have that for $\overline{\psi} \in [\phi, \infty]$ there exists a segment $\mathcal{P}$ such that for all $(\lambda_1, \psi_1), (\lambda_2, \psi_2) \in \mathcal{P}$, it holds that $v(-\lambda_1; \psi_1) = v(-\lambda_2; \psi_2)$ as $p/k_1 \to \psi_1$ and $p/k_2 \to \psi_2$. This implies that

$$R_p(\lambda_1; \phi, \psi_1) = R_p(\lambda_2; \phi, \psi_2).$$

Thus, by triangle inequality, we have

$$|R(\widehat{\boldsymbol{\beta}}_{k_1,\infty}^{\lambda_1}; \boldsymbol{A}, \boldsymbol{0}, \boldsymbol{\beta}_0) - R(\widehat{\boldsymbol{\beta}}_{k_2,\infty}^{\lambda_2}; \boldsymbol{A}, \boldsymbol{0}, \boldsymbol{\beta}_0)| \xrightarrow{\text{a.s.}} 0,$$

which completes the proof.

## A.2 Proof of Proposition 2

When $\boldsymbol{A} = \boldsymbol{X}$ and $\boldsymbol{b} = \boldsymbol{f}_{\mathrm{NL}}$, the generalized risk reduces to

$$
\begin{aligned}
R(\widehat{\boldsymbol{\beta}}_{k,\infty}^\lambda; \boldsymbol{A}, \boldsymbol{b}, \boldsymbol{\beta}_0) &= \frac{1}{n}\|\boldsymbol{X}(\widehat{\boldsymbol{\beta}}_{k,\infty}^\lambda - \boldsymbol{\beta}_0) - \boldsymbol{f}_{\mathrm{NL}}\|_2^2 \\
&= \frac{1}{n}\|\boldsymbol{X}\widehat{\boldsymbol{\beta}}_{k,\infty}^\lambda - (\boldsymbol{X}\boldsymbol{\beta}_0 + \boldsymbol{f}_{\mathrm{NL}})\|_2^2 \\
&= \frac{1}{n}\|\boldsymbol{X}\boldsymbol{A}_k^\lambda(\boldsymbol{X}\boldsymbol{\beta}_0 + \boldsymbol{f}_{\mathrm{NL}}) - (\boldsymbol{X}\boldsymbol{\beta}_0 + \boldsymbol{f}_{\mathrm{NL}})\|_2^2 \\
&= \boldsymbol{\beta}_0^\top \boldsymbol{B}_k^\lambda \widehat{\boldsymbol{\Sigma}} \boldsymbol{B}_k^\lambda \boldsymbol{\beta}_0 + \frac{1}{n}\boldsymbol{f}_{\mathrm{NL}}^\top (\boldsymbol{I}_p - \boldsymbol{X}\boldsymbol{A}_k^\lambda)^\top (\boldsymbol{I}_p - \boldsymbol{X}\boldsymbol{A}_k^\lambda)\boldsymbol{f}_{\mathrm{NL}} \\
&\quad + \frac{2}{n}\boldsymbol{f}_{\mathrm{NL}}^\top (\boldsymbol{I}_p - \boldsymbol{X}\boldsymbol{A}_k^\lambda)^\top \boldsymbol{X}\boldsymbol{B}_k^\lambda \boldsymbol{\beta}_0.
\end{aligned}
$$

The proof then follows analogously as in Theorem 1 by involving Lemma A.1 and Lemma A.2 with $\boldsymbol{A} = n^{-1/2}\boldsymbol{X}$.

When $\boldsymbol{A} = \boldsymbol{X}$ and $\boldsymbol{b} = \boldsymbol{0}$, we have

$$R(\widehat{\boldsymbol{\beta}}_{k,\infty}^\lambda; \boldsymbol{A}, \boldsymbol{b}, \boldsymbol{\beta}_0) = \boldsymbol{\beta}_0^\top \boldsymbol{B}_k^\lambda \widehat{\boldsymbol{\Sigma}} \boldsymbol{B}_k^\lambda \boldsymbol{\beta}_0 + \boldsymbol{f}_{\mathrm{NL}}^\top (\boldsymbol{A}_k^\lambda)^\top \widehat{\boldsymbol{\Sigma}} \boldsymbol{A}_k^\lambda \boldsymbol{f}_{\mathrm{NL}} - 2\boldsymbol{f}_{\mathrm{NL}}^\top (\boldsymbol{A}_k^\lambda)^\top \widehat{\boldsymbol{\Sigma}} \boldsymbol{B}_k^\lambda \boldsymbol{\beta}_0.$$

Invoking Lemma A.2 for $\boldsymbol{f}_{\mathrm{NL}}^\top (\boldsymbol{A}_k^\lambda)^\top \widehat{\boldsymbol{\Sigma}} \boldsymbol{A}_k^\lambda \boldsymbol{f}_{\mathrm{NL}}$ instead of $\boldsymbol{f}_{\mathrm{NL}}^\top (\boldsymbol{I}_p - \boldsymbol{X}\boldsymbol{A}_k^\lambda)^\top (\boldsymbol{I}_p - \boldsymbol{X}\boldsymbol{A}_k^\lambda)\boldsymbol{f}_{\mathrm{NL}}$, the proof follows analogously.

## A.3 Technical lemmas

**Lemma A.1** (Bias of generalized risk)**.** *Suppose the same assumptions in Theorem 1 hold with $\lambda > 0$.*
*Let $B_k^\lambda = I_p - A_k^\lambda X = \mathbb{E}_{I \sim \mathcal{I}_k}[\lambda(X^\top L_I X/k + \lambda I_p)^{-1}]$. For any $A$ with $\limsup \|A\|_{\mathrm{op}}$ bounded*
*almost surely, the following statements hold:*

*(1) If $A \perp\!\!\!\perp (X, y)$, then*

$$\beta_0^\top B_k^\lambda A^\top A B_k^\lambda \beta_0 - \widetilde{c}_p(-\lambda; \phi, \psi, A^\top A) \xrightarrow{\text{a.s.}} 0.$$

*(2) If $A = n^{-1/2} X$, then*

$$\beta_0^\top B_k^\lambda A^\top A B_k^\lambda \beta_0 - \left(1 - \phi \int \frac{v_p(-\lambda; \psi)r}{1 + v_p(-\lambda; \psi)r}\, \mathrm{d}H_p(r)\right)^2 \widetilde{c}_p(-\lambda; \phi, \psi, \Sigma) \xrightarrow{\text{a.s.}} 0.$$

*Here the nonnegative constants $v_p(-\lambda; \psi)$, $\widetilde{v}_p(-\lambda; \phi, \psi, A^\top A)$ and $\widetilde{c}_p(-\lambda; \phi, \psi, A^\top A)$ are defined*
*through the following equations:*

$$\frac{1}{v_p(-\lambda; \psi)} = \lambda + \psi \int \frac{r}{1 + v_p(-\lambda; \psi)r}\, \mathrm{d}H_p(r),$$

$$\widetilde{v}_p(-\lambda; \phi, \psi, A^\top A) = \frac{\phi \operatorname{tr}[A^\top A \Sigma (v_p(-\lambda; \psi)\Sigma + I_p)^{-2}]/p}{v_p(-\lambda; \psi)^{-2} - \phi \int \dfrac{r^2}{(1 + v_p(-\lambda; \psi)r)^2}\, \mathrm{d}H_p(r)},$$

$$\widetilde{c}_p(-\lambda; \phi, \psi, A^\top A) = \beta_0^\top (v_p(-\lambda; \psi)\Sigma + I_p)^{-1}(\widetilde{v}_p(-\lambda; \phi, \psi, A^\top A)\Sigma + A^\top A)(v_p(-\lambda; \psi)\Sigma + I_p)^{-1}\beta_0.$$

*Proof.* Note that $\beta_0$ is independent of

$$B_k^\lambda A^\top A B_k^\lambda = \lambda^2 \mathbb{E}_{I_1, I_2 \sim \mathcal{I}_k}[M_1 A^\top A M_2],$$

where $\widehat{\Sigma}_{1 \cap 2} = X^\top L_{I_1 \cap I_2} X/|I_1 \cap I_2|$ and $M_j = (X^\top L_{I_1} X/k + \lambda I_p)^{-1}$ for $j = 1, 2$. We analyze
the deterministic equivalents of the latter for the two cases.

(1) $\underline{A \perp\!\!\!\perp (X, y).}$

From Lemma D.1 (2), we know that when $A$ is independent to $(X, y)$, it follows that

$$\lambda^2 \mathbb{E}_{I_1, I_2 \sim \mathcal{I}_k}[M_1 A^\top A M_2]$$
$$\simeq (v_p(-\lambda; \psi)\Sigma + I_p)^{-1}(\widetilde{v}_p(-\lambda; \phi, \psi, A^\top A)\Sigma + A^\top A)(v_p(-\lambda; \psi)\Sigma + I_p)^{-1}.$$

Then, by the trace property of deterministic equivalents in Lemma E.3 (4), we have

$$\beta_0^\top B_k^\lambda A^\top A B_k^\lambda \beta_0$$
$$\stackrel{\text{a.s.}}{=\!=\!=} \beta_0^\top (v_p(-\lambda; \psi)\Sigma + I_p)^{-1}(\widetilde{v}_p(-\lambda; \phi, \psi, A^\top A)\Sigma + A^\top A)(v_p(-\lambda; \psi)\Sigma + I_p)^{-1}\beta_0$$
$$\stackrel{\text{a.s.}}{=\!=\!=} \widetilde{c}_p(-\lambda; \phi, \psi, A^\top A).$$

(2) $\underline{A = n^{-1/2} X.}$

From Lemma D.1 (4), it follows that

$$\lambda^2 \mathbb{E}_{I_1, I_2 \sim \mathcal{I}_k}[M_1 \widehat{\Sigma} M_2] \simeq \left(1 - \phi \int \frac{v_p(-\lambda; \psi)r}{1 + v_p(-\lambda; \psi)r}\, \mathrm{d}H_p(r)\right)$$
$$\cdot (1 + \widetilde{v}_p(-\lambda; \phi, \psi))(v_p(-\lambda; \psi)\Sigma + I_p)^{-2}\Sigma.$$

Then, by the trace property of deterministic equivalents in Lemma E.3 (4), we have

$$\beta_0^\top B_k^\lambda \widehat{\Sigma} B_k^\lambda \beta_0 \stackrel{\text{a.s.}}{=\!=\!=} \left(1 - \phi \int \frac{v_p(-\lambda; \psi)r}{1 + v_p(-\lambda; \psi)r}\, \mathrm{d}H_p(r)\right)(1 + \widetilde{v}_p(-\lambda; \phi, \psi)) \cdot$$
$$\beta_0^\top (v_p(-\lambda; \psi)\Sigma + I_p)^{-1}\Sigma (v_p(-\lambda; \psi)\Sigma + I_p)^{-1}\beta_0$$
$$\stackrel{\text{a.s.}}{=\!=\!=} \left(1 - \phi \int \frac{v_p(-\lambda; \psi)r}{1 + v_p(-\lambda; \psi)r}\, \mathrm{d}H_p(r)\right)^2 \widetilde{c}_p(-\lambda; \phi, \psi, \Sigma).$$

$\square$

**Lemma A.2** (Variance term of generalized risk). *Suppose the same assumptions in Theorem 1 hold with $\lambda > 0$. Let $\boldsymbol{A}_k^\lambda = \mathbb{E}_{I \sim \mathcal{I}_k}[(\boldsymbol{X}^\top \boldsymbol{L}_I \boldsymbol{X}/k + \lambda \boldsymbol{I}_p)^{-1} \boldsymbol{X}^\top \boldsymbol{L}_I/k]$ and $\boldsymbol{B}_k^\lambda = \boldsymbol{I}_p - \boldsymbol{A}_k^\lambda \boldsymbol{X} = \mathbb{E}_{I \sim \mathcal{I}_k}[\lambda (\boldsymbol{X}^\top \boldsymbol{L}_I \boldsymbol{X}/k + \lambda \boldsymbol{I}_p)^{-1}]$. For any $\boldsymbol{A}$ with $\limsup \|\boldsymbol{A}\|_{\mathrm{op}}$ bounded almost surely, the following statements hold:*

*(1) If $\boldsymbol{A} \perp\!\!\!\perp (\boldsymbol{X}, \boldsymbol{y})$, then*

$$\boldsymbol{f}_{\mathrm{NL}}^\top (\boldsymbol{A}_k^\lambda)^\top \boldsymbol{A}^\top \boldsymbol{A} \boldsymbol{A}_k^\lambda \boldsymbol{f}_{\mathrm{NL}} - \|f_{\mathrm{NL}}\|_{L^2}^2 (1 + \widetilde{v}_p(-\lambda; \phi, \psi, \boldsymbol{\Sigma})) \xrightarrow{\text{a.s.}} 0.$$

*(2) If $\boldsymbol{A} = n^{-1/2} \boldsymbol{X}$, then*

$$\boldsymbol{f}_{\mathrm{NL}}^\top (\boldsymbol{A}_k^\lambda)^\top \boldsymbol{A}^\top \boldsymbol{A} \boldsymbol{A}_k^\lambda \boldsymbol{f}_{\mathrm{NL}}$$

$$\xrightarrow{\text{a.s.}} \|f_{\mathrm{NL}}\|_{L^2}^2 \left(1 - \phi \int \frac{v_p(-\lambda; \psi)r}{1 + v_p(-\lambda; \psi)r} \, \mathrm{d}H_p(r)\right)^2 \cdot (1 + \widetilde{v}_p(-\lambda; \phi, \psi, \boldsymbol{\Sigma}))$$

$$+ \|f_{\mathrm{NL}}\|_{L^2}^2 \left(2\phi \int \frac{v_p(-\lambda; \psi)r}{1 + v_p(-\lambda; \psi)r} \, \mathrm{d}H_p(r) - 1\right).$$

$$\frac{1}{n} \boldsymbol{f}_{\mathrm{NL}}^\top (\boldsymbol{X} \boldsymbol{A}_k^\lambda - \boldsymbol{I}_n)^\top (\boldsymbol{X} \boldsymbol{A}_k^\lambda - \boldsymbol{I}_n) \boldsymbol{f}_{\mathrm{NL}}$$

$$\xrightarrow{\text{a.s.}} \|f_{\mathrm{NL}}\|_{L^2}^2 \left(1 - \phi \int \frac{v_p(-\lambda; \psi)r}{1 + v_p(-\lambda; \psi)r} \, \mathrm{d}H_p(r)\right)^2 \cdot \sigma^2 (1 + \widetilde{v}_p(-\lambda; \phi, \psi, \boldsymbol{\Sigma})),$$

*Here the nonnegative constants $v_p(-\lambda; \psi)$ and $\widetilde{v}_p(-\lambda; \phi, \psi, \boldsymbol{A}^\top \boldsymbol{A})$ are defined through the following equations:*

$$\frac{1}{v_p(-\lambda; \psi)} = \lambda + \psi \int \frac{r}{1 + v_p(-\lambda; \psi)r} \, \mathrm{d}H_p(r),$$

$$\widetilde{v}_p(-\lambda; \phi, \psi, \boldsymbol{A}^\top \boldsymbol{A}) = \frac{\phi \operatorname{tr}[\boldsymbol{A}^\top \boldsymbol{A} \boldsymbol{\Sigma}(v_p(-\lambda; \psi)\boldsymbol{\Sigma} + \boldsymbol{I}_p)^{-2}]/p}{v_p(-\lambda; \psi)^{-2} - \phi \int \frac{r^2}{(1 + v_p(-\lambda; \psi)r)^2} \, \mathrm{d}H_p(r)}.$$

*Proof.* The two cases are treated separately below.

(1) $\underline{\boldsymbol{A} \perp\!\!\!\perp (\boldsymbol{X}, \boldsymbol{y}).}$

We split the proof into three different parts.

**Part (1) Intersection concentration.** Let $\boldsymbol{f}_{\mathrm{NL}} = \boldsymbol{L}_{I_1 \cap I_2} \boldsymbol{f}_{\mathrm{NL}} + \boldsymbol{L}_{I_1 \setminus I_2} \boldsymbol{f}_{\mathrm{NL}} + \boldsymbol{L}_{I_2 \setminus I_1} \boldsymbol{f}_{\mathrm{NL}} =: \boldsymbol{f}_0 + \boldsymbol{f}_1 + \boldsymbol{f}_2$. Note that

$$\boldsymbol{f}_{\mathrm{NL}}^\top \boldsymbol{A}_k^{\lambda_1 \top} \boldsymbol{A}^\top \boldsymbol{A} \boldsymbol{A}_k^{\lambda_1} \boldsymbol{f}_{\mathrm{NL}} = \mathbb{E}_{I_1, I_2 \sim \mathcal{I}_k} \left[(\boldsymbol{f}_0 + \boldsymbol{f}_1)^\top \frac{\boldsymbol{X}}{k} \boldsymbol{M}_1 \boldsymbol{A}^\top \boldsymbol{A} \boldsymbol{M}_2 \frac{\boldsymbol{X}^\top}{k} (\boldsymbol{f}_0 + \boldsymbol{f}_2)\right].$$

where $\boldsymbol{M}_j = (\boldsymbol{X}^\top \boldsymbol{L}_{I_2} \boldsymbol{X}/k + \lambda \boldsymbol{I}_p)^{-1}$. By conditional independence and Lemma S.8.5 of [51], the cross terms vanish

$$\boldsymbol{f}_1^\top \frac{\boldsymbol{X}}{k} \boldsymbol{M}_1 \boldsymbol{A}^\top \boldsymbol{A} \boldsymbol{M}_2 \frac{\boldsymbol{X}^\top \boldsymbol{L}_2}{k} \boldsymbol{f}_{\mathrm{NL}} \xrightarrow{\text{a.s.}} 0, \quad \text{and} \quad \boldsymbol{f}_{\mathrm{NL}}^\top \frac{\boldsymbol{L}_1 \boldsymbol{X}}{k} \boldsymbol{M}_1 \boldsymbol{A}^\top \boldsymbol{A} \boldsymbol{M}_2 \frac{\boldsymbol{X}^\top}{k} \boldsymbol{f}_2 \xrightarrow{\text{a.s.}} 0. \tag{12}$$

It remains to analyze the quadratic term of $\boldsymbol{f}_0$:

$$\boldsymbol{f}_0^\top \frac{\boldsymbol{L}_{I_1 \cap I_2} \boldsymbol{X}}{k} \boldsymbol{M}_1 \boldsymbol{A}^\top \boldsymbol{A} \boldsymbol{M}_2 \frac{\boldsymbol{X}^\top \boldsymbol{L}_{I_1 \cap I_2}}{k} \boldsymbol{f}_0.$$

By conditioning on $\boldsymbol{L}_{I_1 \cap I_2} \boldsymbol{X}$, from Lemma S.7.10 (1) of [12], we have

$$\boldsymbol{M}_j \simeq \boldsymbol{M}^{\mathrm{det}} := \frac{k}{i_0} (\widehat{\boldsymbol{\Sigma}}_0 + \lambda \boldsymbol{I}_p + \lambda \boldsymbol{C})^{-1},$$

where $\widehat{\boldsymbol{\Sigma}}_0 = \boldsymbol{X}_0^\top \boldsymbol{X}_0/i_0$, $\boldsymbol{X}_0 = \boldsymbol{L}_{I_1 \cap I_2}\boldsymbol{X}$, $\boldsymbol{C} = (k - i_0)/i_0 \cdot (v(-\lambda; \gamma_1, \boldsymbol{\Sigma}_{\boldsymbol{C}_1})\boldsymbol{\Sigma} + \boldsymbol{I}_p)$, $\boldsymbol{C}_1 = i_0(\lambda(k - i_0))^{-1}(\widehat{\boldsymbol{\Sigma}}_0 + \lambda\boldsymbol{I}_p)$ and $i_0 = |I_1 \cap I_2|$. Then we have

$$\frac{\boldsymbol{X}_0}{k}\boldsymbol{M}_1\boldsymbol{A}^\top\boldsymbol{A}\boldsymbol{M}_2\frac{\boldsymbol{X}_0^\top}{k} \simeq \frac{k^2}{i_0^2}\frac{\boldsymbol{X}_0}{k}\boldsymbol{M}^{\mathrm{det}}\boldsymbol{A}^\top\boldsymbol{A}\boldsymbol{M}^{\mathrm{det}}\frac{\boldsymbol{X}_0^\top}{k}$$

$$= \frac{k^2}{i_0^2}\frac{\boldsymbol{X}_0'}{k}\boldsymbol{M}'\boldsymbol{A}'\boldsymbol{M}'\frac{\boldsymbol{X}_0'^\top}{k} \tag{13}$$

where $\boldsymbol{X}_0' = \boldsymbol{X}_0(\boldsymbol{I}_p + \boldsymbol{C})^{-1/2}$, $\boldsymbol{A}' = (\boldsymbol{I}_p + \boldsymbol{C})^{-1/2}\boldsymbol{A}^\top\boldsymbol{A}(\boldsymbol{I}_p + \boldsymbol{C})^{-1/2}$, and $\boldsymbol{M}' = ((\boldsymbol{I}_p + \boldsymbol{C})^{-1/2}\widehat{\boldsymbol{\Sigma}}_0(\boldsymbol{I}_p + \boldsymbol{C})^{-1/2} + \lambda\boldsymbol{I}_p)^{-1}$.

**Part (2) Diagonal concentration.** We next use a similar strategy as in the proof of Lemma A.16 in [14] by showing that the off-diagonal summation vanishes. Let $\boldsymbol{\Sigma}' = (\boldsymbol{I}_p + \boldsymbol{C})^{-1/2}\boldsymbol{\Sigma}(\boldsymbol{I}_p + \boldsymbol{C})^{-1/2}$. Since $\boldsymbol{X}_0 = \boldsymbol{Z}_0\boldsymbol{\Sigma}^{1/2}$, we have $\boldsymbol{X}_0' = \boldsymbol{Z}_0\boldsymbol{\Sigma}'^{1/2}$. Then, the quadratic form becomes:

$$\frac{1}{k^2}\boldsymbol{f}_0^\top\boldsymbol{X}_0'\boldsymbol{M}'\boldsymbol{A}'\boldsymbol{M}'\boldsymbol{X}_0'^\top\boldsymbol{f}_0$$

$$= \frac{i_0}{k^2}\boldsymbol{f}_0^\top\left(\frac{\boldsymbol{Z}_0\boldsymbol{\Sigma}'\boldsymbol{Z}_0^\top}{i_0} + \lambda\boldsymbol{I}_n\right)^{-1}\frac{\boldsymbol{Z}_0}{\sqrt{i_0}}\boldsymbol{\Sigma}'^{\frac{1}{2}}\boldsymbol{A}'\boldsymbol{\Sigma}'^{\frac{1}{2}}\frac{\boldsymbol{Z}_0^\top}{\sqrt{i_0}}\left(\frac{\boldsymbol{Z}_0\boldsymbol{\Sigma}'\boldsymbol{Z}_0^\top}{i_0} + \lambda\boldsymbol{I}_n\right)^{-1}\boldsymbol{f}_0$$

$$=: \frac{i_0}{k^2}\boldsymbol{f}_0^\top\boldsymbol{B}_1^{-1}\boldsymbol{B}_2\boldsymbol{B}_1^{-1}\boldsymbol{f}_0. \tag{14}$$

Note that from Lemma D.4, we have for any $t > 0$,

$$\boldsymbol{B}_1^{-1}\boldsymbol{B}_2\boldsymbol{B}_1^{-1} = \frac{1}{t}(\boldsymbol{B}_1^{-1} - (\boldsymbol{B}_1 + t\boldsymbol{B}_2)^{-1}) + t\boldsymbol{B}_1^{-1}\boldsymbol{B}_2(\boldsymbol{B}_1 + t\boldsymbol{B}_2)^{-1}\boldsymbol{B}_2\boldsymbol{B}_1^{-1}.$$

Let $\boldsymbol{U} \in \mathbb{R}^{n \times n}$ with $U_{ij} = [\boldsymbol{f}_0]_i[\boldsymbol{f}_0]_j\mathbb{1}\{i \neq j\}$. We then have

$$\left|\sum_{1 \leq i \neq j \leq n}[\boldsymbol{B}_1^{-1}\boldsymbol{B}_2\boldsymbol{B}_1^{-1}]_{ij}[\boldsymbol{f}_0]_i[\boldsymbol{f}_0]_j\right|$$

$$= |\langle\boldsymbol{B}_1^{-1}\boldsymbol{B}_2\boldsymbol{B}_1^{-1}, \boldsymbol{U}\rangle|$$

$$\leq \frac{1}{t}|\langle\boldsymbol{B}_1^{-1}, \boldsymbol{U}\rangle| - \frac{1}{t}|\langle(\boldsymbol{B}_1 + t\boldsymbol{B}_2)^{-1}, \boldsymbol{U}\rangle| + t\|\boldsymbol{B}_1^{-1}\|_{\mathrm{op}}^2\|\boldsymbol{B}_2\|_{\mathrm{op}}^2\|(\boldsymbol{B}_1 + t\boldsymbol{B}_2)^{-1}\|_{\mathrm{op}}\|\boldsymbol{U}\|_{\mathrm{tr}}. \tag{15}$$

For the first two terms, Lemma D.3 implies that

$$\frac{1}{k}|\langle\boldsymbol{B}_1^{-1}, \boldsymbol{U}\rangle| \xrightarrow{\text{a.s.}} 0, \quad \text{and} \quad \frac{1}{k}|\langle(\boldsymbol{B}_1 + t\boldsymbol{B}_2)^{-1}, \boldsymbol{U}\rangle| \xrightarrow{\text{a.s.}} 0.$$

For the last term, note that $\|\boldsymbol{B}_2\|_{\mathrm{op}} \leq \|\boldsymbol{A}\|_{\mathrm{op}}^2\|\widehat{\boldsymbol{\Sigma}}_0\|_{\mathrm{op}}$, where $\|\boldsymbol{A}\|_{\mathrm{op}}$ is almost surely bounded as assumed, and $\|\widehat{\boldsymbol{\Sigma}}_0\|_{\mathrm{op}} \leq r_{\max}(1 + \sqrt{\psi^2/\phi})^2$ almost surely as $k, n, p \to \infty$ and $p/n \to \phi \in (0, \infty)$, $p/k \to \psi \in [\phi, \infty]$ (see, e.g., [45]). Also, $\|\boldsymbol{U}\|_* \leq 2\|\boldsymbol{f}_{\mathrm{NL}}\|_2^2 \xrightarrow{\text{a.s.}} 2\|f_{\mathrm{NL}}\|_{L_2}^2 < \infty$ from the strong law of large numbers, Lemma D.5, and $\|\boldsymbol{B}_1\|_{\mathrm{op}} \leq \lambda^{-1}$. Thus, the last term is almost surely bounded. It then follows that $k^{-1}|\langle\boldsymbol{B}_1^{-1}\boldsymbol{B}_2\boldsymbol{B}_1^{-1}, \boldsymbol{U}\rangle| \xrightarrow{\text{a.s.}} 0$. Therefore,

$$\left|\frac{1}{k}\boldsymbol{f}_0^\top\boldsymbol{B}_1^{-1}\boldsymbol{B}_2\boldsymbol{B}_1^{-1}\boldsymbol{f}_0 - \frac{1}{k}\sum_{i=1}^n[\boldsymbol{B}_1^{-1}\boldsymbol{B}_2\boldsymbol{B}_1^{-1}]_{ii}[\boldsymbol{f}_0]_i^2\right| \xrightarrow{\text{a.s.}} 0.$$

**Part (3) Trace concentration.** From the results in [56], it holds that

$$\max_{1 \leq i \leq n}\left|[\boldsymbol{B}_1^{-1}\boldsymbol{B}_2\boldsymbol{B}_1^{-1}]_{ii} - \frac{1}{n}\mathrm{tr}[\boldsymbol{B}_1^{-1}\boldsymbol{B}_2\boldsymbol{B}_1^{-1}]\right| \xrightarrow{\text{a.s.}} 0.$$

Further, $n^{-1}\sum_{i=1}^n\boldsymbol{f}_i^2 \xrightarrow{\text{a.s.}} \|f_{\mathrm{NL}}\|_{L^2}^2$ by strong law of large number and Lemma D.5. Therefore, we have

$$\frac{1}{k}|\boldsymbol{f}^\top\boldsymbol{B}_1^{-1}\boldsymbol{B}_2\boldsymbol{B}_1^{-1}\boldsymbol{f} - \mathrm{tr}[\boldsymbol{B}_1^{-1}\boldsymbol{B}_2\boldsymbol{B}_1^{-1}]\|f_{\mathrm{NL}}\|_{L^2}^2| \xrightarrow{\text{a.s.}} 0. \tag{16}$$

Combining (12), (13), and (16) yields that

$$\left| \boldsymbol{f}^\top (\boldsymbol{A}_k^\lambda)^\top \boldsymbol{A}^\top \boldsymbol{A} \boldsymbol{A}_k^\lambda \boldsymbol{f} - \mathrm{tr}[(\boldsymbol{A}_k^\lambda)^\top \boldsymbol{A}^\top \boldsymbol{A} \boldsymbol{A}_k^\lambda] \cdot \|f_{\mathrm{NL}}\|_{L^2}^2 \right| \xrightarrow{\text{a.s.}} 0.$$

By the trace property $\mathrm{tr}[(\boldsymbol{A}_k^\lambda)^\top \boldsymbol{A}^\top \boldsymbol{A} \boldsymbol{A}_k^\lambda] = \mathrm{tr}[\boldsymbol{A}_k^\lambda (\boldsymbol{A}_k^\lambda)^\top \boldsymbol{A}^\top \boldsymbol{A}]$, it remains to derive the deterministic equivalents of $\boldsymbol{A}_k^\lambda (\boldsymbol{A}_k^\lambda)^\top \boldsymbol{A}^\top \boldsymbol{A}$. Note that

$$\boldsymbol{A}_k^\lambda (\boldsymbol{A}_k^\lambda)^\top \boldsymbol{A}^\top \boldsymbol{A} = \frac{|I_1 \cap I_2|}{k^2} \mathbb{E}_{I_1, I_2 \sim \mathcal{I}_k} [\boldsymbol{M}_1 \widehat{\boldsymbol{\Sigma}}_{1 \cap 2} \boldsymbol{M}_2] \boldsymbol{A}^\top \boldsymbol{A}$$

where $\widehat{\boldsymbol{\Sigma}}_{1 \cap 2} = \boldsymbol{X}^\top \boldsymbol{L}_{I_1 \cap I_2} \boldsymbol{X} / |I_1 \cap I_2|$ and $\boldsymbol{M}_j = (\boldsymbol{X}^\top \boldsymbol{L}_{I_j} \boldsymbol{X} / k + \lambda \boldsymbol{I}_p)^{-1}$ for $j = 1, 2$. The above quantity is well-defined almost surely because $|I_1 \cap I_2|$ converges to some positive quantity almost surely. Next, we analyze the trace term for the two cases.

From Lemma D.1 (3) we know that when $\boldsymbol{A}$ is independent to $(\boldsymbol{X}, \boldsymbol{y})$, it follows that

$$\mathbb{E}_{I_1, I_2 \sim \mathcal{I}_k} [\boldsymbol{M}_1 \widehat{\boldsymbol{\Sigma}}_{1 \cap 2} \boldsymbol{M}_2] \boldsymbol{A}^\top \boldsymbol{A} \simeq \phi^{-1} \widetilde{v}_v(-\lambda; \phi, \psi)(v_p(-\lambda; \psi) \boldsymbol{\Sigma} + \boldsymbol{I}_p)^{-2} \boldsymbol{\Sigma} \boldsymbol{A}^\top \boldsymbol{A}.$$

We now have

$$\boldsymbol{f}_{\mathrm{NL}}{}^\top (\boldsymbol{A}_k^\lambda)^\top \boldsymbol{A}^\top \boldsymbol{A} \boldsymbol{A}_k^\lambda \boldsymbol{f}_{\mathrm{NL}} \xuparrow[]{\text{a.s.}} \|f_{\mathrm{NL}}\|_{L^2}^2 \frac{p}{k} \cdot \frac{|I_1 \cap I_2|}{k} \cdot \frac{1}{p} \mathrm{tr}[\mathbb{E}_{I_1, I_2 \sim \mathcal{I}_k} [\boldsymbol{M}_1 \widehat{\boldsymbol{\Sigma}}_{1 \cap 2} \boldsymbol{M}_2] \boldsymbol{A}^\top \boldsymbol{A}]$$

$$\xuparrow[]{\text{a.s.}} \|f_{\mathrm{NL}}\|_{L^2}^2 \psi \cdot \frac{\phi}{\psi} \cdot \frac{1}{\phi} \widetilde{v}_p(-\lambda; \phi, \psi, \boldsymbol{A}^\top \boldsymbol{A})$$

$$= \|f_{\mathrm{NL}}\|_{L^2}^2 \widetilde{v}_p(-\lambda; \phi, \psi, \boldsymbol{A}^\top \boldsymbol{A}),$$

where the second convergence is from Lemma S.8.3 of [12] and the trace property in Lemma E.3 (4).

(2) $\underline{\boldsymbol{A} = n^{-1/2} \boldsymbol{X}.}$

Instead of working on (13), we use the following decomposition:

$$\boldsymbol{M}_1 \widehat{\boldsymbol{\Sigma}} \boldsymbol{M}_2 = \boldsymbol{M}_1 \widehat{\boldsymbol{\Sigma}}_1 \boldsymbol{M}_2 + \widehat{\boldsymbol{\Sigma}}_2 \boldsymbol{M}_2 - \boldsymbol{M}_1 \widehat{\boldsymbol{\Sigma}}_{1 \cap 2} \boldsymbol{M}_2 + \boldsymbol{M}_1 \widehat{\boldsymbol{\Sigma}}_{(1 \cup 2)^c} \boldsymbol{M}_2$$

$$= \sum_{j=1}^{2} \boldsymbol{M}_j - 2\lambda \boldsymbol{M}_1 \boldsymbol{M}_2 - \boldsymbol{M}_1 \widehat{\boldsymbol{\Sigma}}_{1 \cap 2} \boldsymbol{M}_2 + \boldsymbol{M}_1 \widehat{\boldsymbol{\Sigma}}_{(1 \cup 2)^c} \boldsymbol{M}_2$$

Then, repeating Part (2) and (3) as above, it suffices to derive the deterministic equivalents of $(\boldsymbol{A}_k^\lambda)^\top \widehat{\boldsymbol{\Sigma}} \boldsymbol{A}_k^\lambda$. We decompose this term into:

$$(\boldsymbol{A}_k^\lambda)^\top \boldsymbol{A}^\top \boldsymbol{A} \boldsymbol{A}_k^\lambda = \frac{1}{n}(\boldsymbol{X} \boldsymbol{A}_k^\lambda - \boldsymbol{I}_n)^\top (\boldsymbol{X} \boldsymbol{A}_k^\lambda - \boldsymbol{I}_n) + \frac{1}{n}(\boldsymbol{X} \boldsymbol{A}_k^\lambda + \boldsymbol{X}^\top (\boldsymbol{A}_k^\lambda)^\top) - \frac{1}{n} \boldsymbol{I}_n. \quad (17)$$

For the last two terms of the right hand side, following the similar argument as in Part (1), it holds that

$$\frac{2}{n} \mathrm{tr}[\boldsymbol{X} \boldsymbol{A}_k^\lambda] - 2\phi \int \frac{v_p(-\lambda; \psi) r}{1 + v_p(-\lambda; \psi) r} \, \mathrm{d} H_p(r) \xrightarrow{\text{a.s.}} 0, \quad \text{and} \quad \frac{1}{n} \mathrm{tr}[\boldsymbol{I}_n] = 1. \quad (18)$$

For the first term of the right hand side, notice that it is the variance term of the mean squared error computed on all samples, which is the variance term of the numerator of the generalized cross-validation (GCV) estimator in [13]. It has been shown in Proposition 3.6 of [13] that in the full ensemble, the GCV estimator is consistent with the prediction risk, which is also true for the variance term:

$$\frac{1}{n} \mathrm{tr}[(\boldsymbol{X} \boldsymbol{A}_k^\lambda - \boldsymbol{I}_n)^\top (\boldsymbol{X} \boldsymbol{A}_k^\lambda - \boldsymbol{I}_n)] \quad (19)$$

$$\xrightarrow{\text{a.s.}} \left( 1 - \phi \int \frac{v_p(-\lambda; \psi) r}{1 + v_p(-\lambda; \psi) r} \, \mathrm{d} H_p(r) \right)^2 \cdot \sigma^2 (1 + \widetilde{v}_p(-\lambda; \phi, \psi, \boldsymbol{\Sigma})). \quad (20)$$

Finally, combining (14)-(16) with the above finishes the proof for the first convergence result of $\boldsymbol{A} = n^{-1/2} \boldsymbol{X}$.

Analogously, from (20), we also have

$$\frac{1}{n}\boldsymbol{f}_{\text{NL}}^{\top}(\boldsymbol{X}\boldsymbol{A}_k^{\lambda}-\boldsymbol{I}_n)^{\top}(\boldsymbol{X}\boldsymbol{A}_k^{\lambda}-\boldsymbol{I}_n)\boldsymbol{f}_{\text{NL}}$$

$$\xrightarrow{\text{a.s.}}\|f_{\text{NL}}\|_{L^2}^2\left(1-\phi\int\frac{v_p(-\lambda;\psi)r}{1+v_p(-\lambda;\psi)r}\,\mathrm{d}H_p(r)\right)^2\cdot\sigma^2(1+\widetilde{v}_p(-\lambda;\phi,\psi,\boldsymbol{\Sigma})),$$

which finishes the proof for the second convergence result of $\boldsymbol{A}=n^{-1/2}\boldsymbol{X}$.

$\square$

**Lemma A.3** (Cross terms of generalized risk)**.** *Suppose the same assumptions in Theorem 1 hold with $\lambda > 0$. Let $\boldsymbol{A}_k^{\lambda} = \mathbb{E}_{I\sim\mathcal{I}_k}[(\boldsymbol{X}^{\top}\boldsymbol{L}_I\boldsymbol{X}/k + \lambda\boldsymbol{I}_p)^{-1}\boldsymbol{X}^{\top}\boldsymbol{L}_I/k]$ and $\boldsymbol{B}_k^{\lambda} = \boldsymbol{I}_p - \boldsymbol{A}_k^{\lambda}\boldsymbol{X} = \mathbb{E}_{I\sim\mathcal{I}_k}[\lambda(\boldsymbol{X}^{\top}\boldsymbol{L}_I\boldsymbol{X}/k+\lambda\boldsymbol{I}_p)^{-1}]$. For any $\boldsymbol{A}$ with $\limsup\|\boldsymbol{A}\|_{\text{op}}$ bounded almost surely, it holds that*

$$\boldsymbol{f}_{\text{NL}}^{\top}(\boldsymbol{A}_k^{\lambda})^{\top}\boldsymbol{A}^{\top}\boldsymbol{A}\boldsymbol{B}_k^{\lambda}\boldsymbol{\beta}_0 \xrightarrow{\text{a.s.}} 0.$$

*Proof.* Note that

$$\begin{aligned}
n\|(\boldsymbol{A}_k^{\lambda})^{\top}\boldsymbol{A}^{\top}\boldsymbol{A}\boldsymbol{B}_k^{\lambda}\boldsymbol{\beta}_0\|_2^2 &\le \frac{n}{k}\|\boldsymbol{A}_k^{\lambda}\|_{\text{op}}^2\|\boldsymbol{A}\|_{\text{op}}^4\|\boldsymbol{B}_k^{\lambda}\|_{\text{op}}^2\|\boldsymbol{\beta}_0\|_2^2 \\
&\le \frac{n}{k}\mathbb{E}_{I\sim\mathcal{I}_k}[\|\widehat{\boldsymbol{\Sigma}}_I\|_{\text{op}}^{\frac{1}{2}}\|\lambda(\boldsymbol{X}^{\top}\boldsymbol{L}_I\boldsymbol{X}/k+\lambda\boldsymbol{I}_p)^{-1}\|_{\text{op}}]^2\cdot\|\boldsymbol{B}_k^{\lambda}\|_{\text{op}}^2\|\boldsymbol{A}\|_{\text{op}}^4\|\boldsymbol{\beta}_0\|_2^2 \\
&\le \frac{n}{k}\mathbb{E}_{I\sim\mathcal{I}_k}[\|\widehat{\boldsymbol{\Sigma}}_I\|_{\text{op}}^{\frac{1}{2}}]^2\cdot\|\boldsymbol{A}\|_{\text{op}}^4\|\boldsymbol{\beta}_0\|_2^2.
\end{aligned}$$

Let $s_j^2$ be the singular value of $\widehat{\boldsymbol{\Sigma}}_j$. From the results in [45], we have $\limsup\|\widehat{\boldsymbol{\Sigma}}_I\|_{\text{op}} \le \limsup\max_{1\le i\le p}s_i^2 \le r_{\max}(1+\sqrt{\psi})^2$ almost surely as $k,p \to \infty$ and $p/k \to \psi \in (0,\infty]$. On the other hand, $n/k \xrightarrow{\text{a.s.}} \psi/\phi$ and $\|\boldsymbol{A}\|_2$ is uniformly bounded as assumed. Thus, we have $\limsup n\|(\boldsymbol{A}_k^{\lambda})^{\top}\boldsymbol{A}^{\top}\boldsymbol{A}\boldsymbol{B}_k^{\lambda}\boldsymbol{\beta}_0\|_2^2$ is bounded almost surely.

Similar to the proof of Theorem 1, we decompose $\boldsymbol{f}_{\text{NL}}$ as $\boldsymbol{f}_{\text{NL}} = \boldsymbol{L}_{I_1\cap I_2}\boldsymbol{f}_{\text{NL}} + \boldsymbol{L}_{I_1\setminus I_2}\boldsymbol{f}_{\text{NL}} + \boldsymbol{L}_{I_2\setminus I_1}\boldsymbol{f}_{\text{NL}} =: \boldsymbol{f}_0 + \boldsymbol{f}_1 + \boldsymbol{f}_2$. We then have

$$\begin{aligned}
&\boldsymbol{f}_{\text{NL}}^{\top}\frac{\boldsymbol{L}_{I_1\cap I_2}\boldsymbol{X}}{k}\boldsymbol{M}_1\boldsymbol{A}^{\top}\boldsymbol{A}\boldsymbol{M}_2\frac{\boldsymbol{X}^{\top}\boldsymbol{L}_{I_1\cap I_2}\boldsymbol{X}}{k}\boldsymbol{\beta}_0 \\
&\xrightarrow{\text{a.s.}} \boldsymbol{f}_0^{\top}\frac{\boldsymbol{L}_{I_1\cap I_2}\boldsymbol{X}}{k}\boldsymbol{M}_1\boldsymbol{A}^{\top}\boldsymbol{A}\boldsymbol{M}_2\frac{\boldsymbol{X}^{\top}\boldsymbol{L}_{I_1\cap I_2}\boldsymbol{X}}{k}\boldsymbol{\beta}_0 \\
&\xrightarrow{\text{a.s.}} \frac{i_0}{k}\boldsymbol{f}_0^{\top}\boldsymbol{L}_{I_1\cap I_2}\frac{k^2}{i_0^2}\frac{\boldsymbol{X}_0'}{k}\boldsymbol{M}'\boldsymbol{A}'\boldsymbol{M}'\widehat{\boldsymbol{\Sigma}}_0'\boldsymbol{\beta}_0 \\
&= \frac{i_0}{k}\boldsymbol{f}_0^{\top}\boldsymbol{L}_{I_1\cap I_2}\frac{k^2}{i_0^2}\frac{\boldsymbol{X}_0'}{k}\boldsymbol{M}'\boldsymbol{A}'\boldsymbol{\beta}_0 - \lambda\frac{i_0}{k}\boldsymbol{f}_0^{\top}\boldsymbol{L}_{I_1\cap I_2}\frac{k^2}{i_0^2}\frac{\boldsymbol{X}_0'}{k}\boldsymbol{M}'\boldsymbol{A}'\boldsymbol{M}'\boldsymbol{\beta}_0,
\end{aligned}$$

where the first convergence is from Lemma S.8.5 of [51] and the second convergence is from (13), with $\boldsymbol{X}_0' = \boldsymbol{X}_0(\boldsymbol{I}_p+\boldsymbol{C})^{-1/2}$, $\boldsymbol{A}' = (\boldsymbol{I}_p+\boldsymbol{C})^{-1/2}\boldsymbol{A}^{\top}\boldsymbol{A}(\boldsymbol{I}_p+\boldsymbol{C})^{-1/2}$, and $\boldsymbol{M}' = (\widehat{\boldsymbol{\Sigma}}_0'+\lambda\boldsymbol{I}_p)^{-1}$ and $\widehat{\boldsymbol{\Sigma}}_0' = (\boldsymbol{I}_p+\boldsymbol{C})^{-1/2}\widehat{\boldsymbol{\Sigma}}_0(\boldsymbol{I}_p+\boldsymbol{C})^{-1/2}$. From Lemma D.2, the first term in the above display vanishes. Analogous to (15), the second term also vanishes by splitting and applying Lemma D.2. Thus, we have for $I_1, I_2 \sim \mathcal{I}_k$,

$$\boldsymbol{f}_{\text{NL}}^{\top}\frac{\boldsymbol{L}_{I_1\cap I_2}\boldsymbol{X}}{k}\boldsymbol{M}_1\boldsymbol{A}^{\top}\boldsymbol{A}\boldsymbol{M}_2\frac{\boldsymbol{X}^{\top}\boldsymbol{L}_{I_1\cap I_2}\boldsymbol{X}}{k}\boldsymbol{\beta}_0 \xrightarrow{\text{a.s.}} 0.$$

Finally, by Lemma G.5 (2) of [13], the conclusion follows. $\square$

# B  Proofs of results in Section 4

## B.1  Proof of Theorem 3

We consider two cases.

(1) $\underline{\lambda_1, \lambda_2 > 0.}$

We begin to prove for the full ensemble when $M = \infty$. For $j = 1, 2$ and $I_j \sim \mathcal{I}_{k_j}$, define $\widehat{\boldsymbol{\Sigma}}_j = \boldsymbol{X}^\top \boldsymbol{L}_{I_j} \boldsymbol{X}/n$ and $\boldsymbol{M}_j = (\boldsymbol{X}^\top \boldsymbol{L}_{I_j} \boldsymbol{X}/k + \lambda_j \boldsymbol{I}_p)^+$. Recall that $\boldsymbol{y} = \boldsymbol{X}\boldsymbol{\beta}_0 + \boldsymbol{f}_{\mathrm{NL}}$ and

$$\widehat{\boldsymbol{\beta}}_{k_j,\infty}^{\lambda_j} = \mathbb{E}_{I_j \sim \mathcal{I}_{k_j}} \left[ \left( \frac{1}{k_j} \boldsymbol{X}^\top \boldsymbol{L}_{I_j} \boldsymbol{X} + \lambda_j \boldsymbol{I}_p \right)^{-1} \frac{\boldsymbol{X}^\top \boldsymbol{L}_{I_j} \boldsymbol{y}}{k_j} \right]$$

$$= \mathbb{E}_{I_j \sim \mathcal{I}_{k_j}}[\boldsymbol{M}_j \widehat{\boldsymbol{\Sigma}}_j] \boldsymbol{\beta}_0 + \mathbb{E}_{I_j \sim \mathcal{I}_{k_j}}[\boldsymbol{M}_j \boldsymbol{X}^\top \boldsymbol{L}_{I_j}/k_j] \boldsymbol{f}_{\mathrm{NL}}.$$

Then, for $\boldsymbol{a} \in \mathbb{R}^p$ with bounded $l_2$ norm, we have

$$\boldsymbol{a}^\top (\widehat{\boldsymbol{\beta}}_{k_1,\infty}^{\lambda_1} - \widehat{\boldsymbol{\beta}}_{k_2,\infty}^{\lambda_2}) = \underbrace{\boldsymbol{a}^\top (\mathbb{E}_{I_1 \sim \mathcal{I}_{k_1}}[\boldsymbol{M}_1 \widehat{\boldsymbol{\Sigma}}_1] - \mathbb{E}_{I_2 \sim \mathcal{I}_{k_2}}[\boldsymbol{M}_2 \widehat{\boldsymbol{\Sigma}}_2]) \boldsymbol{\beta}_0}_{T_1}$$

$$+ \underbrace{\boldsymbol{a}^\top (\mathbb{E}_{I_1 \sim \mathcal{I}_{k_1}}[\boldsymbol{M}_1 \boldsymbol{X}^\top \boldsymbol{L}_{I_1}]/k_1 - \mathbb{E}_{I_2 \sim \mathcal{I}_{k_2}}[\boldsymbol{M}_2 \boldsymbol{X}^\top \boldsymbol{L}_I]/k_2) \boldsymbol{f}_{\mathrm{NL}}}_{T_2}. \tag{21}$$

Next, we analyze the three terms separately.

For the first term, from Lemma D.1 (1) we have for $\lambda_j > 0$,

$$\mathbb{E}_{I_j \sim \mathcal{I}_{k_j}}[\boldsymbol{M}_j \widehat{\boldsymbol{\Sigma}}_j] = \boldsymbol{I}_p - \mathbb{E}_{I_j \sim \mathcal{I}_{k_j}}[\lambda_j \boldsymbol{M}_j] \simeq \boldsymbol{I}_p - (v(-\lambda_j; \psi_j) \boldsymbol{\Sigma} + \boldsymbol{I}_p)^{-1},$$

where $v(-\lambda_j; \psi_j)$ is as defined in (40). By Lemma C.1 we have that for $\overline{\psi} \in [\phi, \infty]$ there exists a segment $\mathcal{P}$ such that for all $(\lambda_1, \psi_1), (\lambda_2, \psi_2) \in \mathcal{P}$, it holds that $v(-\lambda_1; \psi_1) = v(-\lambda_1; \psi_2)$ as $p/k_1 \to \psi_1$ and $p/k_2 \to \psi_2$. By the definition of deterministic equivalents (Definition E.1), it follows that

$$T_1 = \mathrm{tr}[\boldsymbol{\beta}_0 \boldsymbol{a}^\top \mathbb{E}_{I_1 \sim \mathcal{I}_{k_1}}[\boldsymbol{M}_1 \widehat{\boldsymbol{\Sigma}}_1]] - \mathrm{tr}[\boldsymbol{\beta}_0 \boldsymbol{a}^\top \mathbb{E}_{I_2 \sim \mathcal{I}_{k_2}}[\boldsymbol{M}_2 \widehat{\boldsymbol{\Sigma}}_2]]$$

$$\xrightarrow{\text{a.s.}} \lim_{p \to \infty} \mathrm{tr}[\boldsymbol{\beta}_0 \boldsymbol{a}^\top (\boldsymbol{I}_p - (v(-\lambda_1; \psi_1) \boldsymbol{\Sigma} + \boldsymbol{I}_p)^{-1})] - \mathrm{tr}[\boldsymbol{\beta}_0 \boldsymbol{a}^\top (\boldsymbol{I}_p - (v(-\lambda_2; \psi_2) \boldsymbol{\Sigma} + \boldsymbol{I}_p)^{-1})]$$

$$= 0.$$

For the second term, notice that by Lemma D.2,

$$\frac{1}{k_j} \boldsymbol{a}^\top \boldsymbol{M}_j \boldsymbol{X}^\top \boldsymbol{L}_{I_{k_j}} \boldsymbol{f}_{\mathrm{NL}} = \boldsymbol{a}^\top \frac{\boldsymbol{Z} \boldsymbol{\Sigma}^{\frac{1}{2}}}{k_j} \left( \frac{\boldsymbol{Z} \boldsymbol{\Sigma} \boldsymbol{Z}^\top}{k_j} + \lambda_j \boldsymbol{I}_p \right)^{-1} \boldsymbol{f}_{\mathrm{NL}} \xrightarrow{\text{a.s.}} 0.$$

From Lemma G.5 (2) of [13], it follows that

$$\boldsymbol{a}^\top \mathbb{E}_{I_j \sim \mathcal{I}_{k_j}}[\boldsymbol{M}_j \boldsymbol{X}^\top \boldsymbol{L}_{I_j}]/k_j \boldsymbol{f}_{\mathrm{NL}} \xrightarrow{\text{a.s.}} 0, \qquad j = 1, 2,$$

and $T_2 \xrightarrow{\text{a.s.}} 0$.

Combining the above results and applying triangle inequality on (21) yields that

$$|\boldsymbol{a}^\top (\widehat{\boldsymbol{\beta}}_{k_1,\infty}^{\lambda_1} - \widehat{\boldsymbol{\beta}}_{k_2,\infty}^{\lambda_2})| \leq |T_1| + |T_2| \xrightarrow{\text{a.s.}} 0,$$

which completes the proof when $M = \infty$. When $M \in \mathbb{N}$, replacing the above expectation by the average over $M$ simple random samples $I_1, I_2 \sim \mathcal{I}_k$ completes the proof.

(2) $\underline{\lambda_1 \lambda_2 = 0.}$

When $\lambda_1 = \lambda_2 = 0$, it is trivially true as $k_1 = k_2$. Otherwise, without loss of generality, we assume $\lambda_1 = 0$ and $\lambda_2 > 0$. We use the same decomposition (21) and first analyze $T_1$. From part one we have that for $\lambda > 0$, $P_{n,\lambda} - Q_{n,\lambda} \xrightarrow{\text{a.s.}} 0$ where

$$P_{n,\lambda} := \mathrm{tr}[\boldsymbol{\beta}_0 \boldsymbol{a}^\top \mathbb{E}_{I_1 \sim \mathcal{I}_{k_1}}[(\boldsymbol{X}^\top \boldsymbol{L}_{I_1} \boldsymbol{X}/k_1 + \lambda \boldsymbol{I}_p)^+ \widehat{\boldsymbol{\Sigma}}_1]],$$

$$Q_{n,\lambda} := \mathrm{tr}[\boldsymbol{\beta}_0 \boldsymbol{a}^\top (\boldsymbol{I}_p - (v(-\lambda; \psi_1) \boldsymbol{\Sigma} + \boldsymbol{I}_p)^{-1})].$$

We next show that

$$\lim_{n \to \infty} \lim_{\lambda \to 0+} P_{n,\lambda} = \lim_{\lambda \to 0+} \lim_{n \to \infty} P_{n,\lambda} = 0, \tag{22}$$

by proving that the function $P_{n,\lambda}$ is equicontinuous family in $\lambda$ over $\Lambda = [0, \lambda_{\max}]$ for any $\lambda_{\max} \in (0, \infty)$ fixed. Note that for all $\lambda \in \Lambda$,

$$|P_{n,\lambda}| \leq \mathbb{E}_{I_1 \sim \mathcal{I}_{k_1}} [\|(\boldsymbol{X}^\top \boldsymbol{L}_{I_1} \boldsymbol{X}/k_1 + \lambda \boldsymbol{I}_p)^+ \widehat{\boldsymbol{\Sigma}}_1\|_{\mathrm{op}}] \|\boldsymbol{\beta}_0 \boldsymbol{a}^\top\|_{\mathrm{tr}} \leq \|\boldsymbol{\beta}_0\|_2^2 \|\boldsymbol{a}\|_2^2$$

and its derivative

$$\left| \frac{\partial}{\partial \lambda} P_{n,\lambda} \right| = \left| \mathbb{E}_{I_1 \sim \mathcal{I}_{k_1}} [\mathrm{tr}[(\boldsymbol{X}^\top \boldsymbol{L}_{I_1} \boldsymbol{X}/k_1 + \lambda \boldsymbol{I}_p)^{-2} \widehat{\boldsymbol{\Sigma}}_1 \boldsymbol{\beta}_0 \boldsymbol{a}^\top]] \right|$$

$$\leq \mathbb{E}_{I_1 \sim \mathcal{I}_{k_1}} [\|(\boldsymbol{X}^\top \boldsymbol{L}_{I_1} \boldsymbol{X}/k_1 + \lambda \boldsymbol{I}_p)^{-2} \widehat{\boldsymbol{\Sigma}}_1\|_{\mathrm{op}}] \|\boldsymbol{\beta}_0 \boldsymbol{a}^\top\|_{\mathrm{tr}}$$

$$\leq \|\boldsymbol{\beta}_0\|_2^2 \|\boldsymbol{a}\|_2^2$$

are uniformly bounded in $\lambda$ almost surely. In the above inequalities, the operator norm is bounded because $\|(\boldsymbol{X}^\top \boldsymbol{L}_{I_1} \boldsymbol{X}/k_1 + \lambda \boldsymbol{I}_p)^{-1} \widehat{\boldsymbol{\Sigma}}_1\|_{\mathrm{op}} \leq s_i/(s_i + \lambda) \leq 1$ and $\|(\boldsymbol{X}^\top \boldsymbol{L}_{I_1} \boldsymbol{X}/k_1 + \lambda \boldsymbol{I}_p)^{-2} \widehat{\boldsymbol{\Sigma}}_1\|_{\mathrm{op}} \leq s_i/(s_i + \lambda)^2 \leq 1/(s_i + \lambda) \leq 1$, by noting that $\limsup \|\widehat{\boldsymbol{\Sigma}}_j\|_{\mathrm{op}} \leq \limsup \max_{1 \leq i \leq p} s_i^2 \leq r_{\max}(1 + \sqrt{\psi_1})^2$ and $\liminf \|\widehat{\boldsymbol{\Sigma}}_j\|_{\mathrm{op}} \geq \liminf \min_{1 \leq i \leq p} s_i^2 \geq r_{\min}(1 - \sqrt{\psi_1})^2$ almost surely as $k_1, p \to \infty$ and $p/k_1 \to \psi_1 \in (0, \infty) \setminus \{1\}$ [45]. On the other hand, since $Q_{n,\lambda}$ is a continuous function of $v(-\lambda; \psi_1)$, we have

$$|Q_{n,\lambda}| \leq \|v(-\lambda_1; \psi_1) \boldsymbol{\Sigma}(v(-\lambda_1; \psi_1)\boldsymbol{\Sigma} + \boldsymbol{I}_p)^{-1}\|_{\mathrm{op}} \|\boldsymbol{\beta}_0 \boldsymbol{a}^\top\|_{\mathrm{tr}} \leq \|\boldsymbol{\beta}_0\|_2^2 \|\boldsymbol{a}\|_2^2$$

$$\left| \frac{\partial}{\partial \lambda} Q_{n,\lambda} \right| = \left| \mathrm{tr}[\boldsymbol{\beta}_0 \boldsymbol{a}^\top (v(-\lambda; \psi_1)\boldsymbol{\Sigma} + \boldsymbol{I}_p)^{-2} \boldsymbol{\Sigma}] \frac{\partial v(-\lambda; \psi_1)}{\partial \lambda} \right|$$

$$\leq \|\boldsymbol{\beta}_0 \boldsymbol{a}^\top\|_{\mathrm{op}} \left| \frac{\partial v(-\lambda; \psi_1)}{\partial \lambda} \right| \int \frac{r}{(1 + v(-\lambda; \psi_1)r)^2} \, \mathrm{d}H_p(r).$$

When $\psi_1 > 1$, $\partial v(-\lambda; \psi_1)/\partial \lambda$ is bounded over $\Lambda$ [13, Lemma E.10 and Lemma E.11] and thus $|\partial Q_{n,\lambda}/\partial \lambda|$ is also bounded almost surely. When $\psi_1 < 1$, from Lemma E.12 of [13] we have

$$\left| \frac{\partial v(-\lambda; \psi_1)}{\partial \lambda} \right| \left| \int \frac{r}{(1 + v(-\lambda; \psi_1)r)^2} \, \mathrm{d}H_p(r) \right| = -\frac{\partial v(-\lambda; \psi_1)}{\partial \lambda} \int \frac{r}{(1 + v(-\lambda; \psi_1)r)^2} \, \mathrm{d}H_p(r)$$

$$= \frac{\displaystyle\int \frac{r}{(1 + v(-\lambda; \psi_1)r)^2} \, \mathrm{d}H_p(r)}{\displaystyle\frac{1}{v(-\lambda; \psi_1)^2} - \psi_1 \int \frac{r^2}{(1 + v(-\lambda; \psi_1)r)^2} \, \mathrm{d}H_p(r)}$$

$$\leq \frac{1}{1 - \psi_1},$$

since $v(-\lambda; \psi_1) \leq v(0; \psi_1) = +\infty$. Therefore, $|\partial Q_{n,\lambda}/\partial \lambda|$ is uniformly bounded over $\Lambda$ for $\psi_1 \in (0, \infty) \setminus \{1\}$. Thus, by the Moore-Osgood theorem, the convergence is uniform in $\lambda$ and (22) follows. Finally, since $T_2$ is bounded analogously as in Part (1), the equivalence holds when $\lambda_1 = 0$.

## B.2 Proof of Proposition 4

From Lemma B.2, it follows that

$$v(-\lambda; \phi_n) \simeq \frac{1}{n} \mathrm{tr} \left[ \left( \frac{1}{n} \boldsymbol{X}\boldsymbol{X}^\top + \lambda \boldsymbol{I}_p \right)^{-1} \right]. \tag{23}$$

By the continuity of function $\phi \mapsto v(-\lambda; \phi)$ from Lemma F.10 and F.11 of [13], we have $v(-\lambda; \phi_n) \simeq v(-\lambda; \phi)$ as $\phi_n = p/n \to \phi$. From Lemma C.1, there exists $\bar{\lambda}_n$ such that

$$v(0; \psi_n) = v(-\bar{\lambda}_n; \phi_n),$$

as $\psi_n \to \bar{\psi}$ and $\phi_n \to \phi$. Involving (23) on the both sides yields that

$$\frac{1}{n} \mathrm{tr} \left[ \left( \frac{1}{n} \boldsymbol{X}\boldsymbol{X}^\top + \bar{\lambda}_n \boldsymbol{I}_p \right)^{-1} \right] \simeq \frac{1}{k} \mathrm{tr} \left[ \left( \frac{1}{k} \boldsymbol{X}\boldsymbol{L}_{I_1}\boldsymbol{X}^\top \right)^+ \right] \simeq \frac{1}{Mk} \sum_{\ell=1}^{M} \mathrm{tr} \left[ \left( \frac{1}{k} \boldsymbol{X}\boldsymbol{L}_{I_\ell}\boldsymbol{X}^\top \right)^+ \right]$$

where $\{I_1, \ldots, I_M\}$ is a simple random sample from $\mathcal{I}_k$.

## B.3 Proof of Corollary 5

We will use a structural equivalence (much more direct than the first-order equivalence considered in the paper) between generalized ridge predictor and the isotropic ridge predictor to prove the result.

The generalized ridge predictor (9), trained on the subsampled dataset $\mathcal{D}_I$, can be expressed as:

$$\widehat{\beta}_k^{\lambda, G}(\mathcal{D}_I) = \left( \frac{1}{k} X^\top L_I X + \lambda G \right)^{-1} \frac{X^\top L_I y}{k}.$$

Observe that we can equivalently manipulate the generalized ridge estimator into:

$$\widehat{\beta}_k^{\lambda, G}(\mathcal{D}_I) = G^{-1/2} \left( \frac{1}{k} G^{-1/2} X^\top L_I X G^{-1/2} + \lambda I_p \right)^{-1} G^{-1/2} \frac{X^\top L_I y}{k}.$$

Recalling from Assumption 2 that $X = Z\Sigma^{1/2}$, where $Z \in \mathbb{R}^{n \times p}$ contains $z_i^\top$ in the $i$-th row for $i \in [n]$, we obtain

$$\widehat{\beta}_k^{\lambda, G}(\mathcal{D}_I) = G^{-1/2} \left( \frac{1}{k} G^{-1/2} \Sigma^{1/2} Z^\top L_I Z \Sigma^{1/2} G^{-1/2} + \lambda I_p \right)^{-1} \frac{G^{-1/2} \Sigma^{1/2} Z^\top L_I y}{k}. \quad (24)$$

We now define $\Sigma_G = G^{-1/2} \Sigma G^{-1/2}$ and consider a transformed feature matrix $X_G = Z\Sigma_G^{1/2}$. Denote the transformed dataset corresponding to the new feature matrix by $\mathcal{D}^G$ (where we keep the response vector as is). Using (24), we can write relate the generalized ridge estimator fitted on the dataset $\mathcal{D}$ to the standard ridge estimator fitted on the data $\mathcal{D}^G$ (both at the same scalar regularization level $\lambda$) by:

$$\widehat{\beta}_k^{\lambda, G}(\mathcal{D}_I) = G^{-1/2} \widehat{\beta}_k^{\lambda}(\mathcal{D}_I^G).$$

Observe that the transformed dataset $\mathcal{D}^G$ satisfies Assumption 2 as the eigenvalues of $G$ are bounded away from 0 and $\infty$. Further, note that both Theorem 1 and Theorem 3 are invariant to linear transformations of the estimator. Thus, we conclude that the results of both the theorems continue to hold. The equivalence path now use the modified $\Sigma_G$, which in turn changes the spectral distribution $H_p$ to that of $\Sigma_G$, in defining the end points via (4), given by:

$$\widetilde{H}_p = \frac{1}{p} \sum_{i=1}^{p} \mathbb{1}_{\{\widetilde{r}_i \leq r\}},$$

where $\widetilde{r}_i$ for $i \in [p]$ are eigenvalues of $\Sigma_G$. This completes the proof.

## B.4 Technical lemmas

**Lemma B.1** (Relationship between $\widehat{m}$ and $\widehat{v}$)**.** *For $X \in \mathbb{R}^{n \times p}$ and $\phi_n = p/n$, define*

$$\widehat{m}(z; \phi_n) = \frac{1}{p} \operatorname{tr}\left[ \left( \frac{1}{n} X^\top X - zI_p \right)^{-1} \right], \quad and \quad \widehat{v}(z; \phi_n) = \frac{1}{n} \operatorname{tr}\left[ \left( \frac{1}{n} X X^\top - zI_p \right)^{-1} \right].$$

*It holds that*

$$\phi_n z \widehat{m}(z; \phi_n) + \phi_n - 1 = z \widehat{v}(z; \phi_n). \quad (25)$$

*Proof.* Let $r$ be the rank of the matrix $X$. Denote by $s_i$, $i = 1, \ldots, r$, the non-zero eigenvalues of the matrix $X^\top X$. Note that these are the same non-zero eigenvalues of the matrix $X X^\top$. Define function $S$ such that for $z \neq 0$,

$$S(z) = \sum_{i=1}^{r} \frac{1}{s_i - z}.$$

For $z \neq 0$, write out $\widehat{v}$ and $\widehat{m}$ in terms of $S(z)$ as:

$$\widehat{v}(z; \phi_n) = \frac{1}{n} \sum_{i=1}^{r} \frac{1}{s_i - z} - \frac{1}{n} \frac{n-r}{z} = \frac{1}{n} S(z) - \frac{1}{n} \frac{n-r}{z}$$

$$\widehat{m}(z; \phi_n) = \frac{1}{p} \sum_{i=1}^{r} \frac{1}{s_i - z} - \frac{1}{p} \frac{p - r}{z} = \frac{1}{p} S(z) - \frac{1}{p} \frac{p - r}{z}.$$

We now expand the left-hand side of (25):

$$
\begin{aligned}
\phi_n z \widehat{m}(z; \phi_n) + \phi_n - 1 &= \frac{p}{n} z \left( \frac{1}{p} S(z) - \frac{1}{p} \frac{p - r}{z} \right) + \frac{p}{n} - 1 \\
&= \frac{1}{n} z S(z) - \frac{p - r}{n} + \frac{p}{n} - 1 \\
&= \frac{1}{n} z S(z) - \frac{n - r}{n} + \frac{n - r}{n} - \frac{p - r}{n} + \frac{p}{n} - 1 \\
&= z \widehat{v}(z; \phi_n) + \frac{n - p}{n} + \frac{p}{n} - 1 \\
&= z \widehat{v}(z; \phi_n),
\end{aligned}
$$

which finishes the proof. $\qquad\square$

**Lemma B.2** (Equivalence of $\widehat{v}$ and $v$). *Suppose Assumptions 1–2 hold. Then it holds that*

$$v(-\lambda; \phi_n) \simeq \frac{1}{n} \operatorname{tr} \left[ \left( \frac{1}{n} \boldsymbol{X} \boldsymbol{X}^\top + \lambda I_p \right)^{-1} \right].$$

*Proof.* From Lemma B.1, we have

$$z \widehat{v}(z; \phi_n) + (1 - \phi_n) = \phi_n z \widehat{m}(z; \phi_n).$$

Substituting $z = -\lambda$ yields that

$$\lambda \frac{1}{n} \operatorname{tr} \left[ \left( \frac{1}{n} \boldsymbol{X} \boldsymbol{X}^\top + \lambda I_p \right)^{-1} \right] + (\phi_n - 1) = \phi_n \lambda \frac{1}{p} \operatorname{tr} \left[ \left( \frac{1}{n} \boldsymbol{X}^\top \boldsymbol{X} + \lambda I_p \right)^{-1} \right]. \tag{26}$$

From Corollary E.4, we have

$$\frac{1}{v(-\lambda; \phi_n)} = \lambda + \phi \operatorname{tr}[\Sigma (v(-\lambda; \phi_n) \Sigma + I_p)^{-1}] / p,$$

$$\frac{1}{p} \operatorname{tr}[(v(-\lambda; \phi_n) \Sigma + I_p)^{-1}] \simeq \frac{1}{p} \lambda \operatorname{tr} \left[ \left( \frac{1}{n} \boldsymbol{X}^\top \boldsymbol{X} + \lambda I_p \right)^{-1} \right].$$

This implies

$$
\begin{aligned}
1 &= \lambda v(-\lambda; \phi_n) + \phi_n \operatorname{tr}[v(-\lambda; \phi_n) \Sigma (v(-\lambda; \phi_n) \Sigma + I_p)^{-1}] / p \\
&= \lambda v(-\lambda; \phi_n) + \phi_n - \phi_n \operatorname{tr}[(v(-\lambda; \phi_n) \Sigma + I_p)^{-1}]) / p \\
&\simeq \lambda v(-\lambda; \phi_n) + \phi_n - \phi_n \lambda \operatorname{tr} \left[ \left( \frac{1}{n} \boldsymbol{X}^\top \boldsymbol{X} + \lambda I_p \right)^{-1} \right] / p \\
&= 1 + \lambda v(-\lambda; \phi_n) - \frac{1}{n} \lambda \operatorname{tr} \left[ \left( \frac{1}{n} \boldsymbol{X} \boldsymbol{X}^\top + \lambda I_p \right)^{-1} \right],
\end{aligned}
$$

where the last equality follows from (26). This concludes the proof. $\qquad\square$

## C   Proof of results in Section 5

### C.1   Proof of Theorem 6

By Assumptions 3–4, for $\boldsymbol{A} = \Sigma^{1/2}$ and $\boldsymbol{b} = \boldsymbol{0}$, $R_p(\lambda; \phi, \psi)$, as defined in (11), converges to

$$R_p(\lambda; \phi, \psi) \xrightarrow{\text{a.s.}} \rho^2 \widetilde{c}(-\lambda; \phi, \psi) + \sigma^2 \widetilde{v}(-\lambda; \phi, \psi) =: \mathscr{R}(\lambda; \phi, \psi) \tag{27}$$

where the nonnegative constants $\widetilde{c}(-\lambda; \phi, \psi)$ and $\widetilde{v}(-\lambda; \phi, \psi)$ are defined through the following equations:

$$\widetilde{v}(-\lambda; \phi, \psi) = \frac{\phi \int \dfrac{r^2}{(1 + v(-\lambda; \psi)r)^2} \, \mathrm{d}H(r)}{v(-\lambda; \psi)^{-2} - \phi \int \dfrac{r^2}{(1 + v(-\lambda; \psi)r)^2} \, \mathrm{d}H(r)},$$

$$\widetilde{c}(-\lambda; \phi, \psi) = (\widetilde{v}(-\lambda; \phi, \psi) + 1) \int \frac{r}{1 + v(-\lambda; \psi)r} \, \mathrm{d}G(r),$$

$$\frac{1}{v(-\lambda; \psi)} = \lambda + \psi \int \frac{r}{1 + v(-\lambda; \psi)r} \, \mathrm{d}H(r).$$

From the proof of Theorem 1, we also have that $\mathscr{R}(\lambda; \phi, \psi) \simeq R_p(\lambda; \phi, \psi) \simeq R(\widehat{\boldsymbol{\beta}}_{k,\infty}^{\lambda}; \boldsymbol{\Sigma}, \mathbf{0}, \boldsymbol{\beta}_0)$.

Since $\mathscr{R}(0; \phi, \psi)$ is a continuous function of $\phi$ and $v(0; \psi)$ and is increasing in $\phi$ for any fixed $\psi$, it follows that for $0 < \phi_1 \leq \phi_2 < \infty$,

$$\min_{\psi \geq \phi_1} \mathscr{R}(0; \phi_1, \psi) \leq \min_{\psi \geq \phi_2} \mathscr{R}(0; \phi_1, \psi) \leq \min_{\psi \geq \phi_2} \mathscr{R}(0; \phi_2, \psi),$$

where the first inequality follows because $\{\psi : \psi \geq \phi_1\} \supseteq \{\psi : \psi \geq \phi_2\}$, and the second inequality follows because $\mathscr{R}(0; \phi, \psi)$ is increasing in $\phi$ for a fixed $\psi$. Thus, $\min_{\psi \geq \phi} \mathscr{R}(0; \phi, \psi)$ is a continuous and monotonically increasing function in $\phi$.

Finally, note that from Lemma C.1, for any $\lambda$, there exists $\psi$ such that $v(0; \phi, \psi) = v(-\lambda; \phi, \phi)$. This implies that $\mathscr{R}(0; \phi, \psi) = \mathscr{R}(\lambda; \phi, \phi)$. Then we have $\min_{\psi \geq \phi} \mathscr{R}(0; \phi, \psi) \leq \min_{\lambda \geq 0} \mathscr{R}(\lambda; \phi, \phi)$. Conversely, since for any $\psi$, there exists $\lambda$ such that $v(0; \phi, \psi) = v(-\lambda; \phi, \phi)$, we also have $\min_{\psi \geq \phi} \mathscr{R}(0; \phi, \psi) \geq \min_{\lambda \geq 0} \mathscr{R}(\lambda; \phi, \phi)$. Combining the two inequalities yields the conclusion.

## C.2 Technical lemmas

In this section, we gather results on certain analytic properties of the fixed-point solution $v(-\lambda; \phi)$ defined in (40).

**Lemma C.1** (Contour of fixed-point solutions). *As $n, p \to \infty$ such that $p/n \to \phi \in (0, \infty)$, for any $\overline{\psi} \in [\phi, +\infty]$, there exists a unique value $\overline{\lambda} \geq 0$ (or conversely for $\overline{\lambda} \in [0, \infty]$, there exists a unique value $\overline{\psi} \in [\phi \vee 1, \infty]$) such that for all $(\lambda, \psi)$ on the path*

$$\mathcal{P} = \{(1 - \theta) \cdot (\overline{\lambda}, \phi) + \theta \cdot (0, \overline{\psi}) \mid \theta \in [0, 1]\},$$

*it holds that*

$$v_p(-\lambda; \psi) = v(-\overline{\lambda}; \phi) = v(-0; \overline{\psi}).$$

*where $v_p(-\lambda; \psi)$ is as defined in (40).*

*Proof.* Since when $\phi < 1$, $v(0; \psi) = +\infty$ for all $\psi \in [\phi, 1]$, we can restrict ourselves to $\psi \in [\phi \vee 1, +\infty]$. From Lemma E.11 (1) of [13], the function $\psi \mapsto v(0; \psi)$ is strictly decreasing over $\psi \in [\phi \vee 1, \infty]$ with range

$$v(0; \phi \vee 1) = \begin{cases} v(0; \phi), & \phi \in (1, \infty) \\ \lim_{\psi \to 1^+} v(0; \psi) = +\infty, & \phi \in (0, 1] \end{cases}, \qquad v(0; +\infty) := \lim_{\psi \to +\infty} v(0; \psi) = 0.$$

From Lemma E.12 (3) of [13], the function $\lambda \mapsto v(-\lambda; \phi)$ is strictly decreasing over $\lambda \in [0, \infty]$ with range

$$v(0; \phi) = \begin{cases} v(0; \phi), & \phi \in (1, \infty) \\ \lim_{\lambda \to 0^+} v(-\lambda; \phi) = +\infty, & \phi \in (0, 1] \end{cases}, \qquad v(-\infty; \phi) := \lim_{\lambda \to +\infty} v(-\lambda; \phi) = 0.$$

Note that $v(0; \phi \vee 1) = v(0; \phi)$. For $\overline{\psi} \in [\phi \vee 1, \infty]$, by the intermediate value theorem, there exists unique $\overline{\lambda} \in [0, \infty]$ such that $v(-\overline{\lambda}; \phi) = v(0; \overline{\psi})$. Furthermore, when $\overline{\psi} \leq 1$, $\overline{\lambda} = 0$ is also the unique

value such that $v(0; \overline{\psi}) = v(-\overline{\lambda}; \phi)$. Conversely, for $\overline{\lambda} \in [0, \infty]$, there also exists $\overline{\psi} \in [\phi \vee 1, \infty]$ such that $v(-\overline{\lambda}; \phi) = v(0; \overline{\psi})$.

Based on the definition of fixed-point solutions, it follows that

$$\frac{1}{v(-\overline{\lambda}; \phi)} = \overline{\lambda} + \phi \int \frac{r}{1 + v(-\overline{\lambda}; \phi)r} \, \mathrm{d}H_p(r) = \overline{\psi} \int \frac{r}{1 + v(0; \overline{\psi})r} \, \mathrm{d}H_p(r) = \frac{1}{v(0; \overline{\psi})}.$$

Then, for any $(\lambda, \psi) = (1 - \theta)(\overline{\lambda}, \phi) + \theta(0, \overline{\psi})$ on the path $\mathcal{P}$, we have

$$\frac{1}{v(-\overline{\lambda}; \phi)} = (1 - \theta)\frac{1}{v(-\overline{\lambda}; \phi)} + \theta\frac{1}{v(0; \overline{\psi})}$$

$$= (1 - \theta)\overline{\lambda} + (1 - \theta)\phi \int \frac{r}{1 + v(-\overline{\lambda}; \phi)r} \, \mathrm{d}H_p(r) + \theta\overline{\psi} \int \frac{r}{1 + v(0; \overline{\psi})r} \, \mathrm{d}H_p(r)$$

$$= \lambda + \psi \int \frac{r}{1 + v(-\overline{\lambda}; \phi)r} \, \mathrm{d}H_p(r).$$

Because $v_p(-\lambda; \psi)$ is the unique solution to the fixed-point equation:

$$\frac{1}{v_p(-\lambda; \psi)} = \lambda + \psi \int \frac{r}{1 + v_p(-\lambda; \psi)} \, \mathrm{d}H_p(r),$$

it then follows that $v_p(-\lambda; \psi) = v(-\overline{\lambda}; \phi) = v(0; \overline{\psi})$. $\qquad\square$

# D  Asymptotic equivalents, concentration results, and other useful lemmas

## D.1  Full-ensemble resolvents

**Lemma D.1** (Full-ensemble resolvents). *Let $\widehat{\boldsymbol{\Sigma}} = \boldsymbol{X}^\top \boldsymbol{X}/n$, $\widehat{\boldsymbol{\Sigma}}_j = \boldsymbol{X}^\top \boldsymbol{L}_{I_j} \boldsymbol{X}/k$ where $I_j \sim \mathcal{I}_k$. Let $\boldsymbol{\Sigma}_{1\cap2} = \boldsymbol{X}^\top \boldsymbol{L}_{I_1\cap I_2} \boldsymbol{X}/|I_1 \cap I_2|$ and $\boldsymbol{C} \in \mathbb{R}^{p \times p}$ with bounded operator norm almost surely. As $n, p, k \to \infty$, $p/n \to \phi$, $p/k \to \psi$, the following asymptotic equivalences hold:*

*(1) Basic ridge resolvent:*

$$\lambda \mathbb{E}_{I_1 \sim \mathcal{I}_k}[(\widehat{\boldsymbol{\Sigma}}_1 + \lambda \boldsymbol{I}_p)^{-1}] \simeq (v_p(-\lambda; \psi)\boldsymbol{\Sigma} + \boldsymbol{I}_p)^{-1}.$$

*(2) Bias resolvent with $\boldsymbol{C} \perp\!\!\!\perp \boldsymbol{X}$:*

$$\mathbb{E}_{I_1, I_2 \sim \mathcal{I}_k}[\lambda^2 (\widehat{\boldsymbol{\Sigma}}_1 + \lambda \boldsymbol{I}_p)^{-1} \boldsymbol{C} (\widehat{\boldsymbol{\Sigma}}_2 + \lambda \boldsymbol{I}_p)^{-1}]$$
$$\simeq (v_p(-\lambda; \psi)\boldsymbol{\Sigma} + \boldsymbol{I}_p)^{-1} \left( \widetilde{v}_p(-\lambda; \phi, \psi, \boldsymbol{C})\boldsymbol{\Sigma} + \boldsymbol{C} \right) (v_p(-\lambda; \psi)\boldsymbol{\Sigma} + \boldsymbol{I}_p)^{-1}.$$

*(3) Variance resolvent with $\boldsymbol{C} \perp\!\!\!\perp \boldsymbol{X}$:*

$$\mathbb{E}_{I_1, I_2 \sim \mathcal{I}_k} \left[ (\widehat{\boldsymbol{\Sigma}}_1 + \lambda \boldsymbol{I}_p)^{-1} \widehat{\boldsymbol{\Sigma}}_{1\cap2} (\widehat{\boldsymbol{\Sigma}}_2 + \lambda \boldsymbol{I}_p)^{-1} \boldsymbol{C} \right]$$
$$\simeq \phi^{-1} \widetilde{v}_v(-\lambda; \phi, \psi)(v_p(-\lambda; \psi)\boldsymbol{\Sigma} + \boldsymbol{I}_p)^{-2}\boldsymbol{\Sigma}\boldsymbol{C}.$$

*(4) Bias resolvent with $\boldsymbol{C} = \widehat{\boldsymbol{\Sigma}}$:*

$$\mathbb{E}_{I_1, I_2 \sim \mathcal{I}_k} \left[ \lambda^2 (\widehat{\boldsymbol{\Sigma}}_1 + \lambda \boldsymbol{I}_p)^{-1} \widehat{\boldsymbol{\Sigma}} (\widehat{\boldsymbol{\Sigma}}_2 + \lambda \boldsymbol{I}_p)^{-1} \right]$$
$$\simeq \widetilde{d}(-\lambda; \phi, \psi)(1 + \widetilde{v}_p(-\lambda; \phi, \psi))(v_p(-\lambda; \psi)\boldsymbol{\Sigma} + \boldsymbol{I}_p)^{-2}\boldsymbol{\Sigma}.$$

*Here $v(-\lambda; \phi)$ is as defined in (41), and we let $\widetilde{v}_p(-\lambda; \phi, \psi) = \widetilde{v}_p(-\lambda; \phi, \psi, \boldsymbol{\Sigma})$,*

$$\widetilde{v}_p(-\lambda; \phi, \psi, \boldsymbol{C}) = \frac{\lim\limits_{k,n,p} \phi \operatorname{tr}[\boldsymbol{C}\boldsymbol{\Sigma}(v_p(-\lambda; \psi)\boldsymbol{\Sigma} + \boldsymbol{I}_p)^{-2}]/p}{v_p(-\lambda; \psi)^{-2} - \phi \int \frac{r^2}{(1 + v_p(-\lambda; \psi)r)^2} \, \mathrm{d}H_p(r)},$$

$$\widetilde{v}_v(-\lambda; \phi, \psi) = \frac{1}{v_p(-\lambda; \psi)^{-2} - \phi \int \frac{r^2}{(1 + v_p(-\lambda; \psi)r)^2} \, \mathrm{d}H_p(r)},$$

$$\widetilde{d}(-\lambda; \phi, \psi) = \left(1 - \phi \int \frac{v_p(-\lambda; \psi)r}{1 + v_p(-\lambda; \psi)r} \, \mathrm{d}H_p(r)\right)^2.$$

*The empirical distribution $H_n$ of eigenvalues can be replaced by the limiting distribution $H$ whenever it exists.*

*Proof.* The proofs for different parts is separated below.

(1) Basic ridge resolvent.

From Definition E.2, we know that $\lambda(\widehat{\Sigma}_j + \lambda I_p)^{-1} \simeq (v_p(-\lambda; \psi)\Sigma + I_p)^{-1}$. By the definition of deterministic equivalents in Definition E.1, for any $A \in \mathbb{R}^{p \times p}$ that has bounded trace norm and is independent to $X$, we have

$$\mathrm{tr}[A(\lambda(\widehat{\Sigma}_j + \lambda I_p)^{-1} - (v_p(-\lambda; \psi)\Sigma + I_p)^{-1})] \xrightarrow{\text{a.s.}} 0.$$

From Lemma G.5 (2) of [13], it follows that

$$\mathrm{tr}[A(\lambda \mathbb{E}_{I_1 \sim \mathcal{I}_k}[(\widehat{\Sigma}_j + \lambda I_p)^{-1}] - (v_p(-\lambda; \psi)\Sigma + I_p)^{-1})]$$
$$= \mathbb{E}_{I_1 \sim \mathcal{I}_k}[\mathrm{tr}[A(\lambda(\widehat{\Sigma}_j + \lambda I_p)^{-1} - (v_p(-\lambda; \psi)\Sigma + I_p)^{-1})]] \xrightarrow{\text{a.s.}} 0, \qquad (28)$$

which implies that

$$\lambda \mathbb{E}_{I_1 \sim \mathcal{I}_k}[(\widehat{\Sigma}_1 + \lambda I_p)^{-1}] \simeq (v_p(-\lambda; \psi)\Sigma + I_p)^{-1}.$$

(2) Bias resolvent with $C \perp\!\!\!\perp X$.

Denote $M_j = (\widehat{\Sigma}_j + \lambda I_p)^{-1}$ for $j = 1, 2$. From Part (c) of the proof for Lemma S.2.4 in [12], it follows that for $I_1, I_2 \sim \mathcal{I}_k$,

$$\lambda^2 M_1 C M_2 \simeq (v_p(-\lambda; \psi)\Sigma + I_p)^{-1} \, (\widetilde{v}_p(-\lambda; \phi, \psi, C)\Sigma + C) \, (v_p(-\lambda; \psi)\Sigma + I_p)^{-1}.$$

By the same argument as in (28), the conclusion follows.

(3) Variance resolvent with $C \perp\!\!\!\perp X$.

From Lemma E.8 (3) of [13], it follows that for $I_1, I_2 \sim \mathcal{I}_k$,

$$M_1 \widehat{\Sigma}_{1 \cap 2} M_2 C \simeq \phi^{-1} \widetilde{v}_v(-\lambda; \phi, \psi)(v_p(-\lambda; \psi)\Sigma + I_p)^{-2}\Sigma C.$$

By the same argument as in (28), the conclusion follows.

(4) Bias resolvent with $C = \widehat{\Sigma}$.

We begin by decomposing the object inside expectation into two terms:

$$\lambda^2 M_1 \widehat{\Sigma} M_2 = \frac{|I_1 \cup I_2|}{n} \lambda^2 M_1 \widehat{\Sigma}_{1 \cup 2} M_2 + \frac{n - |I_1 \cup I_2|}{n} \lambda^2 M_1 \widehat{\Sigma}_{(1 \cup 2)^c} M_2 \qquad (29)$$

where $\widehat{\Sigma}_{1 \cup 2} = X^\top L_{I_1 \cup I_2} X / |I_1 \cup I_2|$ and $\widehat{\Sigma}_{(1 \cup 2)^c} = X^\top (I - L_{I_1 \cup I_2}) X / (n - |I_1 \cup I_2|)$. Next, we analyze each of them.

For the first term, from Lemma D.6 (3) of [13] we have

$$M_1 \widehat{\Sigma}_{1 \cup 2} M_2 \simeq v_p(-\lambda; \psi)^2 (1 + \widetilde{v}_p(-\lambda; \phi, \psi))$$
$$\left(\frac{2(\psi - \phi)}{2\psi - \phi} \frac{1}{\lambda v_p(-\lambda; \psi)} + \frac{\phi}{2\psi - \phi}\right) (v_p(-\lambda; \psi)\Sigma + I_p)^{-2}\Sigma \qquad (30)$$

where $\widetilde{v}_p(-\lambda; \phi, \psi) = \widetilde{v}_p(-\lambda; \phi, \psi, \Sigma)$.

For the second term, from Lemma E.8 (1) of [13] and the product rule of calculus in Lemma E.3 (3), we have

$$\lambda^2 M_1 \widehat{\Sigma}_{(1 \cup 2)^c} M_2 \simeq \lambda^2 M_1 \Sigma M_2.$$

From Part (2) and Lemma E.3 (1), it follows that

$$\lambda^2 \boldsymbol{M}_1 \widehat{\boldsymbol{\Sigma}}_{(1\cup 2)^c} \boldsymbol{M}_2 \simeq \lambda^2 \boldsymbol{M}_1 \boldsymbol{\Sigma} \boldsymbol{M}_2 \simeq (1 + \widetilde{v}_p(-\lambda; \phi, \psi)) \left(v_p(-\lambda; \psi)\boldsymbol{\Sigma} + \boldsymbol{I}_p\right)^{-2} \boldsymbol{\Sigma}. \quad (31)$$

Finally, for the coefficients of the two terms, from Lemma S.8.3 of [12], we have that

$$\frac{|I_1 \cup I_2|}{n} = \frac{|I_1| + |I_2| - |I_1 \cap I_2|}{n} \xrightarrow{\text{a.s.}} \frac{\phi(2\psi - \phi)}{\psi^2}, \qquad \frac{n - |I_1 \cup I_2|}{n} \xrightarrow{\text{a.s.}} \frac{(\psi - \phi)^2}{\psi^2}. \quad (32)$$

Combining (29)-(32) yields that

$$\lambda^2 \boldsymbol{M}_1 \widehat{\boldsymbol{\Sigma}} \boldsymbol{M}_2$$
$$\simeq \frac{1}{\psi^2} \left(2(\psi - \phi)\phi\lambda v_p(-\lambda; \psi) + \phi^2\lambda^2 v_p(-\lambda; \psi)^2 + (\psi - \phi)^2\right)$$
$$(1 + \widetilde{v}_p(-\lambda; \phi, \psi))(v_p(-\lambda; \psi)\boldsymbol{\Sigma} + \boldsymbol{I}_p)^{-2}\boldsymbol{\Sigma}$$
$$= \frac{(\phi\lambda v_p(-\lambda; \psi) + \psi - \phi)^2}{\psi^2}(1 + \widetilde{v}_p(-\lambda; \phi, \psi))(v_p(-\lambda; \psi)\boldsymbol{\Sigma} + \boldsymbol{I}_p)^{-2}\boldsymbol{\Sigma}$$
$$= \left(1 - \phi \int \frac{v_p(-\lambda; \psi)r}{1 + v_p(-\lambda; \psi)r} \, \mathrm{d}H_p(r)\right)^2 (1 + \widetilde{v}_p(-\lambda; \phi, \psi))(v_p(-\lambda; \psi)\boldsymbol{\Sigma} + \boldsymbol{I}_p)^{-2}\boldsymbol{\Sigma}$$

where the last equality follows by substituting

$$\lambda = v_p(-\lambda; \psi)^{-1} - \psi \int r(1 + v_p(-\lambda; \psi)r)^{-1} \, \mathrm{d}H_p(r)$$

based on the fixed-point equation.

$\square$

## D.2 Convergence of random linear and quadratic forms

**Lemma D.2** (Concentration of linear form with independent components and varying coefficients).
*Let $\boldsymbol{z}_i \in \mathbb{R}^p$ for $i = 1, \ldots, n$ be a sequence of random vectors with i.i.d. entries $z_{ij}$, $j = 1, \ldots, p$
such that for each $i, j$, $\mathbb{E}[z_{ij}] = 0$, $\mathbb{E}[z_{ij}^2] = 1$, $\mathbb{E}[|z_{ij}|^{4+\alpha}] \le M_\alpha$ for some $\alpha > 0$ and constant
$M_\alpha < \infty$. Let $\boldsymbol{Z} = [\boldsymbol{z}_1, \ldots, \boldsymbol{z}_n]^\top \in \mathbb{R}^{n \times p}$ be the random matrix formed by concatenating
$\boldsymbol{z}_i$'s. Let $g : \mathbb{R}^p \to \mathbb{R}$ be any measurable function such that $\mathbb{E}[g(\boldsymbol{z}_i)] = 0$ and $\mathbb{E}[\boldsymbol{z}_i g(\boldsymbol{z}_i)] = 0$
for $i = 1, \ldots, n$. Let $\boldsymbol{a}_p \in \mathbb{R}^p$ be a sequence of random vectors independent of $\boldsymbol{z}_p$ such that
$\limsup_p \|\boldsymbol{a}_p\|^2/p \le M_0$ almost surely for a constant $M_0 < \infty$. Let $\boldsymbol{D}$ be a positive semidefinite
matrix such that $\limsup \|\boldsymbol{D}\|_{\mathrm{op}} \le M_0$ almost surely as $p \to \infty$ for some constant $M_0 < \infty$. Then,
as $n, p \to \infty$ such that $p/n \to \phi \in (0, \infty)$, we have*

$$\left| \frac{1}{n} \sum_{i=1}^n \left[ \boldsymbol{a}^\top \left( \frac{\boldsymbol{D}^{\frac{1}{2}} \boldsymbol{Z}^\top \boldsymbol{Z} \boldsymbol{D}^{\frac{1}{2}}}{n} + \lambda \boldsymbol{I}_p \right)^{-1} \boldsymbol{D}^{\frac{1}{2}} \boldsymbol{Z}^\top \right]_i g(\boldsymbol{z}_i) \right| \xrightarrow{\text{a.s.}} 0. \quad (33)$$

*Proof.* We will use the standard leave-one-out trick to break the dependence between the $i$-th
component multiplier in the summation in (33) and $g(\boldsymbol{z}_i)$ for each $i = 1, \ldots, n$. To that end, using
the Woodbury matrix identity, first observe that

$$\left( \frac{\boldsymbol{D}^{\frac{1}{2}} \boldsymbol{Z}^\top \boldsymbol{Z} \boldsymbol{D}^{\frac{1}{2}}}{n} + \lambda \boldsymbol{I}_p \right)^{-1}$$
$$= \left( \frac{\boldsymbol{D}^{\frac{1}{2}} \boldsymbol{Z}_{-i}^\top \boldsymbol{Z}_{-i} \boldsymbol{D}^{\frac{1}{2}}}{n} + \frac{\boldsymbol{D}^{\frac{1}{2}} \boldsymbol{z}_i \boldsymbol{z}_i^\top \boldsymbol{D}^{\frac{1}{2}}}{n} + \lambda \boldsymbol{I}_p \right)^{-1}$$
$$= \left( \frac{\boldsymbol{D}^{\frac{1}{2}} \boldsymbol{Z}_{-i}^\top \boldsymbol{Z}_{-i} \boldsymbol{D}^{\frac{1}{2}}}{n} + \lambda \boldsymbol{I}_p \right)^{-1}$$

$$-\frac{\left(\frac{\boldsymbol{D}^{\frac{1}{2}}\boldsymbol{Z}_{-i}^{\top}\boldsymbol{Z}_{-i}\boldsymbol{D}^{\frac{1}{2}}}{n}+\lambda\boldsymbol{I}_p\right)^{-1}\frac{\boldsymbol{D}^{\frac{1}{2}}\boldsymbol{z}_i\boldsymbol{z}_i^{\top}\boldsymbol{D}^{\frac{1}{2}}}{n}\left(\frac{\boldsymbol{D}^{\frac{1}{2}}\boldsymbol{Z}_{-i}^{\top}\boldsymbol{Z}_{-i}\boldsymbol{D}^{\frac{1}{2}}}{n}+\lambda\boldsymbol{I}_p\right)^{-1}}{1+\frac{\boldsymbol{z}_i^{\top}\boldsymbol{D}^{\frac{1}{2}}\left(\frac{\boldsymbol{D}^{\frac{1}{2}}\boldsymbol{Z}_{-i}^{\top}\boldsymbol{Z}_{-i}\boldsymbol{D}^{\frac{1}{2}}}{n}+\lambda\boldsymbol{I}_p\right)^{-1}\boldsymbol{D}^{\frac{1}{2}}\boldsymbol{z}_i}{n}}. \tag{34}$$

Plugging (34) back into (33), we expand the desired sum into:

$$\frac{1}{n}\sum_{i=1}^{n}\left[\boldsymbol{a}^{\top}\left(\frac{\boldsymbol{D}^{\frac{1}{2}}\boldsymbol{Z}^{\top}\boldsymbol{Z}\boldsymbol{D}^{\frac{1}{2}}}{n}+\lambda\boldsymbol{I}_p\right)^{-1}\boldsymbol{D}^{\frac{1}{2}}\boldsymbol{Z}^{\top}\right]_i g(\boldsymbol{z}_i)$$

$$=\frac{1}{n}\sum_{i=1}^{n}\boldsymbol{a}^{\top}\left(\frac{\boldsymbol{D}^{\frac{1}{2}}\boldsymbol{Z}^{\top}\boldsymbol{Z}\boldsymbol{D}^{\frac{1}{2}}}{n}+\lambda\boldsymbol{I}_p\right)^{-1}\boldsymbol{D}^{\frac{1}{2}}\boldsymbol{z}_i g(\boldsymbol{z}_i)$$

$$=\frac{1}{n}\sum_{i=1}^{n}\boldsymbol{a}^{\top}\left(\frac{\boldsymbol{D}^{\frac{1}{2}}\boldsymbol{Z}_{-i}^{\top}\boldsymbol{Z}_{-i}\boldsymbol{D}^{\frac{1}{2}}}{n}+\lambda\boldsymbol{I}_p\right)^{-1}\boldsymbol{D}^{\frac{1}{2}}\boldsymbol{z}_i g(\boldsymbol{z}_i)$$

$$-\frac{1}{n}\sum_{i=1}^{n}\frac{\boldsymbol{a}^{\top}\left(\frac{\boldsymbol{D}^{\frac{1}{2}}\boldsymbol{Z}_{-i}^{\top}\boldsymbol{Z}_{-i}\boldsymbol{D}^{\frac{1}{2}}}{n}+\lambda\boldsymbol{I}_p\right)^{-1}\frac{\boldsymbol{D}^{\frac{1}{2}}\boldsymbol{z}_i\boldsymbol{z}_i^{\top}\boldsymbol{D}^{\frac{1}{2}}}{n}\left(\frac{\boldsymbol{D}^{\frac{1}{2}}\boldsymbol{Z}_{-i}^{\top}\boldsymbol{Z}_{-i}\boldsymbol{D}^{\frac{1}{2}}}{n}+\lambda\boldsymbol{I}_p\right)^{-1}\boldsymbol{D}^{\frac{1}{2}}\boldsymbol{z}_i g(\boldsymbol{z}_i)}{1+\frac{\boldsymbol{z}_i^{\top}\boldsymbol{D}^{\frac{1}{2}}\left(\frac{\boldsymbol{D}^{\frac{1}{2}}\boldsymbol{Z}_{-i}^{\top}\boldsymbol{Z}_{-i}\boldsymbol{D}^{\frac{1}{2}}}{n}+\lambda\boldsymbol{I}_p\right)^{-1}\boldsymbol{D}^{\frac{1}{2}}\boldsymbol{z}_i}{n}}$$

$$=\frac{1}{n}\sum_{i=1}^{n}\boldsymbol{b}_i^{\top}\boldsymbol{z}_i g(\boldsymbol{z}_i)-\frac{1}{n}\sum_{i=1}^{n}\frac{\boldsymbol{b}_i^{\top}d_i\boldsymbol{z}_i g(\boldsymbol{z}_i)}{1+d_i} \tag{35}$$

$$\leq\frac{1}{n}\sum_{i=1}^{n}(1+d_i)\boldsymbol{b}_i^{\top}\boldsymbol{z}_i g(\boldsymbol{z}_i), \tag{36}$$

where in step (35), we denote by:

$$\boldsymbol{b}_i=\boldsymbol{D}^{\frac{1}{2}}\left(\frac{\boldsymbol{D}^{\frac{1}{2}}\boldsymbol{Z}_{-i}^{\top}\boldsymbol{Z}_{-i}\boldsymbol{D}^{\frac{1}{2}}}{n}+\lambda\boldsymbol{I}_p\right)^{-1}\boldsymbol{a},$$

$$d_i=\frac{\boldsymbol{z}_i^{\top}\boldsymbol{D}^{\frac{1}{2}}\left(\frac{\boldsymbol{D}^{\frac{1}{2}}\boldsymbol{Z}_{-i}^{\top}\boldsymbol{Z}_{-i}\boldsymbol{D}^{\frac{1}{2}}}{n}+\lambda\boldsymbol{I}_p\right)^{-1}\boldsymbol{D}^{\frac{1}{2}}\boldsymbol{z}_i}{n},$$

and step (36) follows since $d_i\geq 0$. It is easy to check that $\limsup_p\|b_i\|_2^2/p\leq C_1<\infty$, and $d_i\xrightarrow{\text{a.s.}}C_2<\infty$ for some constants $C_1$ and $C_2$. Appealing to Lemma S.8.5 of [51], (36) almost surely converges to 0. This finishes the proof. $\qquad\square$

**Lemma D.3** (Concentration of sum of quadratic forms with independent components and independent varying inner matrices). *Let $\boldsymbol{z}_p\in\mathbb{R}^p$ be a sequence of random vector with i.i.d. entries $z_{pi}$, $i=1,\ldots,p$ such that for each $i$, $\mathbb{E}[z_{pi}]=0$, $\mathbb{E}[z_{pi}^2]=1$, $\mathbb{E}[|z_{pi}|^{4+\alpha}]\leq M_\alpha$ for some $\alpha>0$ and constant $M_\alpha<\infty$. Let $\boldsymbol{Z}=[\boldsymbol{z}_1,\ldots,\boldsymbol{z}_n]^{\top}\in\mathbb{R}^{n\times p}$ be the design matrix. Let $g:\mathbb{R}^p\to\mathbb{R}$ be any measurable function such that $\mathbb{E}[z_ig(z_i)]=0$ and $\mathbb{E}[g(z_i)]=1$. Let $\boldsymbol{D}$ be a positive semidefinite matrix such that $\limsup\|\boldsymbol{D}\|_{\text{op}}\leq M_0$ almost surely as $p\to\infty$ for some constant $M_0<\infty$. Then, as $n,p\to\infty$ such that $p/n\to\phi\in(0,\infty)$,*

$$\left|\frac{1}{n}\sum_{1\leq i\neq j\leq n}\left(\frac{\boldsymbol{Z}\boldsymbol{D}\boldsymbol{Z}^{\top}}{n}+\lambda\boldsymbol{I}_n\right)^{-1}_{ij}g(\boldsymbol{z}_i)g(\boldsymbol{z}_j)\right|\xrightarrow{\text{a.s.}}0. \tag{37}$$

*Proof.* The strategy in the proof is to express each of the $ij$-the entry of the resolvent in (37) such that the dependence on $(\boldsymbol{z}_i,\boldsymbol{z}_j)$ and the rest of $(\boldsymbol{z}_k:k\neq i,j)$ is separated. One is then able to use the

uncorrelatedness of $\boldsymbol{z}$ and $g(\boldsymbol{z})$ along with standard concentration of a quadratic form with respect to an independent matrix. Similar strategy has been used in [14] when analyzing kernel ridge regression under proportional asymptotics.

Denote $\boldsymbol{Y} = (\boldsymbol{ZDZ}^\top/n + \lambda \boldsymbol{I}_n)^{-1}$. For each pair of $i, j \in [n]$ and $i \neq j$, we let $\boldsymbol{Z}_{-(ij)} \in \mathbb{R}^{(n-2)\times p}$ be the matrix comprising the $n - 2$ rows of $\boldsymbol{Z}$ excluding the $i$th and $j$th rows, and $\boldsymbol{U} \in \mathbb{R}^{p\times 2}$ be the matrix with columns $\boldsymbol{U}_{ij}\boldsymbol{e}_i = \boldsymbol{z}_i, \boldsymbol{U}\boldsymbol{e}_j = \boldsymbol{z}_j$. We finally define the matrices $\boldsymbol{R}_{-(ij)} \in \mathbb{R}^{p\times p}$ as follows:

$$\boldsymbol{R}_{-(ij)} = \lambda \boldsymbol{D}^{1/2}\left(\boldsymbol{D}^{1/2}\boldsymbol{Z}_{-(ij)}^\top \boldsymbol{Z}_{-(ij)}\boldsymbol{D}^{1/2}/n + \lambda \boldsymbol{I}_p\right)^{-1}\boldsymbol{D}^{1/2},$$

and let $\widetilde{\boldsymbol{Y}}_{ij} = \{Y_{m,\ell}\}_{m,\ell\in\{i,j\}}$ be the submatrix of $\boldsymbol{Y}$. Then, using the block matrix inversion formula (see, e.g., Appendix A.1.4 of [45]) and the Woodbury matrix identity, one can show that

$$\widetilde{\boldsymbol{Y}}_{ij} = \left(\boldsymbol{U}_{ij}^\top \boldsymbol{R}_{-(ij)}\boldsymbol{U}_{ij}/n + \lambda \boldsymbol{I}_2\right)^{-1},$$

$$Y_{ij} = -\frac{\left\langle \boldsymbol{z}_i, \boldsymbol{R}_{-(ij)}\boldsymbol{z}_j\right\rangle/n}{d_{ij}},$$

where $d_{ij}$ is given by:

$$d_{ij} = \left(\lambda + \left\langle \boldsymbol{z}_i, \boldsymbol{R}_{-(ij)}\boldsymbol{z}_j\right\rangle/n\right)\left(\lambda + \left\langle \boldsymbol{z}_i, \boldsymbol{R}_{-(ij)}\boldsymbol{z}_j\right\rangle/n\right) - \left\langle \boldsymbol{z}_i, \boldsymbol{R}_{-(ij)}\boldsymbol{z}_j\right\rangle^2/n^2.$$

Since $\boldsymbol{z}_i$, $\boldsymbol{z}_j$, and $\boldsymbol{R}_{-(ij)}$ are mutually independent, by Lemma S.8.5 of [51], we have $\left\langle \boldsymbol{z}_i, \boldsymbol{R}_{-(ij)}\boldsymbol{z}_j\right\rangle/n \xrightarrow{\text{a.s.}} 0$ and the denominator concentrates on $\lambda^2$. By Lemma G.5 (1) of [13], it follows that $\max_{1\le i\neq j\le n} d_{ij} \xrightarrow{\text{a.s.}} \lambda^2$. Thus, there exists $N_0 \in \mathbb{N}$ such that for all $n \geq N_0$, $\max_{1\le i\neq j\le n} d_{ij} \geq \lambda^2/2$ almost surely. Then, it follows that for $n \geq N_0$,

$$\left|\frac{1}{p}\sum_{1\le i\neq j\le n} Y_{ij}g(\boldsymbol{z}_i)g(\boldsymbol{z}_j)\right| \leq \frac{2}{\lambda^2}\frac{1}{n^2}\sum_{1\le i\neq j\le n}|\left\langle \boldsymbol{z}_i, \boldsymbol{R}_{-(ij)}\boldsymbol{z}_j\right\rangle g(\boldsymbol{z}_i)g(\boldsymbol{z}_j)|$$

$$= \frac{2}{\lambda^2}\frac{1}{n^2}\sum_{1\le i\neq j\le n}(\boldsymbol{z}_i g(\boldsymbol{z}_i))^\top \boldsymbol{R}_{-(ij)}(\boldsymbol{z}_j g(\boldsymbol{z}_j))$$

$$\xrightarrow{\text{a.s.}} 0,$$

where the last convergence is from Lemma S.8.5 of [51] and Lemma G.5 (2) of [13]. $\qquad\square$

**Lemma D.4.** *For any two conforming matrices $\boldsymbol{A}$ and $\boldsymbol{B}$ and any $t \neq 0$, we have*
$$\boldsymbol{A}^{-1}\boldsymbol{B}\boldsymbol{A}^{-1} = (\boldsymbol{A}^{-1} - (\boldsymbol{A}+t\boldsymbol{B})^{-1})/t + t\boldsymbol{A}^{-1}\boldsymbol{B}(\boldsymbol{A}+t\boldsymbol{B})^{-1}\boldsymbol{B}\boldsymbol{A}^{-1}.$$

*Proof.* We recall the Woodbury matrix identity:
$$(\boldsymbol{A} + \boldsymbol{U}\boldsymbol{C}\boldsymbol{V})^{-1} = \boldsymbol{A}^{-1} - \boldsymbol{A}^{-1}\boldsymbol{U}(\boldsymbol{C}^{-1} + \boldsymbol{V}\boldsymbol{A}^{-1}\boldsymbol{U})^{-1}\boldsymbol{V}\boldsymbol{A}^{-1}.$$

This holds for any conforming matrices $\boldsymbol{A}$, $\boldsymbol{U}$, $\boldsymbol{C}$, and $\boldsymbol{V}$. We will need to apply the Woodbury matrix identity twice below.

1. Applying the Woodbury identity for the first time with $\boldsymbol{A} = \boldsymbol{A}, \boldsymbol{U} = t, \boldsymbol{C} = \boldsymbol{B}$, and $\boldsymbol{V} = 1$, we get
$$(\boldsymbol{A} + t\boldsymbol{B})^{-1} = \boldsymbol{A}^{-1} - t\boldsymbol{A}^{-1}(\boldsymbol{B}^{-1} + t\boldsymbol{A}^{-1})^{-1}\boldsymbol{A}^{-1}.$$
Rearranging, this yields
$$(\boldsymbol{A}^{-1} - (\boldsymbol{A}+t\boldsymbol{B})^{-1})/t = \boldsymbol{A}^{-1}(\boldsymbol{B}^{-1} + t\boldsymbol{A}^{-1})^{-1}\boldsymbol{A}^{-1} = \boldsymbol{A}^{-1}(t\boldsymbol{A}^{-1} + \boldsymbol{B}^{-1})^{-1}\boldsymbol{A}^{-1}. \quad (38)$$

2. Applying the Woodbury identity the second time with $\boldsymbol{A} = t\boldsymbol{B}, \boldsymbol{U} = 1, \boldsymbol{C} = \boldsymbol{A}$, and $\boldsymbol{V} = 1$, we get
$$(\boldsymbol{A} + t\boldsymbol{B})^{-1} = t^{-1}\boldsymbol{B}^{-1} - t^{-1}\boldsymbol{B}^{-1}(\boldsymbol{A}^{-1} + t^{-1}\boldsymbol{B}^{-1})^{-1}t^{-1}\boldsymbol{B}^{-1}.$$
Multiplying by $t\boldsymbol{B}$ from the left yields
$$t\boldsymbol{B}(A + t\boldsymbol{B})^{-1} = I - (\boldsymbol{A}^{-1} + t^{-1}\boldsymbol{B}^{-1})^{-1}t^{-1}\boldsymbol{B}^{-1} = I - (t\boldsymbol{A}^{-1} + \boldsymbol{B}^{-1})^{-1}\boldsymbol{B}^{-1}.$$
Now, multiplying by $\boldsymbol{B}$ on the right, notice that
$$t\boldsymbol{B}(A + t\boldsymbol{B})^{-1}\boldsymbol{B} = \boldsymbol{B} - (t\boldsymbol{A}^{-1} + \boldsymbol{B}^{-1})^{-1}. \quad (39)$$

From (38), observe that

$$\boldsymbol{A}^{-1}\boldsymbol{B}\boldsymbol{A}^{-1} - (\boldsymbol{A}^{-1} - (\boldsymbol{A}+t\boldsymbol{B})^{-1})/t = \boldsymbol{A}^{-1}\boldsymbol{B}\boldsymbol{A}^{-1} - \boldsymbol{A}^{-1}(t\boldsymbol{A}^{-1} + \boldsymbol{B}^{-1})^{-1}\boldsymbol{A}^{-1}$$
$$= \boldsymbol{A}^{-1}(\boldsymbol{B} - (t\boldsymbol{A}^{-1} + \boldsymbol{B}^{-1})^{-1})\boldsymbol{A}^{-1}.$$

Using (39), we then have

$$\boldsymbol{A}^{-1}\boldsymbol{B}\boldsymbol{A}^{-1} - (\boldsymbol{A}^{-1} - (\boldsymbol{A}+t\boldsymbol{B})^{-1})/t = t\boldsymbol{A}^{-1}\boldsymbol{B}(\boldsymbol{A}+t\boldsymbol{B})^{-1}\boldsymbol{B}\boldsymbol{A}^{-1}.$$

Rearranging, we arrive at the desired equality:

$$\boldsymbol{A}^{-1}\boldsymbol{B}\boldsymbol{A}^{-1} = (\boldsymbol{A}^{-1} - (\boldsymbol{A}+t\boldsymbol{B})^{-1})/t + t\boldsymbol{A}^{-1}\boldsymbol{B}(\boldsymbol{A}+t\boldsymbol{B})^{-1}\boldsymbol{B}\boldsymbol{A}^{-1}.$$

$\square$

### D.3 Concentration of energy of linear and nonlinear components

**Lemma D.5.** *Under Assumptions 1–2, the best linear estimator $\boldsymbol{\beta}_0$ and the nonlinear component $f_{\mathrm{NL}}$ as defined in* (10) *satisfies that $\|\boldsymbol{\beta}_0\|_2$ and $\|f_{\mathrm{NL}}\|_{L_{4+\delta}}$ are bounded almost surely.*

*Proof.* By Jensen's inequality and Assumption 1, we have $\mathbb{E}[y^2] \leq \mathbb{E}[y^{4+\delta}]^{2/(4+\delta)} \leq C_0^{2/(4+\delta)}$ for some constant $C_0 > 0$.

By the orthogonality, we have that $\mathbb{E}[y^2] = \mathbb{E}[(\boldsymbol{x}^\top\boldsymbol{\beta}_0)^2] + \|f_{\mathrm{NL}}\|_{L^2}^2$, which implies that $\mathbb{E}[(\boldsymbol{x}^\top\boldsymbol{\beta}_0)^2]$ and $\|f_{\mathrm{NL}}\|_{L^2}^2$ is bounded. Since by Assumption 2, the eigenvalues of $\boldsymbol{\Sigma}$ is lower bounded by $r_{\min} > 0$, we have that $\mathbb{E}[(\boldsymbol{x}^\top\boldsymbol{\beta}_0)^2] = \boldsymbol{\beta}_0\boldsymbol{\Sigma}\boldsymbol{\beta}_0 \geq r_{\min}\|\boldsymbol{\beta}_0\|_2^2$ and thus $\|\boldsymbol{\beta}_0\|_2$ is bounded.

Since $f_{\mathrm{NL}} = y - \boldsymbol{x}^\top\boldsymbol{\beta}_0$, by triangle inequality we have that

$$\|f_{\mathrm{NL}}\|_{L_{4+\delta}} \leq \|y\|_{L_{4+\delta}} + \|\boldsymbol{x}^\top\boldsymbol{\beta}_0\|_{L_{4+\delta}}$$
$$\leq \|y\|_{L_{4+\delta}} + C\|\boldsymbol{\beta}_0\|_2)$$

for some constant $C > 0$. The second inequality of the above display is from Lemma 7.8 of [57]. Since $\|y\|_{L_{4+\delta}}$ and $\|\boldsymbol{\beta}_0\|_2$ are bounded, it follows that $\|f_{\mathrm{NL}}\|_{L_{4+\delta}}$ is also bounded. $\square$

## E Asymptotic equivalents: background and known results

**Preliminary background.** In several proofs, we employ the concept of asymptotic equivalence of sequences of random matrices; see [12, 44, 49, 51]. This section provides a brief overview of the associated terminology and calculus principles.

**Definition E.1** (Asymptotic equivalence)**.** *Consider sequences $\{\boldsymbol{A}_p\}_{p\geq1}$ and $\{\boldsymbol{B}_p\}_{p\geq1}$ of (random or deterministic) matrices of growing dimensions. We say that $\boldsymbol{A}_p$ and $\boldsymbol{B}_p$ are equivalent and write $\boldsymbol{A}_p \simeq \boldsymbol{B}_p$ if $\lim_{p\to\infty} |\mathrm{tr}[\boldsymbol{C}_p(\boldsymbol{A}_p - \boldsymbol{B}_p)]| = 0$ almost surely for every sequence of random matrices $\boldsymbol{C}_p$ independent to $\boldsymbol{A}_p$ and $\boldsymbol{B}_p$, with bounded trace norm such that $\limsup_{p\to\infty} \|\boldsymbol{C}_p\|_{\mathrm{tr}} < \infty$ almost surely.*

The notion of asymptotic equivalence of two sequences of random matrices above can be further extended to incorporate conditioning on another sequence of random matrices; see [12] for more details.

**Definition E.2** (Conditional asymptotic equivalence)**.** *Consider sequences $\{\boldsymbol{A}_p\}_{p\geq1}$, $\{\boldsymbol{B}_p\}_{p\geq1}$ and $\{\boldsymbol{D}_p\}_{p\geq1}$ of (random or deterministic) matrices of growing dimensions. We say that $\boldsymbol{A}_p$ and $\boldsymbol{B}_p$ are equivalent given $\boldsymbol{D}_p$ and write $\boldsymbol{A}_p \simeq \boldsymbol{B}_p \mid \boldsymbol{D}_p$ if $\lim_{p\to\infty} |\mathrm{tr}[\boldsymbol{C}_p(\boldsymbol{A}_p - \boldsymbol{B}_p)]| = 0$ almost surely conditional on $\{\boldsymbol{D}_p\}_{p\geq1}$, i.e.,*

$$\mathbb{P}\left(\lim_{p\to\infty} |\mathrm{tr}[\boldsymbol{C}_p(\boldsymbol{A}_p - \boldsymbol{B}_p)]| = 0 \,\middle|\, \{\boldsymbol{D}_p\}_{p\geq1}\right) = 1,$$

*for any sequence of random matrices $\boldsymbol{C}_p$ independent to $\boldsymbol{A}_p$ and $\boldsymbol{B}_p$ conditional on $\boldsymbol{D}_p$, with bounded trace norm such that $\limsup \|\boldsymbol{C}_p\|_{\mathrm{tr}} < \infty$ as $p \to \infty$.*

Below we summarize the calculus rules for conditional asymptotic equivalence in Definition E.2. These are adapted from Lemma S.7.4 and S.7.6 of [12].

**Lemma E.3** (Calculus of asymptotic equivalents). *Let $A_p$, $B_p$, $C_p$ and $D_p$ be sequences of random matrices. The calculus of asymptotic equivalents satisfies the following properties:*

(1) *Equivalence: The relation $\simeq$ is an equivalence relation.*

(2) *Sum: If $A_p \simeq B_p \mid E_p$ and $C_p \simeq D_p \mid E_p$, then $A_p + C_p \simeq B_p + D_p \mid E_p$.*

(3) *Product: If $A_p$ has bounded operator norms such that $\limsup_{p \to \infty} \|A_p\|_{\mathrm{op}} < \infty$, $A_p$ is conditional independent to $B_p$ and $C_p$ given $E_p$ for $p \geq 1$, and $B_p \simeq C_p \mid E_p$, then $A_p B_p \simeq A_p C_p \mid E_p$.*

(4) *Trace: If $A_p \simeq B_p \mid E_p$, then $\mathrm{tr}[A_p]/p - \mathrm{tr}[B_p]/p \to 0$ almost surely when conditioning on $E_p$.*

(5) *Differentiation: Suppose $f(z, A_p) \simeq g(z, B_p) \mid E_p$ where the entries of $f$ and $g$ are analytic functions in $z \in S$ and $S$ is an open connected subset of $\mathbb{C}$. Suppose for any sequence $C_p$ of deterministic matrices with bounded trace norm we have $|\mathrm{tr}[C_p(f(z, A_p) - g(z, B_p))]| \leq M$ for every $p$ and $z \in S$. Then, we have $f'(z, A_p) \simeq g'(z, B_p) \mid E_p$ for every $z \in S$, where the derivatives are taken entrywise with respect to $z$.*

(6) *Unconditioning: If $A_p \simeq B_p \mid E_p$, then $A_p \simeq B_p$.*

(7) *Substitution: Let $v : \mathbb{R}^{p \times p} \to \mathbb{R}$ and $f(v(C), C) : \mathbb{R}^{p \times p} \to \mathbb{R}^{p \times p}$ be a matrix function for matrix $C \in \mathbb{R}^{p \times p}$ and $p \in \mathbb{N}$, that is continuous in the first augment with respect to operator norm. If $v(C) \overset{a.s.}{=} v(D)$ such that $C$ is independent to $D$, then $f(v(C), C) \simeq f(v(D), C) \mid C$.*

**Standard ridge resolvents and various extensions.** In this section, we collect various asymptotic equivalents. The following corollary is a simple consequence of Theorem 1 in [58]. It provides deterministic equivalent for the scaled ridge resolvent.

**Corollary E.4** (Deterministic equivalent for scaled ridge resolvent). *Suppose $x_i \in \mathbb{R}^p$, $1 \leq i \leq n$, are i.i.d. random vectors such that each $x_i = z_i \Sigma^{1/2}$, where $z_i$ is a random vector consisting of i.i.d. entries $z_{ij}$, $1 \leq j \leq p$, satisfying $\mathbb{E}[z_{ij}] = 0$, $\mathbb{E}[z_{ij}^2] = 1$, and $\mathbb{E}[|z_{ij}|^{8+\alpha}] \leq M_\alpha$ for some constants $\alpha > 0$ and $M_\alpha < \infty$, and $\Sigma \in \mathbb{R}^{p \times p}$ is a positive semidefinite matrix satisfying $0 \preceq \Sigma \preceq r_{\max} I_p$ for some constant $r_{\max} < \infty$ (independent of $p$). Let $X \in \mathbb{R}^{n \times p}$ the concatenated matrix with $x_i^\top$, $1 \leq i \leq n$, as rows, and let $\widehat{\Sigma} \in \mathbb{R}^{p \times p}$ denote the random matrix $X^\top X/n$. Let $\gamma = p/n$. Then, for $z \in \mathbb{C}^+$, as $n, p \to \infty$ such that $0 < \liminf \gamma \leq \limsup \gamma < \infty$, and $\lambda > 0$, we have*

$$\lambda(\widehat{\Sigma} + \lambda I_p)^{-1} \simeq (v(-\lambda; \gamma)\Sigma + I_p)^{-1},$$

*where $v(-\lambda; \gamma) > 0$ is the unique solution to the fixed-point equation*

$$\frac{1}{v(-\lambda; \gamma)} = \lambda + \gamma \int \frac{r}{1 + v(-\lambda; \gamma)r} \, \mathrm{d}H_p(r). \tag{40}$$

*Here $H_n$ is the empirical distribution (supported on $\mathbb{R}_{\geq 0}$) of the eigenvalues of $\Sigma$.*

Side remark: The parameter $v(-\lambda; \gamma)$ in Corollary E.4 is also the companion Stieltjes transform of the spectral distribution of the sample covariance matrix $\widehat{\Sigma}$. It is also the Stieltjes transform of the spectral distribution of the gram matrix $XX^\top/n$. This is essentially the characterization we use in proving Proposition 4.

The following lemma uses Corollary E.4 along with calculus of deterministic equivalents (from Lemma E.3), and provides deterministic equivalents for resolvents needed to obtain asymptotic bias and variance of standard ridge regression. It is adapted from Lemma S.6.10 of [51].

**Lemma E.5** (Deterministic equivalents for ridge resolvents associated with generalized bias and variance). *Suppose $x_i \in \mathbb{R}^p$, $1 \leq i \leq n$, are i.i.d. random vectors with each $x_i = z_i \Sigma^{1/2}$, where $z_i \in \mathbb{R}^p$ is a random vector that contains i.i.d. random variables $z_{ij}$, $1 \leq j \leq p$, each with $\mathbb{E}[z_{ij}] = 0$, $\mathbb{E}[z_{ij}^2] = 1$, and $\mathbb{E}[|z_{ij}|^{4+\alpha}] \leq M_\alpha$ for some constants $\alpha > 0$ and $M_\alpha < \infty$, and $\Sigma \in \mathbb{R}^{p \times p}$ is a positive semidefinite matrix with $r_{\min} I_p \preceq \Sigma \preceq r_{\max} I_p$ for some constants $r_{\min} > 0$ and $r_{\max} < \infty$ (independent of $p$). Let $X \in \mathbb{R}^{n \times p}$ be the concatenated random matrix with $x_i$, $1 \leq i \leq n$, as its rows, and define $\widehat{\Sigma} = X^\top X/n \in \mathbb{R}^{p \times p}$. Let $\gamma = p/n$. Then, for $\lambda > 0$, as $n, p \to \infty$ with $0 < \liminf \gamma \leq \limsup \gamma < \infty$, the following statements hold:*

*(1) Bias of ridge regression:*

$$\lambda^2(\widehat{\boldsymbol{\Sigma}} + \lambda\boldsymbol{I}_p)^{-1}\boldsymbol{A}(\widehat{\boldsymbol{\Sigma}} + \lambda\boldsymbol{I}_p)^{-1}$$
$$\simeq (v(-\lambda;\gamma,\boldsymbol{\Sigma})\boldsymbol{\Sigma} + \boldsymbol{I}_p)^{-1}(\widetilde{v}_b(-\lambda;\gamma,\boldsymbol{\Sigma},\boldsymbol{A})\boldsymbol{\Sigma} + \boldsymbol{A})(v(-\lambda;\gamma,\boldsymbol{\Sigma})\boldsymbol{\Sigma} + \boldsymbol{I}_p)^{-1}.$$

*(2) Variance of ridge regression:*

$$(\widehat{\boldsymbol{\Sigma}} + \lambda\boldsymbol{I}_p)^{-2}\widehat{\boldsymbol{\Sigma}}\boldsymbol{A} \simeq \widetilde{v}_v(-\lambda;\gamma,\boldsymbol{\Sigma})(v(-\lambda;\gamma,\boldsymbol{\Sigma})\boldsymbol{\Sigma} + \boldsymbol{I}_p)^{-2}\boldsymbol{\Sigma}\boldsymbol{A}.$$

*Here $v(-\lambda;\gamma,\boldsymbol{\Sigma}) > 0$ is the unique solution to the fixed-point equation*

$$\frac{1}{v(-\lambda;\gamma,\boldsymbol{\Sigma})} = \lambda + \int \frac{\gamma r}{1 + v(-\lambda;\gamma,\boldsymbol{\Sigma})r}\,\mathrm{d}H_n(r;\boldsymbol{\Sigma}), \tag{41}$$

*and $\widetilde{v}_b(-\lambda;\gamma,\boldsymbol{\Sigma})$ and $\widetilde{v}_v(-\lambda;\gamma,\boldsymbol{\Sigma})$ are defined through $v(-\lambda;\gamma,\boldsymbol{\Sigma})$ by the following equations:*

$$\widetilde{v}_b(-\lambda;\gamma,\boldsymbol{\Sigma},\boldsymbol{A}) = \frac{\gamma\,\mathrm{tr}[\boldsymbol{A}\boldsymbol{\Sigma}(v(-\lambda;\gamma,\boldsymbol{\Sigma})\boldsymbol{\Sigma} + \boldsymbol{I}_p)^{-2}]/p}{v(-\lambda;\gamma,\boldsymbol{\Sigma})^{-2} - \int \gamma r^2(1 + v(-\lambda;\gamma,\boldsymbol{\Sigma})r)^{-2}\,\mathrm{d}H_n(r;\boldsymbol{\Sigma})}, \tag{42}$$

$$\widetilde{v}_v(-\lambda;\gamma,\boldsymbol{\Sigma})^{-1} = v(-\lambda;\gamma,\boldsymbol{\Sigma})^{-2} - \int \gamma r^2(1 + v(-\lambda;\gamma,\boldsymbol{\Sigma})r)^{-2}\,\mathrm{d}H_n(r;\boldsymbol{\Sigma}), \tag{43}$$

*where $H_n(\cdot;\boldsymbol{\Sigma})$ is the empirical distribution (supported on $[r_{\min}, r_{\max}]$) of the eigenvalues of $\boldsymbol{\Sigma}$.*

Though Lemma E.5 states the dependency explicitly, we will simply write $H_p(r)$, $v(-\lambda;\gamma)$, $\widetilde{v}_b(-\lambda;\gamma,\boldsymbol{A})$, and $\widetilde{v}_v(-\lambda;\gamma)$ to denote $H_n(r;\boldsymbol{\Sigma})$, $v(-\lambda;\gamma,\boldsymbol{\Sigma})$, $\widetilde{v}_b(-\lambda;\gamma,\boldsymbol{\Sigma},\boldsymbol{A})$, and $\widetilde{v}_v(-\lambda;\gamma,\boldsymbol{\Sigma})$, respectively, for simplifying various notations when it is clear from the context. When $\boldsymbol{A} = \boldsymbol{\Sigma}$, we simply write $\widetilde{v}_b(-\lambda;\gamma) = \widetilde{v}_b(-\lambda;\gamma,\boldsymbol{A})$.

# F  Experiment details

## F.1  Reproducibility and compute details

The source code for generating all of our figures is included with the supplementary material. The source code also includes details about the computational resources used to run the code and other timing details.

## F.2  Simulation details

The covariance matrix of an auto-regressive process of order 1 (AR(1)) is given by $\boldsymbol{\Sigma}_{\mathrm{ar1}}$, where $(\boldsymbol{\Sigma}_{\mathrm{ar1}})_{ij} = \rho_{\mathrm{ar1}}^{|i-j|}$ for some parameter $\rho_{\mathrm{ar1}} \in (0,1)$. Define $\boldsymbol{\beta}_0 := \frac{1}{5}\sum_{j=1}^{5}\boldsymbol{w}_{(j)}$ where $\boldsymbol{w}_{(j)}$ is the eigenvector of $\boldsymbol{\Sigma}_{\mathrm{ar1}}$ associated with the top $j$th eigenvalue $r_{(j)}$. We generated data $(\boldsymbol{x}_i, y_i)$ for $i = 1, \ldots, n$ from a nonlinear model:

$$y_i = \boldsymbol{x}_i^\top\boldsymbol{\beta}_0 + \frac{1}{p}(\|\boldsymbol{x}_i\|_2^2 - \mathrm{tr}[\boldsymbol{\Sigma}_{\mathrm{ar1}}]) + \varepsilon_i, \quad \boldsymbol{x}_i = \boldsymbol{\Sigma}_{\mathrm{ar1}}^{\frac{1}{2}}\boldsymbol{z}_i, \quad z_{ij} \overset{iid}{\sim} \frac{t_5}{\sigma_5}, \quad \varepsilon_i \sim \frac{t_5}{\sigma_5}, \quad \text{(M-AR1)}$$

where $\sigma_5 = \sqrt{5/3}$ is the standard deviation of $t_5$ distribution. The benefit of using the above nonlinear model is that we can clearly separate the linear and the nonlinear components and compute the quantities of interest because $\boldsymbol{\beta}_0$ happens to be the best linear projection.

For the simulations, we set $\rho_{\mathrm{ar1}} = 0.5$. For finite ensembles, the risks are averaged across 50 simulations.

### F.3 Risk comparisons along equivalence paths for finite ensembles

Figure F5 verifies the following linear (in $M$) relationship along the path in Theorem 3 for finite ensembles: $R(\widehat{\boldsymbol{\beta}}^{\lambda_1}_{\lfloor p/\psi_1 \rfloor, M}; \boldsymbol{A}, \boldsymbol{b}, \boldsymbol{\beta}_0) - R(\widehat{\boldsymbol{\beta}}^{\lambda_2}_{\lfloor p/\psi_2 \rfloor, M}; \boldsymbol{A}, \boldsymbol{b}, \boldsymbol{\beta}_0) \simeq \Delta/M$, for some $\Delta$ (independent of $M$), which is eventually almost surely bounded.

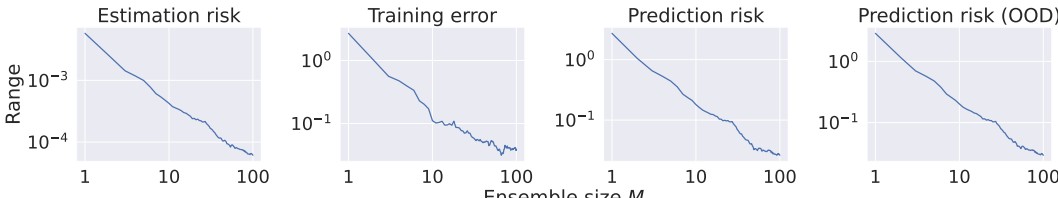

Figure F5: The range (the difference between the maximum and the maximum along the path) of the finite-sample generalized risk of $M$-ensemble ridge estimators on the path through $(\lambda, \psi) = (0, 2)$, for varying ensemble size $M$ under the same setting as in Figure 1. The generalized risks include the estimation risk, the training error, the prediction risk, and the out-of-distribution (OOD) prediction risk.

### F.4 Illustration of risk monotonicity of optimal ridge

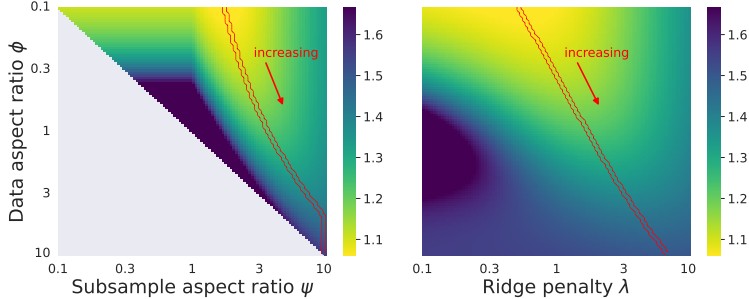

Figure F6: Illustration of risk monotonicity of optimal subsampled ridgeless regression versus optimal regularized ridge regression. The underlying model has a non-isotropic covariance as described in Appendix F.2 with a linearized signal-to-noise ratio of 0.6 and $\|\boldsymbol{f}_{\mathrm{NL}}\|_{L^2} = 1$. This results in a null risk of 1.6, which is kept the same across all regression problems. The left panel shows the limiting risk of the full-ensemble ridgeless regression at various data and subsample aspect ratios $(\phi, \psi)$. The right panel shows the limiting risk of the ridge predictor (on the full data) at various data aspect ratios and regularization penalties $(\phi, \lambda)$. Optimal risks for each data aspect ratio are highlighted using slender red lines in both panels. Observe that the optimal risk in both cases is increasing as a function of $\phi$. In the left panel, in addition, for every subsample aspect ratio $\psi$, the risk of subsampled ridgeless is increasing in the data aspect ratio $\phi$. This in turn implies the risk monotonicity claim for the optimal ridge regression in Theorem 6 through a slick triangle inequality argument (see the proof in Appendix C for more details).

### F.5 Real-world datasets

We conduct experiments on real-world datasets to examine the equivalence in a more general setting. We utilized three image datasets for our experimental analysis: CIFAR 10, MNIST, and USPS [59]. For the CIFAR 10 dataset, we subset the images labeled as "dog" and "cat". For other datasets, we subset the images labeled "3" and "8". Then we treat them as binary labels $y \in \{0, 1\}$ and use the flattened image as our feature vector $\boldsymbol{x}$. The training sample sizes, the feature dimensions, and the test sample sizes $(n, p, n_{\mathrm{te}})$ are (10000, 3072, 2000), (12873, 784, 2145), and (22358, 3072, 7981) for the three datasets, respectively.

For the first experiment in Figure 3, we fix $\phi = p/n$ and $\overline{\psi} = 4\phi$ using the training set of CIFAR-10. For $\lambda_{\overline{\psi}} \in \{0.01, 0.05, 0.1, 1\}$, we compute the data-dependent value of $\overline{\lambda}$ based on Proposition 4. Each value of $\lambda_{\overline{\psi}}$ gives a path between $(\lambda_{\overline{\psi}}, \overline{\psi})$ and $(\overline{\lambda}, \phi)$.

For the second experiment in Figure F7, by varying the values of subsample aspect ratios $\overline{\psi}$, we compare the random Gaussian linear projection of the estimators and the prediction risk at the two endpoints $(\overline{\lambda}, \phi)$ and $(0, \overline{\psi})$, on CIFAR-10, MNIST, and USPS datasets.

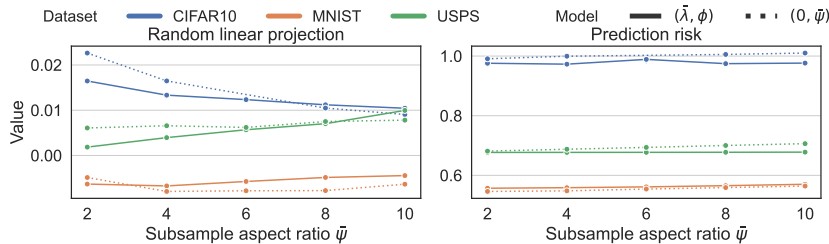

Figure F7: The functional equivalences of subsampling and ridge regularization on different datasets. For each dataset with an aspect ratio $\phi = p/n$ and each value of $\overline{\psi}$, the corresponding ridge penalty $\overline{\lambda}$ is estimated by Proposition 4. The two models with ridge penalty and subsample aspect ratio $(\overline{\lambda}, \phi)$ and $(0, \overline{\psi})$ are compared. Note that the model corresponding to $(\overline{\lambda}, \phi)$ is ridge regression at ridge penalty $\overline{\lambda}$ without subsampling, and the model corresponding to $(0, \overline{\psi})$ is full-ensemble ridgeless at a subsample aspect ratio $\overline{\psi}$.

## F.6  Random features model

The data $(\boldsymbol{x}_i, y_i)$ is generate from the nonlinear model

$$y_i = \boldsymbol{x}_i^\top \boldsymbol{\beta}_0 + \frac{1}{p}(\|\boldsymbol{x}_i\|_2^2 - p) + \varepsilon_i,$$

where $x_{ij} \overset{iid}{\sim} \mathcal{N}(0, 1)$, and $\varepsilon_i \sim \mathcal{N}(0, 1)$. The prediction risk of the ridge ensemble ($M = 100$) is computed based on the random feature $\varphi(\boldsymbol{F}\boldsymbol{x}_i)$ and the response $y_i$, where $\boldsymbol{F} \in \mathbb{R}^{d \times p}$ is the random weight matrix with $F_{ij} \overset{iid}{\sim} \mathcal{N}(0, p^{-1})$. Here, $\varphi$ is a nonlinear activation function (sigmoid, ReLU, or tanh) that is applied entry-wise to $\boldsymbol{F}\boldsymbol{x}_i$. For the experiment, we set $p = 250$, $d = 500$, and $\phi = d/n = 0.1$.

## F.7  Kernel ridge regression

For a given feature map $\Phi : \mathbb{R}^p \to \mathbb{R}^d$, the kernel ridge estimator is defined as:

$$\widehat{\boldsymbol{\beta}}_k^\lambda(\mathcal{D}_I) = \underset{\boldsymbol{\beta} \in \mathbb{R}^p}{\operatorname{argmin}} \sum_{i \in I} (k^{-1/2} y_i - k^{-1/2} \Phi(\boldsymbol{x}_i)^\top \boldsymbol{\beta})^2 + \lambda \|\boldsymbol{\beta}\|_2^2 \sum_{i \in I} (y_i - \Phi(\boldsymbol{x}_i)^\top \boldsymbol{\beta})^2 + \frac{k}{p} \lambda \|\boldsymbol{\beta}\|_2^2.$$

By using the kernel trick, the above optimization problem is equivalent to solving:

$$\widehat{\boldsymbol{\alpha}}_k^\lambda(\mathcal{D}_I) = \underset{\boldsymbol{\alpha} \in \mathbb{R}^k}{\operatorname{argmin}} \, \boldsymbol{\alpha}^\top (\boldsymbol{K}_I + k\lambda \boldsymbol{I}_k) \boldsymbol{\alpha} + \boldsymbol{\alpha}^\top \boldsymbol{y}_I,$$

where $\boldsymbol{K}_I = \boldsymbol{\Phi}_I \boldsymbol{\Phi}_I^\top \in \mathbb{R}^{k \times k}$ is the kernel matrix and $\boldsymbol{\Phi}_I = (\Phi(\boldsymbol{x}_i))_{i \in I} \in \mathbb{R}^{n \times d}$ is the feature matrix. Simple calculation shows that $\widehat{\boldsymbol{\beta}}_k^\lambda(\mathcal{D}_I) = \boldsymbol{\Phi}_I^\top \widehat{\boldsymbol{\alpha}}_k^\lambda(\mathcal{D}_I)$.

Using the same data-generating process as in the previous subsection, the results on kernel ridge regression are illustrated in Figure F8.

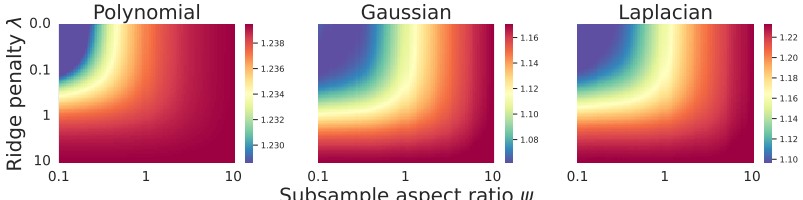

Figure F8: Heat map of the empirical distribution prediction risk of full-ensemble kernel ridge estimators, for varying ridge penalties $\lambda$ and subsample aspect ratio $\psi = p/d$ on the log-log scale. The prediction risk of the ridge ensemble ($M = 100$) is computed using polynomial, Gaussian, and Laplacian kernel, using the default parameters as in Python package `scikit-learn` v1.2.2 [60] without the intercept terms. We scale $\lambda$ by $k/p$ so that we can obtain non-null estimators for the polynomial inner-product kernel (of degree 3).

