# OpenReview forum: "Generalized equivalences between subsampling and ridge regularization"
_NeurIPS.cc/2023/Conference — NeurIPS 2023 poster_

### Official Review · Reviewer_96Gw · 2023-07-04

**Soundness:** 3 good
**Presentation:** 3 good
**Contribution:** 3 good
**Rating:** 7
**Confidence:** 3

**Summary:**

The authors investigate the problem of ridge regression, and prove equivalence results between ridge regularisation and ensambling of weak learners trained on subsamples of the original dataset.
The equivalences hold under very mild assumptions, and notably there is no requirement on the data model.
Two kind of equivalence are proven: i) equivalence at the level of a quite generic class of risks ii) equivalence at the level of the ridge estimator itself.
The equivalence basically say that one can trade a bit of subsampling for a bit of ridge regression without altering the performance of the estimator.

The equivalences hold on paths in the plane defined by the ridge regulariser and the subsampling ratio.
The authors provide both a "theoretical" characterisation of such paths, which requires knowledge of the population covariance of the features, and a "data-driven" characterisation, which requires only access to the sample covariance of the features.

Finally, the authors discuss possible extensions of their results to real-data scenarios and random features regression.


**Strengths:**

The works seems sound, relevant (answering open questions in the literature), well-motivated and well-presented.
Code is available for reproducibility.

I would like to highlight that the authors need basically no structural assumptions on the data model to prove the equivalence.


**Weaknesses:**

I did not identify any substantial weakness in the paper.


**Questions:**

I do not have any question of the authors.
I only suggest the authors to improve Figure 3: the caption could additionally describe the difference in x-axis between left and right panel.

**Limitations:**

The authors do not discuss explicitly limitations. I find that this is not strongly necessary, as the assumptions under which the results hold are clearly presented.
The only improvement I could suggest is for the authors to discuss whether they expect the equivalences they prove to break in some specified setting.

---

> ### Author Rebuttal · Authors · 2023-08-03
>
> Many thanks the nice summary, positive feedback, and the thought-provoking comment about plausible variations of the equivalence paths or even non-equivalences! We are glad that you found the paper interesting and appreciate the kind words.
>
> Below we address the question and limitation raised.
>
> - **[Response to question]** _(Clarification of Figure 3 caption)_: Thanks for the suggestions!
> We have now updated the caption of Figure 3 with additional clarifications on the linear functionals in the left panel and the risk functionals in the right panel and added pointers to the corresponding theorems.
>
> - **[Response to limitation]** _(Plausibility and conditions for different equivalence paths or non-equivalences)_: This is very interesting and thought-provoking comment! With i.i.d. samples, we expect the results to hold more generally, even beyond RMT features; see the discussions in Section 6 of the submitted paper.
> However, we acknowledge that the equivalences could potentially break, particularly if the observations are not sampled in an i.i.d. manner.
> For instance, if the observations follow a specific dependence structure, such as in a time-series setting, then the equivalence paths could be influenced by this dependence structure.
> Furthermore, we expect different variations of the equivalence paths under different sampling strategies, such as sampling with (or without) replacement within each subsampled dataset (and across all subsampled datasets).
> A precise characterization of these variants is an interesting direction for future work.
> Thanks again for this instigating this thought!

---

> > ### Comment · Reviewer_96Gw · 2023-08-10
> >
> > The authors addressed the only minor point I raised.
> > I confirm my initial review and grading, and thank the authors for replying to my curiosity on the possible breaking of the equivalences they prove.

---

### Official Review · Reviewer_MtFr · 2023-07-05

**Soundness:** 4 excellent
**Presentation:** 4 excellent
**Contribution:** 4 excellent
**Rating:** 10
**Confidence:** 5

**Summary:**

This paper shows an asymptotic equivalence between an ensembled+subsampled (E+S) version of ridge regression and the standard version, in the proportional asymptotic regime. The equivalence result shows that there exists a linear path in the space (ridge-parameter, aspect-ratio) along which all estimators yield essentially the same solution.

This equivalence also resolves an important open question regarding the behavior of optimally tuned ridge regularization. Specifically, the generalization error monotonically decreases with overparameterization (assuming the same level of SNR).

**Strengths:**

This paper was delightful to read.

The problem considered is of very broad interest. The contribution is fundamental and is potentially path-breaking. At the very least, it yields a very satisfactory understanding of the effect of ridge regularization under both important settings.
- optimal tuning
- interpolation (lambda = 0)

**Weaknesses:**

The paper can do with a round of proof-reading. There are some minor issues.

Perhaps the title should read "ensembled subsampling" instead of just "subsampling".

A sketch of the proof is missing. It would be good to highlight in a couple of paragraphs the core ideas behind the proof of the main result.

**Questions:**

In Section 2, does M have to grow at a certain rate wrt n ?

It appears that the Conjecture regarding Kernel Ridge Regression may already be within the reach of this paper, following the equivalence result from Sahraee-Ardakan et al. https://arxiv.org/pdf/2201.08082.pdf

**Limitations:**

No negative impact envisioned.

---

> ### Author Rebuttal · Authors · 2023-08-03
>
> Many thanks for the excellent concise summary, encouraging comments, and kind words!
> We are delighted to hear that you found our paper enjoyable to read.
> Paper aside, we also find the structural and risk equivalences quite neat, in understating the effects of ridge regularization, and particularly in the context of optimal ridge tuning and corresponding risk monotonicity.
>
> Below we comment on the general weaknesses raised (abbreviated by **[W]**).
>
> - **[Response to W1]** _(Paper proofreading)_: Thanks, we are revising the manuscript, taking into account typos from other reviewers as well.
>
> - **[Response to W2]** _(Paper title)_: That is a good suggestion!
>     The reason why we opted for simply subsampling is that the structural equivalence result (Theorem 3) holds for any ensemble size $M$, including $M=1$ when there is no ensemble.
>     The generalized risk equivalence (Theorem 1) holds for the "infinite'' ensemble $M \to \infty$.
>     This is why we simply use `"subsampling,'' though we fully agree with the sentiment of the comment.
>
> - **[Response to W3]** _(Proof sketches)_:   Thanks for the suggestion!
>     We agree that proof sketches (at least for Theorems 1 and 3) would be beneficial to the readers.
>     However, due to space constraints, we were not able to include more in the main text of the submitted paper.
>     We will try to squeeze in key ideas in the additional space for the paper revision.
>     The core ideas behind the proof involve: 1) using asymptotic equivalences and calculus (instead of computing the actual risks), which lets us bypass the assumptions on features; and 2) generalizing certain linear and non-linear concentration results using rank-2 perturbations of the ridge resolvent, which lets us handle the uncorrelated non-linear component without assuming independence.
>
> Below we address the two questions raised (abbreviated by **[Q]**).
>
> - **[Response to Q1]** _(Growth rate of ensemble size)_: Good question!
>     We note that we do not actually need an ``infinite'' ensemble when definite the estimator in Section 2.
>     It suffices to consider an ensemble of all possible subsets $\binom{n}{k}$ of size $k$ out of a dataset of observations $n$, which we call a "full'' ensemble. One can show that for any fixed dataset $\mathcal{D}_n$, the ridge estimators fitted on the "full ensemble'' match that with the "infinite ensemble'' almost surely conditioned on $\mathcal{D}_n$, as mentioned in the paragraph after Equation (2) of the submitted paper.
>     For this result, we do not need asymptotics in $n$ or $p$ when defining the full-ensemble estimators in Section 2.
>
>     Now when we consider the equivalence results in Sections 3 and 4, they hold under proportional asymptotics.
>     The result for structural equivalence (in Section 4) holds for any ensemble size $M\in\mathbb{N}$.
>     Thus for this result, one does not need to consider varying $M$.
>     The result for risk equivalence (in Section 3) holds for the infinite ensemble as $M \to \infty$ or equivalently for a full ensemble.
>     Thus for this result, $M$ changes with $n$ as $M(n) = \binom{n}{k}$.
>     When $n=k$, this does not grow to infinity.
>     In other words, one may not have "growing'' number of distinct subsets of size $k$.
>
> - **[Response to Q2]** _(Subsampling equivalences for kernel ridge regression)_: Thanks for the reference!
>     It indeed looks very relevant, and we are working on the kernel extension, and we will add the reference [SAEPRF22] to the list.
>     Because under proportional asymptotics, the behavior of the kernel ridge resolvent is similar to the "linearized'' kernel, the conjecture indeed looks within reach, as you suggested. We are also looking into a similar conjecture (Conjecture I.1 in the submitted paper) for random features regression using universality ideas (e.g., from [HL22], among others).
>
> **References**
>
> [SAEPRF22] Sahraee-Ardakan, Mojtaba, Melikasadat Emami, Parthe Pandit, Sundeep Rangan, and Alyson K. Fletcher. Kernel methods and multi-layer perceptrons learn linear models in high dimensions. arXiv preprint arXiv:2201.08082, 2022.
>
> [HL22] Hu, Hong, and Yue M. Lu. Universality laws for high-dimensional learning with random features. IEEE Transactions on Information Theory, 2022.

---

> > ### Comment · Reviewer_MtFr · 2023-08-21
> > **Thank you for the rebuttal**
> >
> > I am happy with my initial assessment of the paper. Please make the necessary revisions to improve the quality of presentation.

---

### Official Review · Reviewer_nskG · 2023-07-06

**Soundness:** 4 excellent
**Presentation:** 3 good
**Contribution:** 3 good
**Rating:** 7
**Confidence:** 2

**Summary:**

The authors study the relationship between  ridgeless ensembles constructed from subsampled data and a ridge estimator in a setting with mild assumptions on the joint distribution $(Y,X)$.  They establish equivalences for a generalized class of risk functionals, which include quantities related to coefficient estimation and both in and out of sample errors. These equivalences are proven in the case where the ensemble includes estimators trained on all possible subsamples of a given size; the authors extend these results to equivalences between two finite ensembles. The authors use these equivalence results to settle a conjecture in a previous paper about risk monotonicity of ridge regression as a function of $p/n$.


**Strengths:**

- The authors substantially relax the distributional assumptions considered by previous work in the literature. This is conceptually important since it was not known how critical the linearity assumption is for these types of equivalences to hold.
- The theoretical analysis is both novel and technical, invoking various concepts from random matrix theory.
- Overall, the paper is well-written and well-organized.




**Weaknesses:**

While equivalences between ridge regression and subsampling are conceptually interesting, at this stage it appears that consequences for data analysis are a bit limited.  However, the authors generously discuss several potential extensions for which some of the tools developed in the paper may be helpful.


**Questions:**

- In Theorem 3, is the ensemble size $M$ fixed?
- While the paper is well-written overall, some additional exposition/clarification in certain parts may be helpful to readers.  For example, the authors could elaborate more on what they mean by first-order and second-order.  In addition, the coefficient confidence interval case can be explained more.  Also, before theorem 3, it is stated that "We can go a step further and ask if there exist any equivalences for the finite ensemble and if there are any equivalences at the estimator coordinate level between the estimators."  Do you mean that the conditions in this theorem have been previously shown to imply equivalences at the coordinate level and these conditions also imply equivalences with finite ensembles or do you mean something else?


**Limitations:**

The authors are quite forthcoming about various limitations of their work and possible future extensions.

---

> ### Author Rebuttal · Authors · 2023-08-04
>
> Many thanks for the encouraging comments and feedback!
> We are glad to hear that you liked the relaxing of distributional assumptions, the novelty of theoretical analysis, and the clarity of presentation. In the sequel, we will first comment on the weaknesses raised and then address the questions.
>
> Comments on weaknesses follow:
>
> - **[Response to W1]** _(Consequences for practical data analysis)_:  Thank you for the comment!
>     We acknowledge that the immediate practical application of our results may not be readily apparent, but the insights gained can inform practical data analysis.
>     Even though our paper primarily considers ridge regression, which may seem limited for real-world data analysis, it is good to note the close connections between ridge regularization and other forms of regularization methods, such as dropout regularization, noisy training, random sketched regression, among others, as mentioned in the related work.
> Understanding the relationship between subsampling and ridge may provide valuable insights into the behavior of subsampling and other implicit/explicit regularization methods.
> Moreover, our data-dependent characterization of this effect, as presented in Proposition 4, could potentially guide model selection involving other types of regularization for ensemble learning. This understanding could also potentially guide the selection of appropriate subsample sizes in practice.
>
>     Now it is true that the equivalence results are derived under certain assumptions on the features (RMT, in particular), we expect it to be true more generally.
>     For example, we really only need the concentration of the ridge resolvents.
>     Beyond RMT features, for the Marchenko-Pasture law to hold, which forms the backbone of of work, it is recently been shown (see [L22], [CM22], for example) that convex concentration suffices.
>     Thus we expect that the equivalences we establish in the paper to hold for a range of real data distributions.
>     For instance, as illustrated in Figure 3 of the paper, our equivalences seem to hold very well for real image datasets.
>    Formalizing precisely the extend to which these equivalences hold true is indeed on our list of future work!
>
>
> - **[Response to Q1]** _(Ensemble size for structural equivalences)_: Yes, Theorem 3 on structural equivalence works for any ensemble size $M$.
>     We will stress this in the revised version.
>
> - **[Response to Q2]** _(Clarifications on the terminology)_: Thanks for the suggestions!
>     We appreciate the comment and agree that the paper will benefit from additional clarifications on the terminology.
>
>     When we refer to first-order and second-order, we are referring to the order of the functional of the estimators.
>     First-order refers to the equivalence of linear functionals of the estimators.
>     Second-order, on the other hand, refers to the equivalence of quadratic functionals of the estimators.
>
>     Regarding your question about the commentary before Theorem 3, we will clarify it better.
>     When we mention equivalences at the estimator ``coordinate'' level, we mean that each coordinate (of the $p$-dimensional vector of the estimators) asymptotically equals.
>     Such a type of equivalence on finite ensembles has not been studied in the previous works.
>     As summarized in Table 1, the previous related papers only show certain risk equivalences under restricted assumptions.
>
> **References**
>
> [L22] Cosme Louart. Sharp bounds for the concentration of the resolvent in convex concentration settings. arXiv:2201.00284, 2022.
>
> [CM22] Chen Cheng and Andrea Montanari. Dimension free ridge regression. arXiv:2210.08571, 2022.

---

> > ### Comment · Reviewer_nskG · 2023-08-19
> >
> > Thank you for the interesting comments and clarifications in the rebuttal.  I am maintaining my score. Good luck!

---

### Official Review · Reviewer_wEHo · 2023-07-15

**Soundness:** 4 excellent
**Presentation:** 4 excellent
**Contribution:** 3 good
**Rating:** 6
**Confidence:** 3

**Summary:**

This submission establishes equivalences between ridge regression (i.e. $\ell_2$-penalized
linear regression) and ensembles of linear models trained on sub-sampled datasets.
In particular, the authors prove that for a fixed feature/sub-sample-size $d/k$, ratio, there
exists a ridge-regression model with risk asymptotically equivalent to the full
ensemble, i.e. the average of all $k$ models trained on sub-samples of size $k$.
Moreover, this equivalence holds for all convex combinations of the
ridge-regression and full sub-sampled model.
Note that the asymptotics assume $d$, $k$, $n$ approach infinity such that
the ratios $d/k$ and $d/n$ are held constant.
This equivalence result is then extended to other metrics, such as training error
and the weight estimation error, and to "structural" results which
show asymptotic equivalence of the weights of the models.
The authors leverage these equivalences to show that the prediction
risk of the best ridge regression model is monotone increasing in $d/n$.

**Strengths:**

The main strength of this work is the theoretical contributions. In
particular:

- The authors extend exist results on equivalent risk of ridge regression
    models and sub-sampled ensembles to new metrics and to the model weights
    themselves ("structural results").

- The theorems are proved under relaxed conditions compared to previous work.
    In particular, general data distributions are allowed provided a fairly
    weak assumption on the moments holds.

- The authors answer an open problem on the behavior of risk for ridge regression
    models with the optimal regularization constant.

The paper is also well written, with very few typos. I congratulate the authors
on their polished manuscript.
Related work appears to be correctly cited, although this is not my research
area so it is hard for me to check.
Note that I did not check the theoretical derivations in the appendix so I
cannot comment on their correctness.

**Weaknesses:**

The greatest weakness of this paper is that the theoretical results are
somewhat incremental and unlikely to have an impact outside of learning theory.
In particular,

- The connection between ridge regression and sub-sampled ensembles was
    previously established, so that the main contributions of this work
    are weakened conditions and new types of equivalences.

- It's not clear how interesting it is to answer the conjecture from
    Nakkiran et al. While the authors prove that the risk for the ridge
    regression model with optimal regularization constant is monotone
    increasing in the ratio $d/n$, this only applies to the asymptotic
    regime and so its practical importance may be limited.

- While the authors motivate their work by highlighting connections between
    ridge regularization and dropout, noisy training, data augmentation,
    and early stopping. However, these connections are not developed any further
    and I am skeptical the asymptotic equivalences in this submission will
    impact those areas.

**Questions:**

Line 93: Is this supposed to mean that Assumption 2 defines RMT features?
    It's not obvious because the acronym RMT is not defined anywhere and
    not used in Assumption 2.

Line 176: I suggest Changing the name of Assumption 2 from "Feature Vector Distribution"
    to "RMT" features as well as including a definition for the initialism/acronym, since
    it isn't stated anywhere.

Line 156 and Definition E.1: Does $C_p$ need to be bounded away from zero?
    Otherwise the asymptotic
    equivalence definition will be meaningless when $C_p = 0$ almost surely
    for every $p$. Or perhaps when you say "any sequence $C_p$" you mean
    "every sequence"?

Line 183: Shouldn't this solution be $(\\lambda, \\bar{\\psi})$?

Figures 1/2: The paths between equivalent models don't appear to be linear
    in these figures, although Equation 5 seems to indicate that they are always
    linear combinations. Is this because the figures are in log-log scale?

Theorem 3: I don't understand what is "structural" or "first-order" in this
    theorem compared to Theorem 1. Is this because the estimators themselves
    are equivalent, rather than a risk functional of the estimators?
    While is equivalence of risk functionals "second order"?

Theorem 3: "this implies that the predicted values (or even any continuous function
applied to them due to the continuous mapping theorem) of any test point will eventually be the same,
almost surely, with respect to the training data."

The asymptotic equivalence of parameters in Theorem 3 and this
statement seem to imply that Theorem 3 covers both Theorem 1 and Proposition 2
by taking $M = \\infty$ --- is this correct?
If yes, what is the novelty of these two previous results given Theorem 3?

Proposition 4: Is Equation 8 always guaranteed to admit a solution?
    Moreover, what is the utility of this result when the equivalence only
    holds asymptotically? That is, when $n \rightarrow \infty$ and the
    root-finding problem defined by Equation (8) is impractical to solve.

**Limitations:**

As mentioned in the "Weaknesses" section, I think the main limitation of this
submission is the impracticality of the main theoretical results.
Since they extend and generalize a previous result showing asymptotic equivalence
of the risk, I do not see the theoretical derivations having an impact on practice.
Furthermore, the connections to high-interest topics in ML like early stopping
and dropout seem tenuous at best.

Since I am not actively involved in learning theory research, I cannot
comment on the importance of this work for other members of this community.
It would be nice if the authors could provide additional context for their
work, including some comments on the novelty of their proof techniques and so
on. That way I can better understand the impact on this specific research community.

---

> ### Author Rebuttal · Authors · 2023-08-05
>
> Many thanks for the detailed constructive feedback! We appreciate the careful reading and questions.
>
> Below we comment on the weaknesses within the allowed space.
>
> - **[W1]** While it is true that the connection between ridge regression and sub-sampled ensembles has been previously established, our work significantly extends this connection. Apart from generalizing results to hold general functionals, a key aspect of our results compared to prior work is that we do not assume any model for the response for any of these equivalences. We also establish structural equivalences, which were not studied in the prior works. This allows our theoretical results to be applicable to practical data analysis. Furthermore, our data-dependent method for determining equivalent paths is also another novel contribution, which we believe has practical implications as it allows one to determine the level of ``induced'' ridge regularization based on the available data. Please also see our response to **[Q1]** of reviewer **DNS5** for technical novelties and to **[W1]** of reviewer **nskG** for practical impact.
> - **[W2a]** Many common methods, such as ridgeless or lassoless regression, have been recently shown to exhibit non-monotonic behavior in the sample size or the limiting aspect ratio. In this regard, the conjecture from Nakkiran et al. is interesting because the non-monotonicity risk behavior implies that more data can hurt the performance, and it's important to investigate whether optimal ridge regression also suffers from this risk.
> - **[W2b]** The conjecture on the monotonicity of optimal ridge regression is of interest, even in the asymptotic regime. This is because the monotonicity/non-monotonicity is largely governed by the structure and relationship between the bias and variance at various regularization levels. Understanding the bias-variance tradeoff at the optimal regularization is not as affected by the finite-sample effects. Furthermore, as can seen empirically verified, even for $n$ and $p$ in the order of 100, one starts to observe the asymptotic behavior. The asymptotic approach simplifies the proofs and allows us to focus on the essential characteristics of the problem, under minimal assumptions. The extension of our results to the finite sample regime is possible but requires additional assumptions and depends on the specific nature of the distribution of features and response. See also response to **[W2]** of reviewer **DNS5** for more details.
> - **[W3]** Our mention of connections to dropout, noisy training, data augmentation, and early stopping was to highlight the broad relevance of our focus on ridge regularization and subsampling. While the direct application of our asymptotic equivalences to these areas is not immediately clear, we believe our work provides insights that could indirectly inform these areas and inspire future work. For instance, understanding the trade-offs between subsampling and ridge regularization could potentially inform more effective strategies for dropout or data augmentation.
>
> Below we address the questions.
>
> - **[Q1]** We define these precisely in the discussion after Assumption 2. We will clarify this in the revision.
> - **[Q2]** We will make the names of the assumptions "Moment-bounded response" and "RMT features". We will also mention the acronyms explicitly when they first appear.
> - **[Q3]** The definition requires the convergence "for every sequence $C_p$ that is uniformly bounde".
> - **[Q4]** For (4), we fix $\bar{\psi}$ and try to determine the values of $\bar{\lambda}$ and $v$.
>     Here the tuple $(\bar{\lambda},v)$ is a solution to (4) when $\bar{\psi}$ is fixed.
> - **[Q5]** Yes, the figures are in the log-log scale, only for better illustration purposes. We mention this in both captions of the submitted paper.
> - **[Q6]** Yes, "structural equivalence" means that estimators themselves are equivalent, rather than a risk functional of the estimators. Theorem 3 states that ${\mathbf c}^{\top}(\hat{\mathbf{\beta}}\_1 - \hat{\mathbf\beta}\_2) \xrightarrow{a.s.} 0$ for every constant vector ${\mathbf c}$ with bounded norm. This is a linear functional of the difference, so we view this equivalence as a "first-order" result. On the contrary, Theorem 1 and Proposition 2 compare the distance between two estimators to the ground truth ${\mathbf\beta_0}$, i.e., $\|\|\hat{\mathbf\beta}\_1 - {\mathbf\beta_0}\|\|\_{\mathbf A}^2 - \|\|\hat{\mathbf\beta}\_2 - {\mathbf\beta_0}\|\|\_{\mathbf A}^2 \xrightarrow{a.s.} 0$ where $\mathbf A$ is a weight matrix. Hence the latter is in the "second-order" sense.
> - **[Q7]** Theorem 3 does not imply Theorem 1 and Proposition 2. When ${\mathbf A} = {\mathbf c}{\mathbf c}^{\top}$, the latter reduces to $({\mathbf c}^{\top}\hat{\mathbf\beta}_1)^2 - ({\mathbf c}^{\top}\hat{\mathbf\beta}_2)^2 \xrightarrow{a.s.} 0$. But for general $\mathbf A$, there is no direct relationship between the two. For instance, when $\mathbf A=\mathbf I_p$, each coordinate of $\hat{\mathbf\beta}_1 - {\mathbf\beta_2}$ converges to zero almost surely, does not imply $\|\|\hat{\mathbf\beta}_1 - {\mathbf\beta_0}\|\|_2^2 \|\|\hat{\mathbf\beta}_2 - {\mathbf\beta_0}\|\|_2^2 = \|\|\hat{\mathbf\beta}_1\|\|_2^2 - \|\|\hat{\mathbf\beta}_2\|\|_2^2 +2{\mathbf\beta_0}^{\top}(\hat{\mathbf\beta}_1 - \hat{\mathbf\beta_2})$ also converges to zero almost surely.
> - **[Q8]** Yes, (8) always has at least one solution. This is because the RHS of (8) is monotonically decreasing in $\bar{\lambda}_n$, and the LHS always lies within the range of the RHS. One can solve (8) for a given set of $n$ and $p$. Since for sufficiently large $n$ and $p$, the equivalence holds, we can solve (8) in finite samples. While it is true, the theorem statement at the moment does not say anything non-asymptotically, we expect that one can derive a high-probability statement that explicitly characterizes a high probability bound on the difference between the two asymptotically equivalent quantities.

---

> > ### Comment · Reviewer_wEHo · 2023-08-14
> >
> > Many thanks for responding to my review and answering my questions.
> >
> > **[W2b]** Right, I agree that this can be interesting. However, it is a bit
> > strong to say "more data can hurt performance" when the results only apply to
> > the limiting ratio of $d/n$. It seems more accurate to say "more data relative
> > to features". I also think it is appropriate to qualify the claim of having
> > resolved this open question as per Reviewer DNS5's comment that this applies to
> > the proportional limit regime only
> >
> > **[Q6]** and **[Q7]** Thanks for clarifying these issues. I now follow why
> > these are first-order or "structural" results. I think it would be useful to
> > remind unfamiliar readers of the type of convergence proved after Theorem 3,
> > i.e. that it is not almost sure convergence of the estimator difference
> > but of linear functional of the difference.
> >
> > **[Q8]** Great. I think this is worth commenting on in the paper.
> >
> > Overall, I think this is a nice submission. The subject area is niche, but the
> > paper is well-executed and the author response has been helpful. I will
> > consider raising my score after the discussion with the other reviewers.

---

### Official Review · Reviewer_DNS5 · 2023-07-19

**Soundness:** 3 good
**Presentation:** 3 good
**Contribution:** 2 fair
**Rating:** 6
**Confidence:** 4

**Summary:**

This work compares the ridgeless ensemble and the ridge estimators in the proportional limit setting (i.e., $d/n\to \phi$). Prior works [11,12,13] show that these two estimators achieve the same out-of-sample risk. The main contribution of this paper has been to (1) weaken the assumptions and (2) broaden the equivalence from out-of-sample risk to other risks such as empirical risk, in-sample risk, and transfer learning risk. Based on this theory, this work also shows that the out-of-sample risk achieved by the optimal ridge estimator monotonically increases as a function of the data aspect ratio (i.e., $\phi$).



**Strengths:**

+ Excellent presentation. I have enjoyed going through the paper.
+ Compared to [11,12,13], this work has shown equivalence between ridgeless ensemble and ridge estimators in a broader sense and under weaker assumptions. Notably, in this work, the data is allowed to be misspecified and the limiting covariance spectrum needs not to exist, and the proved equivalence holds for transfer learning risk, in-sample risk, empirical risk beside out-of-sample risk.
+ Section 5 is a neat addition to the main results, showing that the optimal ridge induces an out-of-sample risk increasing with respect to the data aspect ratio.



**Weaknesses:**

- The Conjecture 1 in [1] is stated for every finite $n$ and $p$. Section 5 in this work only applies to the proportional limit setting, where $p/n\to \phi, n,p\to\infty$ for a finite $\phi$. While I think Section 5 is very interesting, I am not sure it is proper to claim that "This resolves a recent open problem raised by Nakkiran et al. [1] under general data distributions and mild regularity conditions."


- The theory is limited to the proportional limit regime. To what extent the theory holds in the finite sample regime is unclear.


- The technique novelty could be further clarified.

- I tried to read the proof and believe they are mostly correct. However, there are a number of typos that confuse me (and prevent me from spending more time checking the proof). For an incomplete list:

1. The equation after Line 514. Missing a factor of 2 in the cross-term.
2. Line 520. $B = AX - I$, missing $-I$.
3. Line 534. $v(- \lambda_1; \psi_1) = v(- \lambda_2; \psi_2)$. $\lambda_2$ instead of $\lambda_1$.
4. The first inequality after Line 622. Left hand side of the inequality, $R(0; \phi, \psi)$ should be $R(0; \phi_1, \psi)$?

**Questions:**

Upto my quick glance, it seems the proof methods are largely built upon existing works such as [12] and [13]. I understand that this work has derived broader equivalence results under weaker assumptions compared to [11,12,13]. Would you please clarify what are the new ingredients in this work that allow so?

---

> ### Author Rebuttal · Authors · 2023-08-05
>
> Sincere thanks for the positive feedback, insightful comments, and the list of typos.
> We are happy that you enjoyed reading our paper and appreciated the presentation (that we paid special attention to while writing the paper).
> To echo your sentiment, results in Section 5 are also our favorite!
>
> Below we will first provide our responses to the weaknesses raised (abbreviated by **[W]**).
>
> - **[Response to W1]** _(Finite-sample monotonicity of optimal ridge)_:
>     Thanks for the comment.
>     We agree that our claim regarding the resolution of the open problem (Conjecture 1) raised by Nakkiran et al. in [NVKM21] is perhaps overly broad.
>     Our results in Section 5 indeed apply in the proportional limit setting, where $p/n \rightarrow \phi$ for $\phi \in (0, \infty)$ as $n, p \rightarrow \infty$.
>     We acknowledge that this does not cover every finite $n$ and $p$.
>     We will add the qualifier that the results in Section 5 resolves the conjecture in the proportional asymptotic sense.
> However, we expect that all the asymptotic statements in the paper can be converted to non-asymptotic statements with distributional assumptions on the feature and response distributions.
>     Please see our response to **[W2]** for more details.
>     In that sense, we expect that our Theorem 6 largely covers the meat of general monotonicity claim, though of course the finite-sample analogues are important also.
> We will make sure to point it out.
>
> - **[Response to W2]** _(Non-asymptotic analogues of equivalences)_:
>     Thanks for the feedback.
>     Indeed, our current theoretical framework is primarily developed under the proportional limit regime.
>     This approach simplifies the proofs and allows us to focus on the essential characteristics of the problem.
>     Our empirical results, as shown in the Figures 3 in the paper, suggest that our results hold even for finite $n$ and $p$.
> These empirical findings provide practical validation of our theoretical results beyond the proportional limit regime.
>     The extension of our results to the finite sample regime is possible, but requires additional assumptions.
>     The precise error bounds in the finite sample regime would depend on the specific nature of the distribution of features and response.
>     We expect techniques of [KY17], [CM22], [WHS22], among others, to provide non-asymptotic versions of the main statements in our paper.
>
> - **[Response to W3]** _(Clarification of novelties)_: We will briefly recall our novelties below, compared to previous related works:
>
>     - Generalized risk framework:
>     We consider a generalized risk framework that allows us to evaluate the performance of ensemble ridge estimators relative to the oracle parameter.
>     This is a significant departure from previous works that only focus on specific risk functionals.
>
>     - Structural equivalences:
>     We show structural equivalences in the form of linear functionals of the ensemble ridge estimators. This aspect is not explored in any of the previous papers.
>
>     - Data-dependent method for equivalent paths:
>     We provide a data-dependent method to determine the equivalent paths of $(\lambda, \phi)$.
>     We believe this is an important contribution as it allows us to apply our theoretical findings in practice.
>
>     - Technical novelties: The new technical tools we develop allow us to relax the assumptions on features and responses.
>     See also response to **[Q1]** below for more details on technical novelties.
>
> - **[Response to W4]** _(Typos)_
>     Thanks so much for the list.
>     We have corrected these in our revision.
>
> Next we address the question (abbreviated by **[Q]**).
>
> - **[Response to Q1]** _(Clarification of technical novelties)_: While it is true that our work builds upon the foundations laid by previous studies, including [12] and [13], it is important to note that our contributions are not merely extensions of these works and requires several technical novelties. We elaborate below.
>
>    To derive the equivalence result under minimal assumption on the feature distribution and feature covariance, we use the notion of asymptotic equivalence to compare sequences of matrices of arbitrary dimensions. Using asymptotic equivalences and associated calculus (instead of computing the actual risks) lets us bypass the assumptions on features.
>
>    To accommodate any nonlinear dependence structure between $y$ and $\mathbf{x}$, we generalize the linear and quadratic concentration results to allow for arbitrary models. In contrast, in all previous works, a well-specified linear model $y=\mathbf{x}^{\top}\beta_0 + \epsilon$ is assumed. At a high level, the proof of these results requires rank-2 perturbations of the ridge resolvents to separate out the dependent and independent parts, and exploiting the property of uncorrelation between the features and the nonlinear residuals (after projecting out the linear part). Please see Appendix D of the submitted paper for more details.
>
> **References**
>
> [NVKM21] Preetum Nakkiran, Prayaag Venkat, Sham M. Kakade, and Tengyu Ma. Optimal regularization can mitigate double descent. In International Conference on Learning Representations, 2021.
>
> [KY17] Antti Knowles and Jun Yin. Anisotropic local laws for random matrices. Probability Theory and Related Fields, 2017.
>
> [CM22] Chen Cheng and Andrea Montanari. Dimension free ridge regression. arXiv:2210.08571, 2022.
>
> [WHS22] Alexander Wei, Wei Hu, and Jacob Steinhardt. More than a toy: random matrix models predict how real-world neural representations generalize. In International Conference on Machine Learning, 2022.

---

> > ### Comment · Reviewer_DNS5 · 2023-08-10
> > **Thank you for your response**
> >
> > Thank you for your response.
> >
> > This paper makes several extensions to prior works on the equivalence between ridgeless ensemble and ridge. However, the theoretical results in this work are still limited to the proportional regime as in prior works. So the contribution of this work is not super high given prior works from my perspective. Also, it is worth noting that this work partially resolves Conjecture 1 in [1] in the proportional limit regime (rather than in the finite-sample/finite-dimensional regime).
> >
> > Therefore, I'd like to maintain my initial review and rating.

---

### Decision · Program_Chairs · 2023-09-21

**Decision:**

Accept (poster)

**Comment:**

Overall, this is a solid and well-written paper on a core topic in statistical learning -- establishing a variety of equivalences between ridge regularization to subsampling in ensemble (ridge) estimators. It extends and improves prior literature by weakening assumption and by covering several kinds of learning/risk equivalences (e.g. empirical risk, out-of-sample risk, and transfer learning). It is well-written and synthesizes its results among existing ones in a way that ought to also be useful to readers outside of its sub-field.

One reviewer was especially enthusiastic about the paper overall. Their recommendation is balanced by others (e.g. reviewers DNS5, wEHo) who maintained reservations that the assumptions (asymptotics, the proportional limit regime, and so on) limit the scope of the result somewhat. Either way, I believe that the discussion with these reviewers led to some useful clarifications on the statement of contributions. For example, the discussion with reviewers DNS5 and wEHo stressed that the conjecture from Nakkiran et al is broader than what this paper resolves (since it is stated for finite dimensions, whereas this paper again focuses on proportional asymptotics). Meanwhile, the discussion with reviewer MtFr pointed out a new reference and conjecture to look into -- whether or not it turns out indeed within reach, it may be worth at least a remark.